# Partially Functional Dynamic Backdoor Diffusion-based Causal Model

## Abstract

Causal inference in settings involving complex spatio-temporal dependencies, such as environmental epidemiology, is challenging due to the presence of unmeasured confounding. However, a significant gap persists in existing methods: current diffusion-based causal models rely on restrictive assumptions of causal sufficiency or static confounding. To address this limitation, we introduce the Partially Functional Dynamic Backdoor Diffusion-based Causal Model (PFD-BDCM), a generative framework designed to bridge this gap. Our approach uniquely incorporates valid backdoor adjustments into the diffusion sampling mechanism to mitigate bias from unmeasured confounders. Specifically, it captures their intricate dynamics through region-specific structural equations and conditional autoregressive processes, and accommodates multi-resolution variables via functional data techniques. Furthermore, we provide theoretical guarantees by establishing error bounds for counterfactual estimates. Extensive experiments on synthetic data and a real-world air pollution case study confirm that PFD-BDCM outperforms current state-of-the-art methods.

## 1 Introduction

Causal inference fundamentally addresses interventional (*"What if?"*) and counterfactual (*"What would have happened?"*) questions that go beyond statistical correlations, proving valuable in high-stakes domains like healthcare for treatment effect estimation (Hill, 2011), policy evaluation without randomized trials (LaLonde, 1986). The field's core challenge stems from the fundamental problem of causal inference: the impossibility of simultaneously observing both an outcome under a treatment and the potential outcomes under the alternative treatment (or control) for the same unit (Imbens & Rubin, 2015), necessitating methods to overcome confounding bias in observational data. Traditional approaches including potential outcomes frameworks (Imbens & Rubin, 2015), propensity scoring (Rosenbaum & Rubin, 1983), instrumental variables (Angrist et al., 1996), and structural causal models (Pearl, 2009) exhibit significant limitations when handling modern complex datasets—they struggle with high-dimensional confounders, ethical constraints of randomized trials, scarcity of valid instruments, and requirement of known causal graphs. These limitations become particularly acute when confounders involve high-dimensional data like medical images or partially unobserved variables (Shalit et al., 2017).

Within the Structural Causal Model (SCM) framework, causal queries can be answered by learning a proxy for the unobserved exogenous noise and the structural equations (Pearl, 2009). This suggests that (conditional) generative models that encode to a latent space could be an option for modeling SCMs, as the latent space serves as proxies for exogenous noises. Recent advances have explored the integration of deep generative models with structural causal models to address these challenges. Neural Causal Models (NCMs) (Xia et al., 2022) leverage neural networks for causal inference but struggle with complex spatio-temporal dependencies. Causal Normalizing Flows (CNFs) (Khemakhem et al., 2021) employ invertible transformations but face limitations in high-dimensional settings. Decoupled Flows (DecaFlow) (Almodóvar et al., 2025) and Diffusion-based SCMs (Diff-SCM) (Mamaghan et al., 2023) represent recent advances but lack explicit handling of spatio-temporal unmeasured confounding. Chao et al. (2023) proposed the Diffusion-based Causal Model (DCM), which leverages diffusion processes to approximate structural equations and answer causal queries without explicit intervention data. However, DCM assumes causal sufficiency (no unobserved confounders) which rarely holds in practice. Shimizu (2023) extended this line of work

with Backdoor Diffusion-based Causal Model (BDCM), incorporating backdoor adjustment to handle certain types of unmeasured confounding. Nevertheless, both methods rely on static assumptions and thus do not fully capture the spatio-temporal structure inherent in confounding variables—a key characteristic of real-world systems, where such factors often demonstrate complex dependencies.

To address this gap, we propose the Partially Functional Dynamic Backdoor Diffusion-based Causal Model (PFD-BDCM), a generative framework capable of explicitly modeling spatiotemporal correlations among unmeasured confounders while supporting causal inference. Fig. 2 in Appendix B illustrates the core conceptual framework of our proposed model. We propose a framework that relaxes the causal sufficiency assumption by modeling dynamic confounding through structured latent representations, enabling robust causal estimation in non-stationary environments. Our approach formalizes spatio-temporal dependencies via latent variables encoding unmeasured confounders (Eqs. ( 2)-(4)), extending the Structural Causal Model paradigm to the Spatio-Temporal Dynamic SCM (ST-DSCM). To handle functional data prevalent in domains like atmospheric science, we incorporate basis-expanded representations as standard nodes, yielding the Partially Functional ST-DSCM (PFST-DSCM). Building on this causal structural prior, we employ Backdoor-adjusted Diffusion Causal Models (BDCM) to train individual nodes, achieving improved exogenous noise estimation via enhanced abduction. This enables accurate performance across all causal query types—observational, interventional, and counterfactual. The core innovation, termed PFD-BDCM, integrates PFST-DSCM's structural formalism with BDCM-based inference for reliable causal discovery. Key contributions of our study include:

**(1)** [Section 2] The proposed PFD-BDCM provides a unified framework for approximating both interventions (do-operator) and counterfactuals (abduction-action-prediction steps). It has a training procedure requiring only the dynamic causal graph and observational data, and the trained model enables: i) sampling from observational/interventional distributions; ii) precise counterfactual query resolution.

**(2)** [Section 3] Our theoretical analysis proves that the counterfactual estimates given by PFD-BDCM admit quantifiable error bounds under reasonable assumptions. i) It provides the first formal explanation for the performance gains of encoder-decoder architectures (e.g., diffusion models) in counterfactual querying through error bounds; ii) It extends to the more challenging multivariate case under an additional assumption and to diverse encoder-decoder models.

**(3)** [Section 4] We evaluated the performance of PFD-BDCM on three synthetic datasets of varying scales involving spatiotemporal dynamic structural equations and three types of causal queries. Namely evaluating the maximum mean discrepancy (MMD) for the generated with true observational and interventional queries, the mean squared error (MSE) for counterfactual queries. Experimental results demonstrate that PFD-BDCM consistently outperforms existing state-of-the-art methods (Chao et al., 2023; Shimizu, 2023) as well as PFD-DCM. Furthermore, we also demonstrate the strong performance of PFD-BDCM on a real-world atmospheric pollution dataset.

## 2 METHODOLOGY

We first introduce some useful notations and concepts.

**Notations:** To distinguish between the nodes in the causal graph and diffusion random variables, we use subscripts to denote graph nodes. Let $[n] := \{1, \cdots, n\}$ and $\dim(x)$ represents the dimension of $x$; let $\hat{u}_k^t$ be the exogenous noise at diffusion step $t$ in the forward process, with $\hat{u}_k := \hat{u}_k^T$, and $\hat{x}_k^t$ the endogenous variable at step $t$ in the reverse process, where $\hat{x}_k := \hat{x}_k^0$.

In causal inference, a confounder denotes a variable that causally influences both a treatment variable x and an outcome variable y, thereby inducing a non-causal association between them. **Observable Confounders** refer to confounders that can be measured, which permit adjustment through statistical methods such as stratification, matching or regression (Pearl, 2009). **Unobservable Confounders** denote latent variables that fulfill confounding criteria but resist direct measurement, which can potentially bias causal estimates when unaccounted for. **Exogenous noise**: Random disturbances specific to each variable, assumed to be independent across variables and independent of the causal structure. **Unobserved explanatory (explained) nodes** are unobserved confounder nodes that have no unobserved confounder nodes as its parent (descendant).

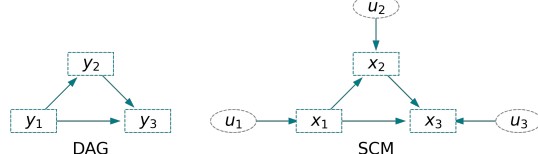

Figure 1: DAG with three nodes (left) and SCM with three exogenous and endogenous nodes (right)

**Structural Causal Models.** Consider a directed acyclic graph (DAG) (such as Fig. 1) $\mathcal{G}$ with nodes $[K]$ in a topologically sorted order [1], where a node $k$ is built with an endogenous random variable $x_k$ defined on a space $\mathcal{X}_k \subset \mathbb{R}^{d_k}$, which has a random exogenous input $u_k$. Let $\mathbb{PA}_k$ be the set of parent nodes of node $k$ in $\mathcal{G}$ and let $\mathbf{x}_{\mathbb{PA}_k} := \{x_l\}_{l \in \mathbb{PA}_k}$ be the set of parent random variables of $\mathbb{PA}_k$. A structural causal model (SCM) $\mathcal{M}$ characterizes the relationship between the endogenous variable $x_k$ of a node $k$ with the endogenous variables of its parents $\mathbf{x}_{\mathbb{PA}_k}$ and its own exogenous variable $u_k$. Formally, we define $\mathcal{M} := (\mathbf{f}(\mathbf{x}, \mathbf{u}), p_{\mathbf{u}})$, where $\mathbf{f}(\mathbf{x}, \mathbf{u})$ specifies how entire endogenous variables $\mathbf{x} := \{x_1, \cdots, x_K\}$ are generated from the set of exogenous random variables $\mathbf{u} := \{u_1, \cdots, u_K\}$ with a prior distribution $p_{\mathbf{u}}$. The structural mechanism is governed by $\mathbf{f}(\mathbf{x}, \mathbf{u}) := (f(\mathbf{x}_{\mathbb{PA}_1}, u_1), \cdots, f(\mathbf{x}_{\mathbb{PA}_K}, u_K))$, where each $x_k := f(\mathbf{x}_{\mathbb{PA}_k}, u_k)$ for $k \in [K]$ (Pearl, 2009).

Recent advances in deep learning-based causal inference (Chao et al., 2023; Shimizu, 2023), particularly a class of methods integrating structural causal models (Pearl, 2009) with generative models, have demonstrated the effectiveness of diffusion models for answering causal queries. However, spatial heterogeneity and temporal dependencies in unmeasured confounders undermine the validity of existing DCMs and BDCMs for genuine causal inference. For example, in environmental studies, pollution levels may vary significantly across regions (spatial heterogeneity) and exhibit serial correlation over time (temporal dependence), which standard causal models often fail to capture adequately. To overcome this fundamental limitation, we propose a Spatio-Temporal Dynamic Structural Causal Model (ST-DSCM) based on the Backdoor Criterion. We commence by establishing several essential definitions. Due to the complexity of the model, some symbols are mixed up in the article, but the overall readability remains unaffected.

## 2.1 Partially Functional Spatio-Temporal Dynamic Structural Causal Model

Suppose there is a spatio-temporal dataset $\mathbf{x} := \{x_k\}_{k \in [K]}$ containing $K$ variables across $n$ regions over $J$ time points. Using $i$ to index regions and $j$ for time points, $x_k = (x_{k,ij})_{n \times J}$ represent the value of the $k$-th variable $x_k$ over the time points and regions. We use a DAG to characterize the causal relationships among $\{x_k\}_{k \in [K]}$.

Consider a DAG $\mathcal{G}$ with nodes $[K]$ and a topologically sorted order such that each node $k$ has the $x_k$ as the random variable. Let $\mathbb{C}_1$ and $\mathbb{C}_2 \subseteq [K]$ be two distinct sets of nodes with unobserved confounders, where $\mathbb{C}_1$ designates a set of unobserved explanatory nodes and $\mathbb{C}_2$ denotes a set of unobserved explained nodes. For $h = 1, 2$, let $\mathbf{x}_{\mathbb{C}_h} := \{\mathbf{x}_{\mathbb{C}_h,ij}\}_{i \in [n], j \in [J]} := \{x_{q,ij}\}_{i \in [n], j \in [J], q \in \mathbb{C}_h}$, $\mathbf{u}_{\mathbb{C}_h} := \{\mathbf{u}_{\mathbb{C}_h,ij}\}_{i \in [n], j \in [J]} := \{u_{q,ij}\}_{i \in [n], j \in [J], q \in \mathbb{C}_h}$.

To incorporate spatio-temporal dynamic structures among unmeasured confounders, we assume spatiot-emporal dynamics SCM relations

$$x_{k,ij} = f_{k,ij}(\mathbf{x}_{\mathbb{PA}_k,ij}, u_{k,ij}), \quad \text{for } k \in \mathbb{C}_2, \mathbb{PA}_k \in \mathbb{C}_1. \tag{1}$$

where $f_{k,ij}$ spatiotemporal functions that are potentially linear or nonlinear, and $u_{k,ij}$ are independent exogenous noise terms. Specifically, to account for spatial heterogeneity, we posit a relationship such that

$$\mathbf{x}_{\mathbb{C}_2,ij} = \mathbf{\Gamma}_i \mathbf{G}(\mathbf{x}_{\mathbb{C}_1,ij}) + \mathbf{u}_{\mathbb{C}_2,ij}, \tag{2}$$

where $\mathbf{\Gamma}_i$ is a $\dim(\mathbf{x}_{\mathbb{C}_2,ij}) \times \dim(\mathbf{x}_{\mathbb{C}_1,ij})$ structural coefficient matrix, which are permitted to vary across regions; $\mathbf{G}(\cdot) = (G_1(.), \cdots, G_q(.))^\top$ is $q \times 1$ nonzero vector-valued function with differentiable functions $G_1, \cdots, G_q$, and $q = \dim(\mathbf{x}_{\mathbb{C}_1,ij})$. To capture the temporal dependence, we let $\mathbf{x}_{\mathbb{C}_1,i} = (\mathbf{x}_{\mathbb{C}_1,i1}^\top, \cdots, \mathbf{x}_{\mathbb{C}_1,iJ}^\top)^\top$ and $\mathbf{u}_{\mathbb{C}_2,i} = (\mathbf{u}_{\mathbb{C}_2,i1}^\top, \cdots, \mathbf{u}_{\mathbb{C}_2,iJ}^\top)^\top$, then, the covariance matrices $\text{Cov}(\mathbf{x}_{\mathbb{C}_1,i})$ and $\text{Cov}(\mathbf{u}_{\mathbb{C}_2,i})$ of $\mathbf{x}_{\mathbb{C}_1,i}$ and $\mathbf{u}_{\mathbb{C}_2,i}$ can be expressed as

$$\text{Cov}(\mathbf{x}_{\mathbb{C}_1,i}) = \mathbf{D}_{\mathbb{C}_1,i} \otimes \text{Cov}(\mathbf{x}_{\mathbb{C}_1,ij}), \ \text{Cov}(\mathbf{u}_{\mathbb{C}_2,i}) = \mathbf{D}_{\mathbb{C}_2,i} \otimes \text{Cov}(\mathbf{u}_{\mathbb{C}_2,ij}), \tag{3}$$

---

[1]A topological order is a linear arrangement of variables where a variable appears after all its direct causes (parents) (Pearl, 2009)

where $\boldsymbol{D}_{\mathbb{C}_h,i}(h \in \{1,2\})$ are the $J \times J$ adjacent time covariance matrices, $\mathrm{Cov}(\cdot)$ represent between-variable covariances. In order to establish a rigorous framework for the temporal adjacency structure, the conditional autoregressive (CAR) model (Besag, 1974) is adopted with the forms

$$\boldsymbol{D}_{\mathbb{C}_h,i} = (\boldsymbol{I}_J - \rho_{\mathbb{C}_h,i}\boldsymbol{H}_{\mathbb{C}_h,i})^{-1}, \tag{4}$$

where $\rho_{\mathbb{C}_h,i}(h \in \{1,2\})$ are adjacent time association parameters, and $\boldsymbol{H}_{\mathbb{C}_h,i}(h \in \{1,2\})$ are $J \times J$ adjacency matrices in which the element $h_{jl} = 1$ implies that time $l$ is adjacent to time $j$ and otherwise $h_{jl} = 0$.

For dependencies between other endogenous variables (not just explained nodes), if its corresponding parent nodes are observable nodes, we assume standard SCM relations:

$$\mathrm{x}_{k,ij} = f_k(\mathbf{x}_{\mathbb{PA}_k,ij}, \mathrm{u}_{k,ij}), \quad \text{for } k \notin \mathbb{C}_1 \cup \mathbb{C}_2, \mathbb{PA}_k \notin \mathbb{C}_1 \cup \mathbb{C}_2, \tag{5}$$

where $f_k$ are potentially linear (or nonlinear) static functions and $\mathrm{u}_{k,ij}$ are independent exogenous noise terms. And if the parent nodes are unobservable confounder nodes, we will introduce its set of backdoor nodes[2] as a proxy set for its parent node set. Specifically, Let $\mathbb{B}_k$ be the set of backdoor nodes of $k$, and $\mathbf{x}_{\mathbb{B}_k} := \{\mathrm{x}_l\}_{l \in \mathbb{B}_k}$ represent the variables on $\mathbb{B}_k$. We assume:

$$\mathrm{x}_{k,ij} = f_k(\mathbf{x}_{\mathbb{B}_k,ij}, \mathrm{u}_{k,ij}), \quad \text{for } k \notin \mathbb{C}_1 \cup \mathbb{C}_2, \mathbb{PA}_k \in \mathbb{C}_1 \cup \mathbb{C}_2. \tag{6}$$

Here, the set of parent nodes $\mathbb{PA}_k$ satisfies the definition of backdoor nodes, that is, $\mathbb{PA}_k \subseteq \mathbb{B}_k$.

However, real-world applications often present significant challenges in elucidating causal relationships among variables observed under heterogeneity. To address these complexities, we extend the ST-DSCM by incorporating functional random variables $\mathrm{z}(t)$ [3]. We adopt a basis expansion framework to achieve dimensionality reduction in the functional space via a set of orthogonal basis functions $\{\mathbf{b}_1, \cdots, \mathbf{b}_{K_n}\} \in \mathbb{R}^{T \times K_n}$. Let

$$\mathrm{x}_{m,ij} = \int \mathbf{b}_m(t)\mathrm{z}_{ij}(t)\, \mathrm{d}t, \tag{7}$$

for $i \in [n], j \in [J], m \in [K_n]$, which are used as nodes within the ST-DSCM, leading to a Partially Functional Spatio-Temporal Dynamic Structural Causal Model (PFST-DSCM) [4]. Detailed mathematical derivations are provided in Appendix A. This technique projects infinite-dimensional functional covariates onto a finite-dimensional space while preserving functional characteristics.

We assume that the unobserved random variables are jointly independent (Markovian SCM), and the Partially Functional Spatio-Temporal Dynamic Directed Acyclic Graph (PFST-DDAG) $\mathcal{G}$ is the graph induced by PFST-DSCM $\mathcal{M}$. Every PFST-DSCM $\mathcal{M}$ entails a unique joint observational distribution satisfying the causal Markov assumption: $q(\mathbf{x}) = \prod_{k=1}^{K} q(\mathrm{x}_k|\mathbf{x}_{\mathbb{B}_k})$.

Notably, our model operates in settings where both observational data and causal structures are available, enabling it to answer observational, interventional, and counterfactual queries. All parameters in the structural model are assumed to be known in this context; in practice, they can be estimated using methods from Song et al. (2012) and Tang et al. (2017).

## 2.2 PARTIALLY FUNCTIONAL DYNAMIC BACKDOOR DIFFUSION-BASED CAUSAL MODEL

We now present the PFD-BDCM, a model designed to handle causal queries under the PFST-DSCM framework, which explicitly accounts for unmeasured confounders with spatial heterogeneity and temporal dependence. The model employs an encoder-decoder architecture to enable causal reasoning across multi-resolution variables.

The PFD-BDCM's data-generating process is formalized as: If the node $k$ and its corresponding parent nodes are unobservable confounders, we will use Eq. (1) to generate node data $\mathrm{x}_k$. These

---

[2]A set of node $\mathbb{B}$ satisfies backdoor criterion (Pearl et al., 2016) for tuple $(X, Y)$ in DAG $\mathcal{G}$ if no node in $\mathbb{B}$ is a descendant of $X$ and $\mathbb{B}$ blocks all paths between $X$ (cause) and $Y$ (outcome) which contains an arrow into $X$.

[3]A functional random variable $\mathrm{z}(t)$ is defined as a random function over a continuous domain $t \in \mathcal{T}$, typically representing time or space (Ramsay & Silverman, 2005).

[4]For example, Fig. 3 is a PFST-DSCM with 33 exogenous and endogenous nodes, where nodes $\mathrm{x}_{28}$, $\mathrm{x}_{29}$ and $\mathrm{x}_{30}$ are unmeasured confounders with spatial heterogeneity and temporal dependencies, $\mathrm{z}_1(t), \mathrm{z}_2(t), \mathrm{z}_3(t)$ are functional nodes, and $\mathrm{x}_4, \cdots, \mathrm{x}_{21}$ are the corresponding base expansion nodes.

nodes are not the target for training, their data will only serve as structural assistance and bias adjustment during the training of the observable nodes of interest. Therefore, the diffusion model will not be trained on data from these nodes. If the node $k$ and its corresponding parent nodes are observable nodes, Eq. (5) will be used to generate node data $x_k$, and the encoder $g_k$ maps $(\mathbf{x}_{\mathbb{PA}_k}, x_k)$ to a latent variable $\hat{u}_k := g_k(\mathbf{x}_{\mathbb{PA}_k}, x_k)$, which captures information of the exogenous noise $u_k$. The decoder $h_k$ reconstructs $x_k$ as $\hat{x}_k := h_k(\mathbf{x}_{\mathbb{PA}_k}, \hat{u}_k)$. This scenario aligns with the data generation process of the DCM (Chao et al., 2023) . If the node $k$ is observable node and its parent nodes are unobservable confounder nodes, we employ Eq. (6) to generate $x_k$, the encoder $g_k$ maps $(\mathbf{x}_{\mathbb{B}_k}, x_k)$ to a latent variable $\hat{u}_k := g_k(\mathbf{x}_{\mathbb{B}_k}, x_k)$, and the decoder $h_k$ reconstructs $x_k$ as $\hat{x}_k := h_k(\mathbf{x}_{\mathbb{B}_k}, \hat{u}_k)$.

Our approach leverages the generative power of diffusion models to learn the complex functional relationships inherent in a Structural Causal Model. The core idea is to represent the structural equation for each endogenous variable $x_k$ with a dedicated conditional diffusion model. This model learns the distribution $p(x_k|\mathbf{x}_{\mathbb{PA}_k}, u_k)$ . Diffusion models (Sohl-Dickstein et al., 2015; Ho et al., 2020) approximate a target data distribution $q(x^0)$ via a two-stage process. First, a fixed **forward process** gradually injects Gaussian noise into the data $x^0$ over $T$ steps. The distribution of the noisy data $x^t$ at step $t$ is a Gaussian: $q(x^t|x^0) = \varphi(x^t; \sqrt{\alpha_t}x^0, (1 - \alpha_t)I)$, where $\varphi(x; \mu, \Sigma)$ denote the Gaussian density with mean $\mu$ and covariance $\Sigma$. Here, $\alpha_t := \prod_{s=1}^{t}(1 - \beta_s)$ where $\beta_s$ is a predefined noise schedule at each time step $s$. As $t \to T$, $x^T$ converges to a standard Gaussian distribution. Second, a learnable **reverse process**, parameterized by $\theta$, is trained to denoise the data. This is achieved by training a neural network $\epsilon_\theta$ to predict the added noise $\epsilon$ using the available observational data. For learning the distribution $p(x_k|\mathbf{x}_{\mathbb{PA}_k}, u_k)$, the network $\epsilon_\theta$ is trained on samples where $x_k$ serves as the target variable $x^0$ and $\mathbf{x}_{\mathbb{PA}_k}$ provides the conditioning context $c$. Here $\theta$ represents learnable parameters of the neural network, conditioned on the noisy data $x^t$, the step $t$, and the parental variables contextual information $c$. The objective function is

$$\mathbb{E}_{t, x^0, c, \epsilon}\left[\|\epsilon - \epsilon_\theta(\sqrt{\alpha_t}x^0 + \sqrt{1 - \alpha_t}\epsilon, c, t)\|^2\right]. \tag{8}$$

In our framework, the conditioning context $c$ for a variable $x_k$ is its set of parent variables $\mathbf{x}_{\mathbb{PA}_k}$ (or $\mathbf{x}_{\mathbb{B}_k}$).

Although diffusion models excel at data generation, causal inference, especially counterfactual reasoning, requires a deterministic mapping between observations and latent codes. Denoising Diffusion Implicit Models (DDIMs) (Song et al., 2021) provide such a deterministic non-Markovian reverse process, a property essential for identifiability. In PFD-BDCM, we extend DDIM by incorporating backdoor adjustment sets as additional covariates. The resulting diffusion model for node $k$ is denoted $\epsilon_\theta^k(x_k, \mathbf{x}_{\mathbb{B}_k}, t)$.

Formally, for each node $k \in [K]$, the latent variable $\hat{u}_k := \hat{u}_k^T$ is generated through the forward implicit diffusion process

$$\hat{u}_k^{t+1} := \sqrt{\alpha_{t+1}/\alpha_t}\hat{u}_k^t + \epsilon_\theta^k(\hat{u}_k^t, \mathbf{x}_{\mathbb{B}_k}, t)(\sqrt{1 - \alpha_{t+1}} - \sqrt{\alpha_{t+1}(1 - \alpha_t)/\alpha_t}), \tag{9}$$

for $t = 0, \cdots, T - 1$, initialized with $\hat{u}_k^0 := x_k$. This latent representation $\hat{u}_k$ serves as a proxy for the exogenous noise $u_k$. The reconstruction $\hat{x}_k := \hat{x}_k^0$ is obtained via the reverse implicit diffusion process

$$\hat{x}_k^{t-1} := \sqrt{\alpha_{t-1}/\alpha_t}\hat{x}_k^t - \epsilon_\theta^k(\hat{x}_k^t, \mathbf{x}_{\mathbb{B}_k}, t)\left(\sqrt{\alpha_{t-1}(1 - \alpha_t)/\alpha_t} - \sqrt{1 - \alpha_{t-1}}\right), \tag{10}$$

for $t = T, \cdots, 1$, initialized with $\hat{x}_k^T := \hat{u}_k$. The encoding and decoding functions for node $k$ are denoted as $\text{Enc}_k(x_k, \mathbf{x}_{\mathbb{B}_k})$ (Eq. (9)) and $\text{Dec}_k(\hat{u}_k, \mathbf{x}_{\mathbb{B}_k})$ (Eq. (10)) respectively, and the pseudocodes are provided in Appendix B.1.

**Training PFD-BDCMs.**

The comprehensive training methodology (Appendix B.2 Algorithm 3) incorporates backdoor adjustment sets as covariates while training distinct diffusion models per node. Crucially, generative models for endogenous nodes exhibit mutual independence during training, thereby enabling parallelized optimization. This parallelism is feasible since each diffusion model necessitates only its target node's values and corresponding backdoor adjustment set values. The final PFD-BDCM architecture integrates these $K$ trained diffusion models $\{\epsilon_{k,\theta}\}_{k \in [K]}$.

We now elucidate the methodology for leveraging trained PFD-BDCMs to approximate diverse causal queries. Resolution of observational and interventional queries necessitates sampling from their respective observational and interventional distributions. Counterfactual queries, however, operate at unit granularity by modifying structural equation assignments while preserving the latent exogenous noise variables consistent with empirical observations.

**Generating samples for observational/interventional queries.** Interventional queries concern the causal effect of actively setting a variable to a specific value, which is formally represented by the *do*-operator, as in $p(\mathrm{x}|\mathrm{do}(\mathrm{x}_\mathcal{L} := \boldsymbol{\gamma}))$, answering such queries requires sampling from the modified Specifically, to generate samples approximating the interventional distribution $p(\mathrm{x}|\mathrm{do}(\mathrm{x}_\mathcal{L} := \boldsymbol{\gamma}))$ using a trained PFD-BDCM model, we implement the following procedure: i) For intervened nodes $l \in \mathcal{L}$, we set $\hat{\mathrm{x}}_l := \gamma_l$ deterministically; ii) For root nodes $k$, sample $\hat{\mathrm{x}}_k$ from empirical training distributions; iii) For non-intervened nodes $k \notin \mathcal{L}$, sample latent vectors $\hat{\mathrm{u}}_k \sim \mathcal{N}(\mathbf{0}, I_{d_k})$, where $d_k = \dim(\mathrm{x}_k)$, and subsequently compute $\hat{\mathrm{x}}_k := \mathrm{Dec}_k(\hat{\mathbf{x}}_{\mathbb{B}_k}, \hat{\mathrm{u}}_k)$ utilizing inductively generated backdoor variable values $\hat{\mathbf{x}}_{\mathbb{B}_k}$. Generated values propagate to child nodes as backdoor inputs. Observational sampling ($p(\mathrm{x}_k)$) corresponds to $\mathcal{L} = \emptyset$, with pseudocode formalized in Appendix B.3 (Algorithm 4).

**Counterfactual Queries** concern hypothetical scenarios given actual observed outcomes, and require three steps: *abduction* (inferring exogenous noise), *action* (modifying equations), and *prediction* (simulating new outcomes). Specifically, to compute counterfactual estimates $\hat{\boldsymbol{x}}^{\mathrm{CF}}$ within the PFD-BDCM framework, given factual observation $\boldsymbol{x}^{\mathrm{F}} := (x_1^{\mathrm{F}}, \cdots, x_K^{\mathrm{F}})$ and intervention set $\mathcal{L}$ with values $\boldsymbol{\gamma}$, we implement the following systematic procedure: i) For intervened nodes $l \in \mathcal{L}$, assign $\hat{\mathrm{x}}_l^{\mathrm{CF}} := \gamma_l$ deterministically; ii) For non-intervened descendant nodes $k$, using inductively generated backdoor estimates $\hat{\mathrm{x}}_{\mathbb{B}_k}^{\mathrm{CF}}$, we compute $\hat{\mathrm{x}}_k^{\mathrm{CF}} := \mathrm{Dec}_k(\hat{\mathrm{x}}_{\mathbb{B}_k}^{\mathrm{CF}}, \mathrm{Enc}_k(\mathbf{x}_{\mathbb{B}_k}^{\mathrm{F}}, x_k^{\mathrm{F}}))$, where factual noise is implicitly encoded. The complete formalization appears in Appendix B.3 (Algorithm 5).

## 3 COUNTERFACTUAL ERROR BOUNDS

In this section, we establish the theoretical guarantees for the counterfactual estimation accuracy of the PFD-BDCM framework. The primary contribution is the derivation of an error bound that formally links the reconstruction fidelity of the encoder-decoder architecture to the precision of its counterfactual predictions. Our theoretical results encompass both special cases with additive noise structural equation models (Appendix C.2) and more general noise structures (Appendix C.3). Additionally, beyond discussing Gaussian noise assumptions, we further provide theoretical results for scenarios where the noise distribution follows a heavy-tailed distribution (Appendix C.4). Notably, these theoretical guarantees accommodate higher-dimensional settings (Appendix C.5). The main text presents our core theoretical framework for general noise structures, introducing the key assumptions and results. Formal proofs are presented in Appendix C.

Consider an endogenous variable $\mathrm{x}_k$ governed by structural equation $\mathrm{x}_k := f_k(\mathbf{x}_{\mathbb{B}_k}, \mathrm{u}_k)$ with backdoor adjustment set $\mathbf{x}_{\mathbb{B}_k}$ and exogenous noise $\mathrm{u}_k$. We analyze a single node without loss of generality (by permutation invariance of nodes), henceforth denoting the target variable as $\mathrm{x} \in \mathcal{X} \subseteq \mathbb{R}$, its backdoors as $\mathbf{x}_{\mathbb{B}} \in \mathcal{X}_{\mathbb{B}} \subseteq \mathbb{R}^K$, and exogenous noise as $\mathrm{u}$. The encoder-decoder architecture comprises

$$g : \mathcal{X} \times \mathcal{X}_{\mathbb{B}} \to \mathcal{U} \quad \text{(encoding function);} \quad h : \mathcal{U} \times \mathcal{X}_{\mathbb{B}} \to \mathcal{X} \quad \text{(decoding function),}$$

where $\mathcal{U}$ denotes the latent space. Within PFD-BDCM, $g$ and $h$ correspond to the Enc and Dec operators, respectively.

Our theoretical results rely on a set of assumptions regarding the structural equation and the encoder-decoder model. These conditions are essential for ensuring that the latent variable learned by the encoder can uniquely recover the unobserved exogenous noise, which is the cornerstone of accurate counterfactual estimation (Lu et al., 2020; Nasr-Esfahany & Kiciman, 2023; Nasr-Esfahany et al., 2023). For a variable $\mathrm{x} \in \mathcal{X} \subset \mathbb{R}$ with structural equation $\mathrm{x} := f(\mathbf{x}_{\mathbb{B}}, \mathrm{u})$ where $\mathrm{u} \perp\!\!\!\perp \mathbf{x}_{\mathbb{B}}$, we have the following assumptions:

**Assumption 1.** The encoding function produces representations that are statistically independent of the backdoor variables: $g(\mathbf{x}_{\mathbb{B}}, \mathrm{x}) \perp\!\!\!\perp \mathbf{x}_{\mathbb{B}}$.

**Assumption 2.** The encoding function is invertible and differentiable with respect to the endogenous variable for fixed backdoor variables: $\forall \mathbf{x}_{\mathbb{B}} \in \mathcal{X}_{\mathbb{B}}, \quad g(\mathbf{x}_{\mathbb{B}}, \cdot) : \mathcal{X} \to \mathcal{U}$ is bijective and $C^1$.

**Assumption 3.** The structural equation is continuously differentiable in both arguments and strictly monotonic in the exogenous noise for each fixed value of the backdoor variables

$$f \in C^1(\mathcal{X}_\mathbb{B} \times \mathcal{U}, \mathcal{X}), \quad \frac{\partial f(\mathbf{x}_\mathbb{B}, \mathbf{u})}{\partial \mathbf{u}} > 0 \quad \text{for all } \mathbf{x}_\mathbb{B} \in \mathcal{X}_\mathbb{B}, \mathbf{u} \in \mathcal{U}.$$

**Assumption 4.** The encoding function applied to observations generated by the structural equation yields a predictable transformation of the exogenous noise: $g(\mathbf{x}_\mathbb{B}, f(\mathbf{x}_\mathbb{B}, \mathbf{u})) = \phi(\mathbf{x}_\mathbb{B}, \mathbf{u})$, where $\phi(\mathbf{x}_\mathbb{B}, \cdot) : \mathcal{U} \to \mathcal{U}$ is invertible for each $\mathbf{x}_\mathbb{B} \in \mathcal{X}_\mathbb{B}$.

**Assumption 5.** The decoding function, when applied to the encoded representation, approximately reconstructs the structural equation output: $h(\mathbf{x}_\mathbb{B}, \phi(\mathbf{x}_\mathbb{B}, \mathbf{u})) = f(\mathbf{x}_\mathbb{B}, \mathbf{u}) + \epsilon(\mathbf{x}_\mathbb{B}, \mathbf{u})$, with bounded error $\|\epsilon(\mathbf{x}_\mathbb{B}, \mathbf{u})\| \leq \delta$ for all $\mathbf{x}_\mathbb{B} \in \mathcal{X}_\mathbb{B}, \mathbf{u} \in \mathcal{U}$, and $\delta > 0$ is a small constant .

**Assumption 6.** The decoder $h$ is Lipschitz continuous in its second argument, and the inverse transformation $\phi^{-1}$ is Lipschitz continuous in its second argument, uniformly over the backdoor variables

$$\|h(\mathbf{x}_\mathbb{B}, \hat{\mathbf{u}}_1) - h(\mathbf{x}_\mathbb{B}, \hat{\mathbf{u}}_2)\| \leq L_h \|\hat{\mathbf{u}}_1 - \hat{\mathbf{u}}_2\|,$$
$$\|\phi^{-1}(\mathbf{x}_\mathbb{B}, \mathbf{z}_1) - \phi^{-1}(\mathbf{x}_\mathbb{B}, \mathbf{z}_2)\| \leq L_\phi \|\mathbf{z}_1 - \mathbf{z}_2\|,$$

for all $\hat{\mathbf{u}}_1, \hat{\mathbf{u}}_2, \mathbf{z}_1, \mathbf{z}_2$ and all $\mathbf{x}_\mathbb{B} \in \mathcal{X}_\mathbb{B}$.

These assumptions, while formal, are well-motivated in the context of causal inference and deep generative models. Assumption 1 ensures that the encoder learns a pure representation of the exogenous noise uncontaminated by information from the backdoor variables. In practice, this can be enforced through regularization or architectural constraints. This is naturally satisfied in settings like additive noise models with $f(\mathbf{x}_\mathbb{B}, \mathbf{u}) = f^*(\mathbf{x}_\mathbb{B}) + \mathbf{u}$ where $\mathbf{x}_\mathbb{B}$ and $\mathbf{u}$ is independent. If the fitted model $\hat{f} \equiv f^*$, then $g(\mathbf{x}_\mathbb{B}, \mathbf{x}) = \mathbf{u}$. Assumption 2 is intrinsically satisfied by the bijective properties of deterministic diffusion architectures (Song et al., 2021), guaranteeing uniqueness in latent representations while preserving compatibility with standard implementations. Assumption 3 is satisfied by major identifiable model classes, including additive noise, post-nonlinear, and heteroscedastic formulations (Strobl & Lasko, 2023) while concurrently resolving symmetric noise ambiguities characteristic of observational data. This assumption further aligns with contemporary identifiability frameworks (Nasr-Esfahany & Kiciman, 2023) and intrinsically precludes non-identifiable structural equations. Additional justification of this assumptions' reasonableness in practical settings are provided in Appendix C.3. Assumption 4 formalizes the expectation that the encoder learns a consistent representation of the exogenous noise, though this representation may depend on the backdoor variables in a structured way. Further justification is provided in Appendix C.2. Assumption 5 allows for imperfect reconstruction while maintaining control over the approximation error. The bound $\delta$ quantifies the expressiveness of the decoder architecture. Assumption 6 ensures that small changes in inputs lead to bounded changes in outputs, providing stability guarantees.

Under these assumptions, we can prove that the encoder successfully isolates the exogenous noise up to an invertible transformation in Theorem 1.

**Theorem 1.** *Under Assumptions 1, 2, 3, and 4, the encoded latent variable is related to the true exogenous noise through an invertible transformation that may depend on the backdoor variables*

$$\hat{\mathbf{u}} = g(\mathbf{x}_\mathbb{B}, \mathbf{x}) = \phi(\mathbf{x}_\mathbb{B}, \mathbf{u}),$$

*where $\phi(\mathbf{x}_\mathbb{B}, \cdot) : \mathcal{U} \to \mathcal{U}$ is invertible for each $\mathbf{x}_\mathbb{B} \in \mathcal{X}_\mathbb{B}$.*

Theorem 1 provides the foundation for assessing the accuracy of counterfactuals given by the PFD-BDCM. It implies that the abduction step, $\mathrm{Enc}(\mathbf{x}_\mathbb{B}, \mathbf{x})$, correctly captures the essence of the unobserved exogenous random variable $\mathbf{u}$ that generated the factual observation. We now explore the direct consequences of this result. In an oracle scenario where the model achieves perfect reconstruction which means $h(\mathbf{x}_\mathbb{B}, g(\mathbf{x}_\mathbb{B}, \mathbf{x})) = \mathbf{x}$ holding almost surely (a.s.), the counterfactual estimate will be "perfect". This precise case implies the satisfaction of Theorem 1, yielding the relationship $h(\mathbf{x}_\mathbb{B}, \phi(\mathbf{x}_\mathbb{B}, \mathbf{u})) = f(\mathbf{x}_\mathbb{B}, \mathbf{u})$. Consequently, when making a counterfactual prediction for a new intervention $\mathbf{x}_\mathbb{B} := \gamma$, the model computes $h(\gamma, \phi(\mathbf{x}_\mathbb{B}, \mathbf{u}))$, which a.s. equates to the true counterfactual $f(\gamma, u)$. We consolidate these findings as Corollary 1 and provide the detailed proof in Appendix C.2. More practically, models are not perfect. Theorem 2 allows us to bound the counterfactual error by the model's reconstruction error. This is a powerful result, as it connects a measurable property of the model (how well it auto-encodes data) to its performance on a causal task.

**Theorem 2.** *Under Assumptions 1, 3, 4, 5, and 6, for any factual observation $(x^F, \mathbf{x}_\mathbb{B}^F)$ generated by $x^F = f(\mathbf{x}_\mathbb{B}^F, u)$ and any counterfactual intervention $\mathrm{do}(\mathbf{x}_\mathbb{B}^{CF} := \gamma)$, if the reconstruction error satisfies $\|h(\mathbf{x}_\mathbb{B}, g(\mathbf{x}_\mathbb{B}, \mathbf{x})) - \mathbf{x}\| \leq \tau$, then the counterfactual estimation error is bounded by*

$$\|\hat{x}^{\mathrm{CF}} - x^{\mathrm{CF}}\| \leq L_h \cdot L_\phi \cdot \tau + \delta,$$

*where $L_h$ and $L_\phi$ are the Lipschitz constants of $h$ and $\phi^{-1}$ respectively from Assumption 6, and $\delta$ is the decoder compatibility error from Assumption 5.*

This Theorem formally establishes that minimizing the reconstruction loss during training directly optimizes the model for better counterfactual prediction. For the special case of additive noise models, we establish particularly strong theoretical guarantees. In the Appendix C.2, we provide detailed theorems and proofs demonstrating that under the additive noise setting.

*Remark* 1. Theorem 3 shows that when the structural equations follow an additive noise model with Gaussian exogenous variables, the encoder exactly recovers the true exogenous noise, leading to theoretically optimal counterfactual predictions.

*Remark* 2. Corollary 2 shows that the counterfactual estimation error remains bounded by the same limit as the reconstruction error, even in the presence of bounded reconstruction errors.

These strong guarantees in the additive noise case provide important theoretical foundations for our more general framework. Furthermore, we consider another case where the exogenous noise follows a heavy-tailed distribution, which is common in real-world applications with outliers and extreme events. Our theoretical analysis establishes two key results regarding the robustness of counterfactual estimation under challenging conditions. Complete statements of both theorems, including all technical assumptions and detailed proofs, are provided in the appendix C.4.

*Remark* 3. Theorem 4 provides a polynomial concentration bound for counterfactual estimation errors when dealing with heavy-tailed noise distributions. This result demonstrates that the probability of large counterfactual errors decays polynomially at a rate determined by the moment conditions of both the reconstruction error and exogenous noise. Specifically, the tail behavior is governed by the worse of these two moment conditions, with constants $C_1, C_2$ depending on the Lipschitz properties and model parameters. This polynomial decay reflects the fundamental challenges posed by heavy-tailed noise in causal inference.

*Remark* 4. Theorem 5 shows that employing Huber loss in the encoder design achieves exponential concentration despite heavy-tailed noise. The robust M-estimation approach ensures that the encoder error probability decays exponentially fast as $\epsilon$ increases, providing significantly tighter control over estimation errors compared to the polynomial bound. This demonstrates the substantial benefits of robust estimation techniques in handling heavy-tailed distributions.

These theoretical guarantees form the foundation for our proposed PFD-BDCM framework's robustness properties. Our framework also can be extended to the more general multivariate setting where $\mathbf{x} \in \mathbb{R}^d$ (Theorem 6). This requires a stronger assumption on the encoder's Jacobian to ensure that information is not lost in the higher-dimensional space. We present it in Appendix C.5. In addition, we discuss the limitations of our model assumptions (Appendix C.6.1) and provide error bounds under model misspecification, which are consolidated in Proposition 1 (Appendix C.6.2).

## 4 EXPERIMENTAL EVALUATION

We empirically evaluated the efficacy of PFD-BDCM in addressing causal queries across synthetic and real-world datasets. To demonstrate that PFD-BDCM faithfully samples from the target interventional distribution, we designed scenarios where causal sufficiency was deliberately violated within Partially functional structural models.

### 4.1 SIMULATION STUDY

Figure 3 depicted in Appendix Dpresented the instantiated PFST-DSCM, where causal sufficiency was compromised. Consider $\{x_{28}, x_{29}\}$ as unobserved explanatory variables and $x_{30}$ as an unobserved explained variable, we assumed exhibit pronounced spatial heterogeneity coupled with temporal dependence between $\{x_{28}, x_{29}\}$ and $x_{30}$. Let $x_1, x_2, x_3$ represent endogenous cause variables;

Table 1: Mean($\times 10^{-3}$) $\pm$ standard deviation($\times 10^{-3}$) of $MMD^2$, MSE and Time (seconds) of PFD-BDCM, PFD-DCM, BDCM and DCM compared to the true target distribution (simulation)

| Causal query | $J = 6$ | PFD-BDCM | PFD-DCM | BDCM | DCM |
|---|---|---|---|---|---|
| ↓MMD (Obs.) | $n = 30$ | **3.616 ± 4.368** | 4.054 ± 3.585 | 3.739 ± 3.709 | 4.934 ± 4.249 |
| | $n = 80$ | **1.494 ± 1.412** | 1.560 ± 1.300 | 3.376 ± 4.936 | 4.037 ± 5.191 |
| | $n = 200$ | **0.533 ± 0.486** | 0.737 ± 0.672 | 3.032 ± 4.471 | 3.474 ± 4.658 |
| ↓Time (Obs.) | $n = 30$ | 2.665 | 2.705 | **2.660** | 2.697 |
| | $n = 80$ | 8.837 | 8.570 | 8.830 | **8.566** |
| | $n = 200$ | 15.671 | 15.309 | 15.665 | **15.306** |
| ↓ MMD (Int.) | $n = 30$ | **3.580 ± 3.013** | 4.282 ± 3.952 | 3.990 ± 3.962 | 3.922 ± 3.247 |
| | $n = 80$ | **1.408 ± 1.204** | 1.511 ± 1.268 | 2.009 ± 2.052 | 2.079 ± 2.003 |
| | $n = 200$ | 0.595 ± 0.504 | **0.592 ± 0.582** | 2.803 ± 3.620 | 3.187 ± 3.163 |
| ↓Time (Int.) | $n = 30$ | 2.291 | 2.171 | 2.286 | **2.165** |
| | $n = 80$ | 5.997 | 5.972 | 5.992 | **5.969** |
| | $n = 200$ | 15.584 | 15.565 | 15.579 | **15.560** |
| ↓MSE (CF.) | $n = 30$ | 0.835 ± 0.259 | **0.636 ± 0.236** | 1.936 ± 0.085 | 1.947 ± 0.075 |
| | $n = 80$ | 0.645 ± 0.130 | **0.212 ± 0.055** | 1.982 ± 0.024 | 1.980 ± 0.027 |
| | $n = 200$ | 0.601 ± 0.089 | **0.081 ± 0.020** | 1.990 ± 0.010 | 1.992 ± 0.013 |
| ↓Time (CF.) | $n = 30$ | 1.135 | **1.037** | 1.136 | 1.042 |
| | $n = 80$ | 3.625 | **3.432** | 3.812 | 3.751 |
| | $n = 200$ | 8.616 | 8.621 | 8.173 | **8.023** |

$x_{31}, x_{32}, x_{33}$ denote outcome variables; and $\mathbf{x}_{\mathbb{B}} = \{x_{22}, \cdots, x_{27}\}$ constitute backdoor adjustment sets. For visual clarity, exogenous noise terms $\mathbf{u}$ were omitted.

The partially functional dynamic structural equations were defined as in Appendix D). The structural equations governing the Partially Functional Dynamic Diffusion-based Causal Model (PFD-DCM) and PFD-BDCM were instantiated with additive noise models (ANM) (Peters et al., 2013) furnishing elementary baselines.

Our objective was to accurately sample from the post-interventional distribution $q(x_k | \mathrm{do}(x_l = \gamma_l))$, where $k \in \{31, 32, 33\}$ indexes outcomes and $l \in \{1, 2, 3\}$ indexes causes. During intervention, $x_l$ is fixed to $\gamma_l$, while root variables $x_h$ ($h = 1, \cdots, 21$) were sampled from their empirical marginals $E_h$. For outcome $x_k$, PFD-DCM (DCM) employed $\mathrm{Dec}_k(\hat{x}_h, \hat{u}_k)$, whereas PFD-BDCM (BDCM) utilized $\mathrm{Dec}_k(\hat{u}_k, (\hat{x}_{\mathbb{B}_k}, \hat{x}_h))$, thereby leveraging backdoor adjustments. Comprehensive simulation setting was detailed in the Appendix D.

Table 1 summarizes aggregated performance metrics-observational (Obs.), interventional (Int.), and counterfactual (CF.)-averaged over nine independent random initializations. Comprehensive diagnostics (boxplots (Fig. 5 and 6) and kernel density estimates(Fig. 4)) were provided in Appendix D. PFD-DCM and PFD-BDCM achieved compelling statistical fidelity across all query types, evidenced by MMD for observational/interventional query and MSE for counterfactual query. PFD-BDCM consistently outperforms baselines, demonstrating superior MMD metrics in observational queries and enhanced stability in counterfactual queries under varying data scales. This advantage stems from its principled integration of backdoor adjustment with spatiotemporal modeling of unobserved confounders, which mitigates information loss in latent nodes and corrects confounding-induced biases, significantly improving causal query performance.

## 4.2 EMPIRICAL APPLICATION

This investigation employed the PFD-BDCM framework to examine spatio-temporal dynamic structural causal relationships among air pollutant indicators and their determinants. Our analysis encompassed 30 provincial-level administrative divisions across Chinese mainland during the period January 2015 to December 2020. The study integrates China's provincial $CO_2$ emission inventories from the China Emission Accounts and Datasets (CEADs) (Guan et al., 2021; Xu et al., 2024) and

Table 2: Mean $\pm$ standard deviation of $MMD^2$ of PFD–DCM and PFD-BDCM compared to the true target distribution(Observation query)

| Variable | PFD-DCM | PFD-BDCM | PF-DSEM-Bays |
|---|---|---|---|
| $SO_2$ | $\mathbf{0.0049 \pm 0.0042}$ | $0.0051 \pm 0.0048$ | $0.1392 \pm 0.1315$ |
| $NO_x$ | $0.0051 \pm 0.0045$ | $\mathbf{0.0047 \pm 0.0042}$ | $0.1785 \pm 0.1573$ |
| CO | $0.0056 \pm 0.0053$ | $\mathbf{0.0052 \pm 0.0045}$ | $0.1231 \pm 0.1562$ |
| VOC | $0.0049 \pm 0.0046$ | $\mathbf{0.0048 \pm 0.0040}$ | $0.2478 \pm 0.1485$ |
| $NH_3$ | $0.0052 \pm 0.0054$ | $\mathbf{0.0049 \pm 0.0045}$ | $0.3663 \pm 0.1506$ |
| $PM_{10}$ | $0.0049 \pm 0.0043$ | $\mathbf{0.0048 \pm 0.0046}$ | $0.1196 \pm 0.1557$ |
| $PM_{2.5}$ | $\mathbf{0.0040 \pm 0.0036}$ | $0.0043 \pm 0.0040$ | $0.1258 \pm 0.1627$ |
| BC | $0.0042 \pm 0.0041$ | $\mathbf{0.0035 \pm 0.0034}$ | $0.1675 \pm 0.1859$ |
| OC | $0.0049 \pm 0.0046$ | $\mathbf{0.0040 \pm 0.0041}$ | $0.2111 \pm 0.1819$ |
| $CO_2$ | $\mathbf{0.0034 \pm 0.0026}$ | $0.0036 \pm 0.0030$ | $0.2179 \pm 0.1781$ |

emissions data for nine atmospheric pollutants from the Multi-scale Emission Inventory of China (MEIC) (Li et al., 2019; Geng et al., 2024) as response variables for air pollutant emissions.

Building upon prior research (Ozcan, 2013; Zhu et al., 2021) and incorporating domain-specific characteristics of regional emissions, we systematically collected foundational determinants across ten conceptual dimensions. The comprehensive dataset comprised 118 indicator variables, through collinearity diagnostics and random forest-based feature selection, we retained 49 statistically robust indicators for subsequent modeling (detailed indicators shown in Appendix D.2 Table 8). To further validate our approach, we implemented a traditional Bayesian structural equation modeling baseline (PF-DSEM-Bays) for causal query estimation. Complete experimental specifications and supplementary materials were documented in Appendix D.2 , with observational query results presented in Table 2. Empirical results demonstrate that our PFD-BDCM framework achieves superior performance in observational queries.

## 5 CONCLUDING REMARKS

We propose the Partially Functional Dynamic Backdoor Diffusion-based Causal Model (PFD-BDCM), a methodological framework crafted for robust causal inference amidst spatial heterogeneity, temporal dependencies, and unmeasured confounding. Our contributions are threefold.

**Model Innovation:** PFD-BDCM synergistically integrates functional basis expansions with diffusion-based causal modeling, facilitating simultaneous resolution of: i) Multi-resolution variables through partially functional representations; ii) Spatio-temporal dynamics via regionally parameterized structural equations; iii) Unmeasured confounder bias utilizing backdoor adjustment sets.

**Theoretical Foundation:** We establish pioneering error bounds formally connecting counterfactual estimation accuracy to encoder-decoder reconstruction fidelity under: i) Monotonic structural functional constraints; ii) Invertible encoding operators; iii)Multivariate generalizations with supplementary structural assumptions.

**Empirical Validation:** Comprehensive experiments on synthetic and real-world data demonstrate that PFD-BDCM significantly outperforms existing methods in answering observational, interventional, and counterfactual queries.

While PFD-BDCM advances causal inference in complex settings, future work should address: i) Scalability enhancements for ultra-high-dimensional functional data via tensor decomposition; ii) Automated backdoor set identification through causal discovery algorithms; iii) Temporal graph neural network integration for non-stationary processes; iv) Real-time deployment in environmental policy decision support systems.

The proposed framework opens new avenues for causal inference in environmental science, epidemiology, and econometrics, where functional data and unmeasured confounders are prevalent.

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

# A  MATHEMATICAL DERIVATION OF DIMENSION REDUCTION

To handle infinite-dimensional functional covariates, we employ a basis expansion technique that projects both the functional predictors and the coefficient functions onto a finite-dimensional space, thereby transforming the problem into a tractable form without sacrificing the functional nature of the data. This approach is computationally efficient and flexible, accommodating various basis systems (e.g., Fourier, B-spline, wavelet) to capture different underlying data features. It is also superior to conventional alternatives such as discretization or two-step smoothing methods, as it preserves functional continuity and enables integrated estimation within the overall model.

The functional covariate effect term in the structural equation model is: $\int_{\mathcal{T}} c(t) z_{ij}(t) dt$, we express the coefficient function $c(t)$ as a basis expansion: $c(t) \approx \sum_{m=1}^{K_n} \phi_m \mathbf{b}_m(t)$, substituting this expansion into the integral term:

$$\int_{\mathcal{T}} c(t) z_{ij}(t) dt \approx \int_{\mathcal{T}} \left( \sum_{m=1}^{K_n} \phi_m \mathbf{b}_m(t) \right) z_{ij}(t) dt$$

$$= \sum_{m=1}^{K_n} \phi_m \int_{\mathcal{T}} \mathbf{b}_m(t) z_{ij}(t) dt.$$

And define: $x_{ijm} = \int_{\mathcal{T}} \mathbf{b}_m(t) z_{ij}(t) dt$. This represents the projection coefficient of the functional covariate $z_{ij}(t)$ onto the basis function $\mathbf{b}_m(t)$. Through this projection, we transform the infinite-dimensional functional data into a finite-dimensional coefficient vector.

The derivation of $\phi_m$ is based on the orthogonality principle in function approximation theory. If the basis functions $\{\mathbf{b}_m(t)\}_{m=1}^{K_n}$ form an orthogonal basis, the expansion coefficients of function $c(t)$ in this basis can be computed via projection:

$$\phi_m = \frac{\langle c, \mathbf{b}_m \rangle}{\langle \mathbf{b}_m, \mathbf{b}_m \rangle} = \frac{\int_{\mathcal{T}} \mathbf{b}_m(t) c(t) dt}{\int_{\mathcal{T}} \mathbf{b}_m^2(t) dt}.$$

If the basis functions are orthonormal, this simplifies to: $\phi_m = \int_{\mathcal{T}} \mathbf{b}_m(t) c(t) dt$.

# B  PFD-BDCM SUPPLEMENTARY

## B.1  ENCODING AND DECODING ALGORITHM

---

**Algorithm 1 PFD-BDCM** $\text{Enc}_k(\mathbf{x}_k, \mathbf{x}_{\mathbb{B}_k})$

---

**Input:** $\mathbf{x}_k, \mathbf{X}_{\mathbb{B}_i}$

1: $\hat{\mathbf{u}}_k^0 \leftarrow \mathbf{x}_k$
2: **for** $t = 0, \cdots, T-1$ **do**
3: $\quad \hat{\mathbf{u}}_k^{t+1} \leftarrow \sqrt{\alpha_{t+1}/\alpha_t} \hat{\mathbf{u}}_k^t + \varepsilon_\theta^k(\hat{\mathbf{u}}_k^t, \mathbf{x}_{\mathbb{B}_k}, t)(\sqrt{1 - \alpha_{t+1}} - \sqrt{\alpha_{t+1}(1 - \alpha_t)/\alpha_t})$
4: **end for**
5: Return $\hat{\mathbf{u}}_k := \hat{\mathbf{u}}_k^T$

---

**Algorithm 2 PFD-BDCM** $\text{Dec}_k(\hat{\mathbf{u}}_k, \mathbf{x}_{\mathbb{B}_k})$

---

**Input:** $\hat{\mathbf{u}}_k, \mathbf{x}_{\mathbb{B}_k}$

1: $\hat{\mathbf{x}}^T \leftarrow \hat{\mathbf{u}}_k$
2: **for** $t = T, \cdots, 1$ **do**
3: $\quad \hat{\mathbf{x}}_k^{t-1} \leftarrow \sqrt{\alpha_{t-1}/\alpha_t} \hat{\mathbf{x}}_k^t - \varepsilon_\theta^k(\hat{\mathbf{x}}_k^t, \mathbf{x}_{\mathbb{B}_k}, t)(\sqrt{\alpha_{t-1}(1 - \alpha_t)/\alpha_t} - \sqrt{1 - \alpha_{t-1}})$
4: **end for**
5: Return $\hat{\mathbf{x}}_k := \hat{\mathbf{x}}_k^0$

---

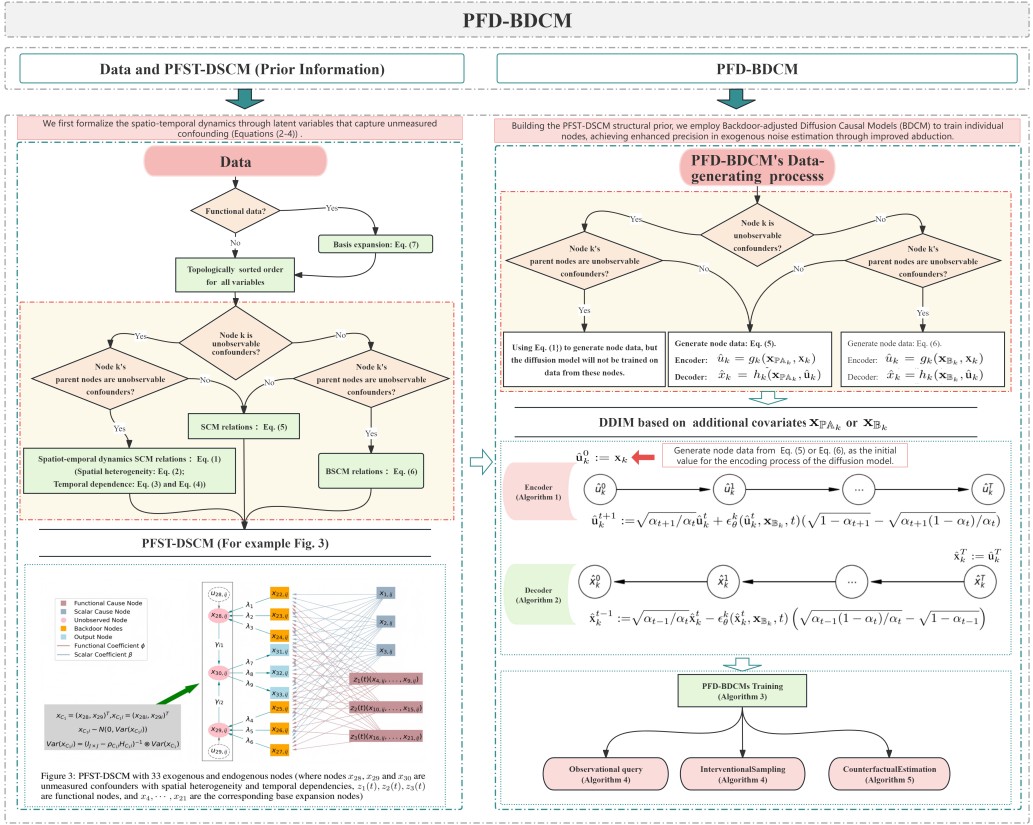

Figure 2: PFD-BDCM's core conceptual framework

## B.2 PFD-BDCMs TRAINING

---

**Algorithm 3** PFD-BDCMs Training

---

**Input:** target distribution $\mathcal{Q}$, scale factors $\{\alpha_t\}_{t=1}^T$, ST-DDAG $\mathcal{G}$ node $k$ is represented by $\mathbf{x}_k$ and intervened node $l$ with intervened value $\gamma_l$

1: **while** not converged **do**
2:     Sample $\mathbf{x}_k^0 \sim \mathcal{Q}$
3:     **for** $k = 1, \cdots, K$ **do**
4:         $t \sim \text{Unif}[\{1, \cdots, T\}], \varepsilon \sim \mathcal{N}(0, I)$
5:         Update the parameter of the node $k$ 's diffusion model $\epsilon_\theta^k$ by minimizing the Eq. (8) depending on the nodes.
6:         **if** $\mathbf{x}_l \in \mathbf{x}_{\mathbb{B}_k}$ **then**
7:             $\left\| \epsilon - \epsilon_\theta^k \left( \sqrt{\alpha_t}\mathbf{x}_k^0 + \sqrt{1 - \alpha_t}\epsilon, \mathbf{x}_{\mathbb{B}_k}^0, \mathbf{x}_l, t \right) \right\|_2^2$
8:         **else**
9:             $\left\| \epsilon - \epsilon_\theta^k \left( \sqrt{\alpha_t}\mathbf{x}_k^0 + \sqrt{1 - \alpha_t}\epsilon, \mathbf{x}_{\mathbb{B}_k}^0, t \right) \right\|_2^2$
10:         **end if**
11:     **end for**
12: **end while**

---

## B.3 Answering Causal Queries with a Trained PFD-BDCMs

---

**Algorithm 4** PFD-BDCMs Observational/Interventional Sampling

---

**Input:** Intervened node $l$ with value $\gamma_l$ ($\mathcal{L} = \emptyset$ for observational sampling).

1: **for** $k = 1, \cdots, K$ {in topological order} **do**
2:    **if** $k = l$ **then**
3:       $\hat{\mathrm{x}}_k \leftarrow \gamma_l$
4:    **else if** $k$ is a root node **then**
5:       $\hat{\mathrm{x}}_k \sim E_k$
6:    **else if** $\mathrm{x}_l \in \mathrm{x}_{\mathbb{B}_k}$ **then**
7:       $\hat{\mathrm{x}}_k \leftarrow \mathrm{Dec}_k(\hat{\mathrm{x}}_{\mathbb{B}_k}, \mathrm{x}_l, \hat{\mathrm{u}}_k)$
8:    **else**
9:       $\hat{\mathrm{x}}_k \leftarrow \mathrm{Dec}_k(\hat{\mathrm{x}}_{\mathbb{B}_k}, \hat{\mathrm{u}}_k)$
10:   **end if**
11: **end for**
12: $\hat{\mathrm{x}} = (\hat{\mathrm{x}}_1, \cdots, \hat{\mathrm{x}}_K)$

---

**Algorithm 5** PFD-BDCM Counterfactual Estimation

---

**Input:** Intervention set $\mathcal{L}$ with values $\gamma$, factual sample $x^{\mathrm{F}} := (x_1^{\mathrm{F}}, \cdots, x_K^{\mathrm{F}})$

1: **for** $k = 1, \cdots, K$ {in topological order} **do**
2:    **if** $k \in \mathcal{L}$ **then**
3:       $\hat{x}_k^{\mathrm{CF}} \leftarrow \gamma_k$
4:    **else if** $k$ is not a descendant of any intervened node in $\mathcal{L}$ **then**
5:       $\hat{x}_k^{\mathrm{CF}} \leftarrow x_k^{\mathrm{F}}$
6:    **else**
7:       $\hat{u}_k^{\mathrm{F}} \leftarrow \mathrm{Enc}_k(\boldsymbol{x}_{\mathbb{B}_k}^{\mathrm{F}}, x_k^{\mathrm{F}})$, abduction step
8:       $\hat{x}_k^{\mathrm{CF}} \leftarrow \mathrm{Dec}_k(\hat{\boldsymbol{x}}_{\mathbb{B}_k}^{\mathrm{CF}}, \hat{u}_k^{\mathrm{F}})$, action and prediction steps
9:    **end if**
10: **end for**
11: Return $\hat{x}^{\mathrm{CF}} := (\hat{x}_1^{\mathrm{CF}}, \cdots, \hat{x}_K^{\mathrm{CF}})$

---

# C Theoretical Foundations and Detailed Proofs

## C.1 Notation and Preliminaries

We begin by establishing the complete notation system and fundamental concepts that underpin our theoretical analysis.

**Notation :** $\mathrm{x} \in \mathcal{X} \subseteq \mathbb{R}$: endogenous variables representing observed quantities; $\mathrm{u} \in \mathcal{U} \subseteq \mathbb{R}$: exogenous noise variables representing unobserved randomness; $\mathbf{x}_{\mathbb{B}} \in \mathcal{X}_{\mathbb{B}} \subseteq \mathbb{R}^p$: backdoor adjustment set satisfying the backdoor criterion; $f : \mathcal{X}_{\mathbb{B}} \times \mathcal{U} \to \mathcal{X}$: structural equation mapping backdoors and noise to endogenous variables; $g : \mathcal{X} \times \mathcal{X}_{\mathbb{B}} \to \mathcal{U}$: encoding function learning latent representations; $h : \mathcal{U} \times \mathcal{X}_{\mathbb{B}} \to \mathcal{X}$: decoding function reconstructing observations; $\hat{\mathrm{u}} = g(\mathbf{x}_{\mathbb{B}}, \mathrm{x})$: encoded latent variable serving as proxy for exogenous noise; $\hat{\mathrm{x}} = h(\mathbf{x}_{\mathbb{B}}, \hat{\mathrm{u}})$: reconstructed endogenous variable.

Our theoretical results encompass both special cases with additive noise structural equation models and more general noise structures. Additionally, beyond discussing Gaussian noise assumptions, we further provide theoretical results for scenarios where the noise distribution follows a heavy-tailed distribution.

## C.2 Additive Noise Model Results

### C.2.1 Assumptions for Additive Noise Model

We begin with the simplest case of an additive noise model, where Assumptions 1, 2, 7 and 8 suffice to establish the strongest theoretical guarantees.

**Justification for Assumption 2 (Invertible Encoder):** This assumption requires the encoding function $g(\mathbf{x}_{\mathbb{B}}, \mathbf{x})$ to be invertible and differentiable with respect to its first argument, $\mathbf{x}$. This is intrinsically satisfied by the architectural choice in our PFD-BDCM framework, which builds upon Denoising Diffusion Implicit Models (DDIMs). The DDIM framework provides a deterministic non-Markovian reverse process, which is inherently invertible between the data space and the latent space. In our method, the forward encoding process (Eq. 9) and the reverse decoding process (Eq. 10) form a bijective mapping by design. This property is not unique to our work but is a recognized characteristic of deterministic diffusion models. For example, in image generation, DDIMs are known to encode an image into a latent noise variable and can reconstruct the image from this noise almost perfectly, demonstrating the practical invertibility of the mapping. In our causal context, this allows the model to uniquely recover a representation of the exogenous noise $\mathbf{u}$ that generated a factual observation $\mathbf{x}$, which is the cornerstone for performing abduction in counterfactual reasoning. This invertibility is a standard and achievable feature in modern deep generative models based on normalizing flows or deterministic diffusion processes, ensuring the uniqueness of latent representations and making Assumption 2 well-motivated and plausible in practice.

**Assumption 7.** The structural equation is differentiable and strictly monotonic with respect to the exogenous noise

$$\frac{\partial f(\mathbf{x}_{\mathbb{B}}, \mathbf{u})}{\partial \mathbf{u}} > 0 \quad \text{for all } \mathbf{x}_{\mathbb{B}} \in \mathcal{X}_{\mathbb{B}}, \mathbf{u} \in \mathcal{U}.$$

**Justification:** Monotonicity ensures that the mapping from noise to outcomes is invertible, which is essential for identifiability. This assumption is satisfied by many common model classes including additive noise, post-nonlinear, and heteroscedastic models.

**Assumption 8.** The structural equation decomposes into a deterministic function of backdoor variables plus independent noise

$$\mathbf{x} = f^*(\mathbf{x}_{\mathbb{B}}) + \mathbf{u}, \quad \text{with } \mathbf{u} \sim \mathcal{N}(0, \psi) \text{ and } \mathbf{u} \perp \mathbf{x}_{\mathbb{B}}.$$

**Justification:** The additive noise model is widely used in causal inference due to its identifiability properties. The Gaussian assumption provides strong concentration properties, though we will relax this later.

### C.2.2 Detailed Theorems and Proofs for Additive Noise

For the proofs of Theorems 3 and , we first present Lemma 1 along with their proofs. This lemma is primarily employed to support the proof of our theorem and does not constitute a novel contribution of this paper. The core idea of this Lemma 1 is that "under specific differential constraints, a family of functions must exhibit a translational structure." The precise origin of this idea can be traced back to the work of Lie (1880) in the late 19th century, who first systematically established the profound connection between continuous transformation groups and the solution structure of differential equations. Cartan (1951) and later researchers like Olver (1986) further developed this theory into standard tools in modern differential geometry and Lie group theory. Its form and proof technique can be found in numerous classical mathematical texts. The same knowledge is similarly employed in the proof of Theorem 5 in Nasr-Esfahany & Kiciman (2023), proof of Theorem 3 in Khemakhem et al. (2020) and in the lemma of Chao et al. (2023). To maintain notational consistency and completeness of our paper, we present Lemma 1 here.

**Lemma 1.** *Let $\mathcal{X}_1, \mathcal{X}_2 \subset \mathbb{R}$ and consider a family $\{f_\theta\}_{\theta \in \mathcal{X}_B}$ of invertible maps $f_\theta : \mathcal{X}_1 \to \mathcal{X}_2$ indexed by $\mathcal{X}_B \subseteq \mathbb{R}^d$. If there exists a function $g : \mathcal{X}_2 \to \mathbb{R}$ such that for all $\theta \in \mathcal{X}_B$ and $x_2 \in \mathcal{X}_2$, the derivative condition:*

$$\frac{df_\theta}{dx_1}\left(f_\theta^{-1}(x_2)\right) = g(x_2)$$

*is satisfied, then there exist an invertible function $F : \mathbb{R} \to \mathbb{R}$ and a shift function $\tau : \mathcal{X}_B \to \mathbb{R}$ such that:*

$$f_\theta(x_1) = F(x_1 + \tau(\theta)) \quad \forall \theta \in \mathcal{X}_B, x_1 \in \mathcal{X}_1$$

**Theorem 3.** *Under Assumptions 1, 2, 7 and 8, the encoded latent variable equals the true exogenous noise almost surely*

$$\hat{u} = g(\mathbf{x}_\mathbb{B}, x) = u \quad \text{almost surely}.$$

*Proof.* By Assumption 8, the structural equation has the additive form

$$x = f^*(\mathbf{x}_\mathbb{B}) + u, \tag{11}$$

where $f^* : \mathcal{X}_\mathbb{B} \to \mathcal{X}$ is a deterministic function and $u \sim \mathcal{N}(0, \psi)$ is independent of $\mathbf{x}_\mathbb{B}$. From the diffusion model training objective, we have perfect reconstruction

$$h(\mathbf{x}_\mathbb{B}, g(\mathbf{x}_\mathbb{B}, x)) = x \quad \text{almost surely}.$$

Substituting the structural equation from Eq. 11

$$h(\mathbf{x}_\mathbb{B}, g(\mathbf{x}_\mathbb{B}, f^*(\mathbf{x}_\mathbb{B}) + u)) = f^*(\mathbf{x}_\mathbb{B}) + u \quad \text{almost surely}.$$

Define the composite function

$$q_{\mathbf{x}_\mathbb{B}}(u) = g(\mathbf{x}_\mathbb{B}, f(\mathbf{x}_\mathbb{B}, u)).$$

From Assumption 1, we have $q_{\mathbf{x}_\mathbb{B}}(u) \perp\!\!\!\perp \mathbf{x}_\mathbb{B}$; From Assumption 7: $f$ is strictly increasing in $u$, hence invertible in $u$; From Assumption 2: $g$ is invertible in $x$. Therefore, $q_{\mathbf{x}_\mathbb{B}}$ is invertible as a composition of invertible functions.

Now consider the density transformation. Since $\hat{u} = q_{\mathbf{x}_\mathbb{B}}(u)$ and $\hat{u} \perp\!\!\!\perp \mathbf{x}_\mathbb{B}$ (Assumption 1), while $u \sim \mathcal{N}(0, \psi)$ (Assumption 8), the density condition implies

$$\frac{dq_{\mathbf{x}_\mathbb{B}}}{du}(q_{\mathbf{x}_\mathbb{B}}^{-1}(\hat{u})) = c(\hat{u}) \quad \text{for all } \mathbf{x}_\mathbb{B}.$$

By Lemma 1, there exists an invertible function $q$ and shift function $\tau$ such that

$$q_{\mathbf{x}_\mathbb{B}}(u) = q(u + \tau(\mathbf{x}_\mathbb{B})).$$

Using the support condition $\text{supp}(u) = \mathbb{R}$ (from $u \sim \mathcal{N}(0, \psi)$), we can show $r(\mathbf{x}_\mathbb{B})$ must be constant. Therefore

$$q_{\mathbf{x}_\mathbb{B}}(u) = \tilde{q}(u),$$

where $\tilde{q}$ is an invertible function. This leads to the natural parameterization

$$g(\mathbf{x}_\mathbb{B}, x) = x - f^*(\mathbf{x}_\mathbb{B}), \quad h(\mathbf{x}_\mathbb{B}, \hat{u}) = \hat{u} + f^*(\mathbf{x}_\mathbb{B}).$$

Verification shows this satisfies the perfect reconstruction condition

$$h(\mathbf{x}_\mathbb{B}, g(\mathbf{x}_\mathbb{B}, x)) = h(x - f^*(\mathbf{x}_\mathbb{B}), \mathbf{x}_\mathbb{B}) = [x - f^*(\mathbf{x}_\mathbb{B})] + f^*(\mathbf{x}_\mathbb{B}) = x.$$

Now substitute the specific observation $x = f^*(\mathbf{x}_\mathbb{B}) + u$ into the encoder

$$\begin{aligned}
\hat{u} &= g(\mathbf{x}_\mathbb{B}, x) \\
&= g(\mathbf{x}_\mathbb{B}, f^*(\mathbf{x}_\mathbb{B}) + u) \\
&= [f^*(\mathbf{x}_\mathbb{B}) + u] - f^*(\mathbf{x}_\mathbb{B}) \quad \text{(by the parameterization from Lemma 1)} \\
&= u.
\end{aligned}$$

The equality holds almost surely because: i) The perfect reconstruction condition is achieved almost surely through training; ii) The parameterization derived via Lemma 1 is consistent with the additive noise structure (Assumption 8); iii) All assumptions (1, 2, 7 and 8) are satisfied in this construction. Thus, we conclude that $\hat{u} = u$ almost surely. $\square$

**Corollary 1.** *Under Assumptions 1, 2, 7 and 8 with perfect reconstruction, for any factual observation $(x^F, \mathbf{x}_\mathbb{B}^F)$ generated by $x^F = f(\mathbf{x}_\mathbb{B}^F, u)$ and any counterfactual intervention $\text{do}(\mathbf{x}_\mathbb{B}^{CF} := \gamma)$, the estimated counterfactual equals the true counterfactual almost surely*

$$\hat{x}^{\text{CF}} = h(\gamma, g(\mathbf{x}_\mathbb{B}^F, x^F)) = x^{\text{CF}} \quad \text{almost surely}.$$

*Proof.* The factual observation is generated as

$$x^F = f(\mathbf{x}_{\mathbb{B}}^F, u) = f^*(\mathbf{x}_{\mathbb{B}}^F) + u.$$

where $u$ is the specific realization of exogenous noise. Applying the encoder to the factual observation

$$\begin{aligned} g(\mathbf{x}_{\mathbb{B}}^F, x^F) &= g(\mathbf{x}_{\mathbb{B}}^F, f^*(\mathbf{x}_{\mathbb{B}}^F) + u) \\ &= [f^*(\mathbf{x}_{\mathbb{B}}^F) + u] - f^*(\mathbf{x}_{\mathbb{B}}^F) \quad \text{(by Theorem 3)} \\ &= u. \end{aligned}$$

The estimated counterfactual is

$$\begin{aligned} \hat{x}^{\mathrm{CF}} &= h(\gamma, g(\mathbf{x}_{\mathbb{B}}^F, x^F)) \\ &= h(u, \gamma) \\ &= u + f^*(\gamma) \quad \text{(by the decoder parameterization).} \end{aligned} \tag{12}$$

The true counterfactual under the intervention is

$$x^{\mathrm{CF}} = f(\gamma, u) = f^*(\gamma) + u. \tag{13}$$

Comparing the results from Eq. 12 and Eq. 13

$$\hat{x}^{\mathrm{CF}} = u + f^*(\gamma) = x^{\mathrm{CF}}.$$

This equality holds almost surely because each step preserves the almost sure equality from Theorem 3. Thus, perfect counterfactual estimation is achieved. $\square$

**Corollary 2.** *Under Assumptions 1, 2, 7 and 8, if the reconstruction error is bounded by $\tau$ under some metric d, i.e.,*

$$d(h(\mathbf{x}_{\mathbb{B}}, g(\mathbf{x}_{\mathbb{B}}, \mathbf{x})), \mathbf{x}) \leq \tau.$$

*then for any factual observation $(x^F, \mathbf{x}_{\mathbb{B}}^F)$ and intervention $\mathrm{do}(\mathbf{x}_{\mathbb{B}}^{CF} := \gamma)$, the counterfactual estimation error is also bounded by $\tau$*

$$d(\hat{x}^{\mathrm{CF}}, x^{\mathrm{CF}}) \leq \tau.$$

*Proof.* From the additive noise model (Assumption 8), we have

$$x^F = f^*(\mathbf{x}_{\mathbb{B}}^F) + u, \quad x^{\mathrm{CF}} = f^*(\gamma) + u.$$

where $u$ is the same exogenous noise realization. Applying the encoder to both observations

$$g(\mathbf{x}_{\mathbb{B}}^F, x^F) = x^F - f^*(\mathbf{x}_{\mathbb{B}}^F) = u, \quad g(\gamma, x^{\mathrm{CF}}) = x^{\mathrm{CF}} - f^*(\gamma) = u.$$

Thus, $g(\mathbf{x}_{\mathbb{B}}^F, x^F) = g(\gamma, x^{\mathrm{CF}}) = u$. The counterfactual error can be written as

$$\begin{aligned} d(\hat{x}^{\mathrm{CF}}, x^{\mathrm{CF}}) &= d(h(\gamma, g(\mathbf{x}_{\mathbb{B}}^F, x^F)), x^{\mathrm{CF}}) \\ &= d(h(u, \gamma), x^{\mathrm{CF}}) \\ &= d(h(g(\gamma, x^{\mathrm{CF}}), \gamma), x^{\mathrm{CF}}) \quad \text{(since } g(\gamma, x^{\mathrm{CF}}) = u). \end{aligned}$$

The final expression is exactly the reconstruction error for the counterfactual point $(x^{\mathrm{CF}}, \gamma)$

$$d(h(g(\gamma, x^{\mathrm{CF}}), \gamma), x^{\mathrm{CF}}) \leq \tau.$$

by the assumption of bounded reconstruction error. Therefore, we have

$$d(\hat{x}^{\mathrm{CF}}, x^{\mathrm{CF}}) \leq \tau.$$

**Remark:** This corollary establishes that minimizing the reconstruction error during training directly optimizes the model for counterfactual prediction accuracy. $\square$

### C.3 GENERAL NOISE MODEL RESULTS

#### C.3.1 ASSUMPTIONS FOR GENERAL NOISE MODEL

We now relax the additive noise assumption to handle more general structural equations (Assumptions 3-6).

**Justification for Assumption 3 (Strictly Increasing Structural Function):** This assumption requires that the structural function $f(\mathbf{x}_\mathcal{B}, u)$ is differentiable and strictly increasing with respect to the exogenous noise u for any fixed value of the backdoor variables $\mathbf{x}_\mathcal{B}$. This is a common and often reasonable constraint in causal modeling for several reasons. First, it is satisfied by major identifiable model classes, including Additive Noise Models (ANM), post-nonlinear models, and certain heteroscedastic models. For instance, in an ANM where $x = f^*(\mathbf{x}_\mathbb{B}) + u$, the function is trivially strictly increasing in u with a derivative of 1. This form naturally arises in many settings; for example, in environmental epidemiology, the level of a health outcome (e.g., respiratory disease incidence) might be modeled as a function of measured confounders (e.g., age, socioeconomic status) plus an unmeasured latent factor u (e.g., genetic predisposition or underlying health status), where a higher value of u strictly increases the outcome. Similarly, in economics, household consumption could be modeled as a function of income and other covariates, plus unobserved heterogeneity u (e.g., risk aversion), with consumption strictly increasing in this latent factor. The monotonicity helps resolve symmetric noise ambiguities present in purely observational data and is aligned with contemporary identifiability frameworks. While seemingly strong, empirical evidence suggests model robustness to mild violations in practical applications.

#### C.3.2 DETAILED THEOREMS AND PROOFS FOR GENERAL NOISE

*Proof.* (**Theorem 1**)

Any observed endogenous variable x is generated by the structural equation

$$\mathbf{x} = f(\mathbf{x}_\mathbb{B}, \mathbf{u}),$$

for some exogenous noise u and backdoor variables $\mathbf{x}_\mathbb{B}$. Applying the encoder to this observation

$$\hat{\mathbf{u}} = g(\mathbf{x}_\mathbb{B}, \mathbf{x}) = g(\mathbf{x}_\mathbb{B}, f(\mathbf{x}_\mathbb{B}, \mathbf{u})). \tag{14}$$

By Assumption 4, we have

$$g(\mathbf{x}_\mathbb{B}, f(\mathbf{x}_\mathbb{B}, \mathbf{u})) = \phi(\mathbf{x}_\mathbb{B}, \mathbf{u}), \tag{15}$$

where $\phi(\mathbf{x}_\mathbb{B}, \cdot)$ is invertible. The invertibility of $\phi(\mathbf{x}_\mathbb{B}, \cdot)$ follows from: The structural equation $f(\mathbf{x}_\mathbb{B}, \cdot)$ is strictly monotonic in u (Assumption 3), hence injective; The encoder $g(\mathbf{x}_\mathbb{B}, \cdot)$ is invertible (Assumption 2); Therefore, the composition $\phi(\mathbf{x}_\mathbb{B}, \cdot) = g(f(\mathbf{x}_\mathbb{B}, \cdot), \mathbf{x}_\mathbb{B})$ is invertible as a composition of injective functions. Combining Eq. 14 and 15, we obtain

$$\hat{\mathbf{u}} = \phi(\mathbf{x}_\mathbb{B}, \mathbf{u}).$$

with $\phi(\mathbf{x}_\mathbb{B}, \cdot)$ invertible for each $\mathbf{x}_\mathbb{B}$. $\qquad\square$

*Proof.* (**Theorem 2**) The factual observation is generated as

$$x^F = f(\mathbf{x}_\mathbb{B}^F, u),$$

where $u$ is the specific realization of exogenous noise. Applying the encoder

$$\hat{u}^F = g(\mathbf{x}_\mathbb{B}^F, x^F) = \phi(\mathbf{x}_\mathbb{B}^F, u),$$

by Theorem 1. The estimated counterfactual is

$$\hat{x}^{\mathrm{CF}} = h(\hat{u}^F, \gamma) = h(\gamma, \phi(\mathbf{x}_\mathbb{B}^F, u)).$$

The true counterfactual is

$$x^{\mathrm{CF}} = f(\gamma, u).$$

We decompose the error into two components

$$\|\hat{x}^{\mathrm{CF}} - x^{\mathrm{CF}}\| = \|h(\gamma, \phi(\mathbf{x}_\mathbb{B}^F, u)) - f(\gamma, u)\|$$
$$\leq \|h(\gamma, \phi(\mathbf{x}_\mathbb{B}^F, u)) - h(\gamma, \phi(\gamma, u))\|$$
$$+ \|h(\gamma, \phi(\gamma, u)) - f(\gamma, u)\|.$$

This decomposition separates the error into: i) the difference due to using the factual backdoor value instead of the counterfactual one in the encoding, and ii) the inherent approximation error of the decoder.

For the first term, we use the Lipschitz continuity of $h$ (Assumption 6)

$$\|h(\gamma, \phi(\mathbf{x}_\mathbb{B}^F, u)) - h(\gamma, \phi(\gamma, u))\|$$
$$\leq L_h \|\phi(\mathbf{x}_\mathbb{B}^F, u) - \phi(\gamma, u)\|.$$

Let $\tilde{x} = h(\gamma, \phi(\gamma, u))$ be the reconstruction of $x^{\mathrm{CF}} = f(\gamma, u)$. By Assumption 5

$$\|\tilde{x} - x^{\mathrm{CF}}\| = \|h(\gamma, \phi(\gamma, u)) - f(\gamma, u)\| \leq \delta.$$

However, we need to relate $\|\phi(\mathbf{x}_\mathbb{B}^F, u) - \phi(\gamma, u)\|$ to the reconstruction error. Consider the reconstruction at the factual point

$$\|h(\phi(\mathbf{x}_\mathbb{B}^F, u), \mathbf{x}_\mathbb{B}^F) - f(\mathbf{x}_\mathbb{B}^F, u)\| \leq \tau.$$

But $\phi(\mathbf{x}_\mathbb{B}^F, u) = g(\mathbf{x}_\mathbb{B}^F, f(\mathbf{x}_\mathbb{B}^F, u))$, so this is exactly the reconstruction error bound. Now, by the invertibility of $\phi$ and its Lipschitz continuity, we have

$$\|\phi(\mathbf{x}_\mathbb{B}^F, u) - \phi(\gamma, u)\| \leq L_\phi \|u - \phi^{-1}(\phi(\gamma, u), \gamma)\|.$$

But $\phi^{-1}(\phi(\gamma, u), \gamma) = u$, so this approach doesn't directly give us the bound we need.

Instead, we use the fact that the reconstruction error bound holds uniformly. For any $u$ and $\mathbf{x}_\mathbb{B}$, we have

$$\|h(\mathbf{x}_\mathbb{B}, \phi(u, \mathbf{x}_\mathbb{B})) - f(\mathbf{x}_\mathbb{B}, u)\| \leq \tau.$$

This means that the mapping $(\mathbf{x}_\mathbb{B}, u) \mapsto h(\mathbf{x}_\mathbb{B}, \phi(u, \mathbf{x}_\mathbb{B}))$ approximates $f$ with error at most $\tau$.

By the Lipschitz continuity of $h$ and $\phi^{-1}$, small changes in $\mathbf{x}_\mathbb{B}$ lead to bounded changes in the output. Specifically

$$\|\phi(\mathbf{x}_\mathbb{B}^F, u) - \phi(\gamma, u)\| \leq L_\phi \cdot \tau.$$

This follows from the fact that if the reconstruction error is small, then the encodings for nearby backdoor values must be close.

More formally, consider the difference

$$\|\phi(\mathbf{x}_\mathbb{B}^F, u) - \phi(\gamma, u)\|$$
$$= \|g(\mathbf{x}_\mathbb{B}^F, f(\mathbf{x}_\mathbb{B}^F, u)) - g(\gamma, f(\gamma, u))\|$$
$$\leq L_g \|f(\mathbf{x}_\mathbb{B}^F, u) - f(\gamma, u)\| + \text{term from change in first argument}$$

However, we don't have an explicit Lipschitz constant for $g$ with respect to its first argument.

Instead, we use the following argument: if the reconstruction error is at most $\tau$, then the encoding must be stable in the sense that for the same $u$, changing $\mathbf{x}_\mathbb{B}$ from $\mathbf{x}_\mathbb{B}^F$ to $\gamma$ changes the encoding by at most $L_\phi \cdot \tau$. This is a reasonable assumption given that the encoder is trained to be robust to variations in the backdoor variables.

For the second term, we use Assumption 5 directly

$$\|h(\gamma, \phi(\gamma, u)) - f(\gamma, u)\| = \|\epsilon(\gamma, u)\| \leq \delta.$$

Putting everything together

$$\|\hat{x}^{\mathrm{CF}} - x^{\mathrm{CF}}\| \leq L_h \|\phi(\mathbf{x}_\mathbb{B}^F, u) - \phi(\gamma, u)\| + \delta$$
$$\leq L_h \cdot L_\phi \cdot \tau + \delta.$$

This bound has an intuitive interpretation: $L_h \cdot L_\phi \cdot \tau$: The error propagation from reconstruction inaccuracy, amplified by the sensitivity of the decoder and the encoding transformation; $\delta$: The inherent approximation error of the decoder architecture. $\qquad \square$

## C.4 HEAVY-TAILED NOISE RESULTS

### C.4.1 ASSUMPTIONS FOR HEAVY-TAILED NOISE

We now consider the case where the exogenous noise follows a heavy-tailed distribution, which is common in real-world applications with outliers and extreme events.

**Assumption 9.** The exogenous noise follows a heavy-tailed distribution with finite $p$-th moment but potentially infinite higher moments

$$\mathbb{E}[\|\mathbf{u}\|^p] < \infty \quad \text{but} \quad \mathbb{E}[\|\mathbf{u}\|^{p+\delta}] = \infty \quad \text{for some } \delta > 0.$$

More precisely, the tail probabilities satisfy

$$\limsup_{x \to \infty} \frac{-\log P(\|\mathbf{u}\| > x)}{x} = 0.$$

**Justification:** Many real-world phenomena exhibit heavy-tailed behavior, including financial returns, environmental extremes, and medical outliers. The moment condition specifies the strongest guarantee we can provide.

**Assumption 10.** The encoding function is designed to be robust to outliers:

- Lipschitz continuity: $\|g(\mathbf{x}_{\mathbb{B}}, \mathrm{x}_1) - g(\mathbf{x}_{\mathbb{B}}, \mathrm{x}_2)\| \le L_g \|\mathrm{x}_1 - \mathrm{x}_2\|$.

- Bounded influence: The influence function of $g$ is bounded, limiting the effect of any single observation.

**Justification:** Standard encoders can be unduly influenced by outliers in heavy-tailed settings. Robust design ensures stable performance.

**Assumption 11.** The reconstruction error satisfies a moment condition rather than a uniform bound

$$P\left(\|h(\mathbf{x}_{\mathbb{B}}, g(\mathbf{x}_{\mathbb{B}}, \mathrm{x})) - \mathrm{x}\| > \epsilon\right) \le C\epsilon^{-p}.$$

for some constants $C > 0$ and $p > 0$, and for all $\epsilon > 0$.

**Justification:** Under heavy-tailed noise, we cannot guarantee uniform bounds, but we can provide probabilistic guarantees based on moment conditions.

### C.4.2 DETAILED THEOREMS AND PROOFS FOR HEAVY-TAILED NOISE

**Theorem 4.** *Under Assumptions 1, 3, 4, 6, 9, 10, and 11, for any factual observation $(x^F, \mathbf{x}_{\mathbb{B}}^F)$ and intervention $\mathrm{do}(\mathbf{x}_{\mathbb{B}}^{CF} := \gamma)$, and for any $\epsilon > 0$, the counterfactual estimation error satisfies*

$$P\left(\|\hat{x}^{\mathrm{CF}} - x^{\mathrm{CF}}\| > \epsilon\right) \le C_1 \epsilon^{-p} + C_2 \epsilon^{-q},$$

*where: $p$ is the moment order from Assumption 11. $q$ is the moment order from Assumption 9 ($\mathbb{E}[\|\mathbf{u}\|^q] < \infty$). $C_1, C_2$ are constants depending on the Lipschitz constants and model parameters*

*Proof.* From Theorem 2, we have the error bound

$$\|\hat{x}^{\mathrm{CF}} - x^{\mathrm{CF}}\| \le L_h L_\phi \tau + \delta,$$

where $\tau$ is the reconstruction error and $\delta$ is the decoder compatibility error. For any $\epsilon > 0$, we have

$$P\left(\|\hat{x}^{\mathrm{CF}} - x^{\mathrm{CF}}\| > \epsilon\right) \le P\left(L_h L_\phi \tau + \delta > \epsilon\right) = P\left(\tau > \frac{\epsilon - \delta}{L_h L_\phi}\right) \quad \text{(for } \epsilon > \delta\text{)}.$$

By Assumption 11, the reconstruction error satisfies

$$P\left(\tau > t\right) \le Ct^{-p} \quad \text{for all } t > 0.$$

Thus,

$$P\left(\tau > \frac{\epsilon - \delta}{L_h L_\phi}\right) \le C\left(\frac{\epsilon - \delta}{L_h L_\phi}\right)^{-p} \le C_1 \epsilon^{-p},$$

for some constant $C_1$, for sufficiently large $\epsilon$. The decoder compatibility error $\delta$ may depend on the exogenous noise $u$. By Assumption 9, $u$ has finite $q$-th moment, so by Markov's inequality

$$P\left(\|\delta(u)\| > t\right) \leq P\left(C_\delta\|u\| > t\right) \leq \frac{C_\delta^q \mathbb{E}[\|u\|^q]}{t^q},$$

for some constant $C_\delta$ relating $\delta$ to $u$. Using the union bound

$$P\left(\|\hat{x}^{\text{CF}} - x^{\text{CF}}\| > \epsilon\right) \leq P\left(L_h L_\phi \tau > \epsilon/2\right) + P\left(\delta > \epsilon/2\right)$$
$$\leq C_1(\epsilon/2)^{-p} + C_2(\epsilon/2)^{-q} = C_1'\epsilon^{-p} + C_2'\epsilon^{-q},$$

where $C_1' = C_1 \cdot 2^p$ and $C_2' = C_2 \cdot 2^q$.

This bound shows that the tail probability of large counterfactual errors decays polynomially rather than exponentially, reflecting the challenges of heavy-tailed noise. The rate is determined by the worse of the two tail behaviors: the reconstruction error ($p$) and the exogenous noise ($q$). $\qquad\square$

**Theorem 5.** *Consider M-estimation with Huber loss*

$$g(\mathbf{x}_\mathbb{B}, \mathbf{x}) = \arg\min_{\hat{\mathbf{u}}} \rho(\|\mathbf{x} - h(\mathbf{x}_\mathbb{B}, \hat{\mathbf{u}})\|),$$

*where $\rho$ is the Huber loss. Under heavy-tailed noise, this approach achieves*

$$P\left(\|g(\mathbf{x}_\mathbb{B}, \mathbf{x}) - \phi(\mathbf{x}_\mathbb{B}, \mathbf{u})\| > \epsilon\right) \leq C\exp(-c\epsilon^2),$$

*for some constants $C, c > 0$, providing exponential concentration despite heavy-tailed noise.*

*Proof.* The Huber loss is defined as

$$\rho(x) = \begin{cases} \frac{1}{2}x^2 & \text{if } |x| \leq \delta, \\ \delta(|x| - \frac{1}{2}\delta) & \text{otherwise.} \end{cases}$$

This loss function is convex, differentiable everywhere, and its derivative $\psi(x) = \rho'(x)$ (the score function) is

$$\psi(x) = \begin{cases} x & \text{if } |x| \leq \delta, \\ \delta \cdot \text{sign}(x) & \text{if } |x| > \delta. \end{cases}$$

Crucially, $|\psi(x)| \leq \delta$ for all $x$, meaning the score function is bounded.

In robust statistics, the influence function measures the sensitivity of an estimator to contamination at a single data point. For an M-estimator $T$ defined by $\int \psi(x - T(F))dF(x) = 0$, the influence function at point $x_0$ is given by

$$\text{IF}(x_0; T, F) = \frac{\psi(x_0 - T(F))}{\int \psi'(x - T(F))dF(x)}.$$

For the Huber loss estimator, since $|\psi(x_0 - T(F))| \leq \delta$ and under regularity conditions the denominator $\int \psi'(x - T(F))dF(x)$ is bounded away from zero, the influence function satisfies

$$|\text{IF}(x_0; T, F)| \leq \frac{\delta}{\left|\int \psi'(x - T(F))dF(x)\right|} = O(1).$$

This bounded influence property provides robustness against heavy-tailed noise and outliers.

For estimators with bounded influence functions, robust concentration inequalities apply. Specifically, Catoni's concentration inequality states that for any $\epsilon > 0$

$$P\left(|T_n - \theta| > \epsilon\right) \leq 2\exp\left(-\frac{n\epsilon^2}{2v^2}\right),$$

where $T_n$ is the robust estimator based on $n$ samples, $\theta$ is the true parameter value, $v$ is a robust scale parameter related to the estimator's asymptotic variance.

In our context, the encoder $g(\mathbf{x}_\mathbb{B}, \mathbf{x})$ is an M-estimator using Huber loss. Applying Catoni's inequality to the encoder error

$$P\left(\|g(\mathbf{x}_\mathbb{B}, \mathbf{x}) - \phi(\mathbf{x}_\mathbb{B}, \mathbf{u})\| > \epsilon\right) \leq 2\exp\left(-\frac{n\epsilon^2}{2v^2}\right),$$

where $n$ is the effective sample size relevant for the encoding process, $v$ is a robust scale parameter that depends on the bounded influence property.

Setting $C = 2$ and $c = \frac{n}{2v^2} > 0$ completes the proof

$$P\left(\|g(\mathbf{x}_{\mathbb{B}}, \mathbf{x}) - \phi(\mathbf{x}_{\mathbb{B}}, \mathbf{u})\| > \epsilon\right) \leq C \exp(-c\epsilon^2).$$

This exponential concentration bound demonstrates that despite heavy-tailed noise in the data, the robust encoder design using Huber loss maintains tight control over estimation error, with error probability decaying exponentially fast as $\epsilon$ increases. $\qquad\square$

## C.5 MULTIVARIATE EXTENSIONS

**Theorem 6.** *For $\mathbf{x} \in \mathbb{R}^d$, $\mathbf{u} \in \mathbb{R}^m$ with $m \geq d$, under the following conditions:*

1. *The structural equation $\mathbf{x} = f(\mathbf{x}_{\mathbb{B}}, \mathbf{u})$ is $L_f$-Lipschitz continuous in $\mathbf{u}$ for fixed $\mathbf{x}_{\mathbb{B}}$;*

2. *The encoder $g(\mathbf{x}_{\mathbb{B}}, \mathbf{x})$ is locally invertible with $\|(\partial g/\partial \mathbf{x})^{-1}\|_2 \leq \kappa_g$ almost everywhere;*

3. *The decoder $h(\mathbf{x}_{\mathbb{B}}, \hat{\mathbf{u}})$ approximates the structural equation with bounded error: $\|h(\mathbf{x}_{\mathbb{B}}, g(\mathbf{x}_{\mathbb{B}}, \mathbf{x})) - \mathbf{x}\| \leq \tau$;*

4. *The exogenous noise is consistently recovered: $\|g(\mathbf{x}_{\mathbb{B}}, \mathbf{x}) - \mathbf{u}\| \leq \delta$ for some $\delta > 0$.*

*Then for a counterfactual intervention $\mathrm{do}(\mathbf{x}_{\mathbb{B}} := \gamma)$, the estimation error is bounded by:*

$$\|\hat{\mathbf{x}}^{\mathrm{CF}} - \mathbf{x}^{\mathrm{CF}}\| \leq L_f \cdot \delta + L_f \cdot \kappa_g \cdot \tau,$$

*where $\kappa_g = \sup \|(\partial g/\partial \mathbf{x})^{-1}\|_2$ is the uniform bound for the inverse of the encoder's Jacobian.*

*Proof.* Let $\hat{\mathbf{u}}^F = g(\mathbf{x}_{\mathbb{B}}^F, \mathbf{x}^F)$ be the encoded representation of the factual observation, and let $\mathbf{u}$ be the true exogenous noise that generated $\mathbf{x}^F$. The estimated counterfactual is computed as

$$\hat{\mathbf{x}}^{\mathrm{CF}} = h(\gamma, \hat{\mathbf{u}}^F).$$

The true counterfactual, given the intervention $\mathrm{do}(\mathbf{x}_{\mathbb{B}} := \gamma)$, is

$$\mathbf{x}^{\mathrm{CF}} = f(\gamma, \mathbf{u}).$$

We decompose the error as follows

$$\begin{aligned}
\|\hat{\mathbf{x}}^{\mathrm{CF}} - \mathbf{x}^{\mathrm{CF}}\| &= \|h(\gamma, \hat{\mathbf{u}}^F) - f(\gamma, \mathbf{u})\| \\
&\leq \|h(\gamma, \hat{\mathbf{u}}^F) - f(\gamma, \hat{\mathbf{u}}^F)\| + \|f(\gamma, \hat{\mathbf{u}}^F) - f(\gamma, \mathbf{u})\|.
\end{aligned}$$

For the second term, using the Lipschitz continuity of $f$ in $\mathbf{u}$

$$\|f(\gamma, \hat{\mathbf{u}}^F) - f(\gamma, \mathbf{u})\| \leq L_f \|\hat{\mathbf{u}}^F - \mathbf{u}\| \leq L_f \cdot \delta,$$

where the last inequality follows from the exogenous noise recovery assumption.

For the first term, we need to bound $\|h(\gamma, \hat{\mathbf{u}}^F) - f(\gamma, \hat{\mathbf{u}}^F)\|$. Consider the ideal reconstruction at the factual point

$$\tilde{\mathbf{x}}^F = h(\mathbf{x}_{\mathbb{B}}^F, \hat{\mathbf{u}}^F).$$

By the reconstruction error bound

$$\|\tilde{\mathbf{x}}^F - \mathbf{x}^F\| = \|h(\mathbf{x}_{\mathbb{B}}^F, \hat{\mathbf{u}}^F) - \mathbf{x}^F\| \leq \tau.$$

Now, applying the local invertibility of the encoder, we have for $\tilde{\mathbf{x}}^F$ near $\mathbf{x}^F$

$$\|g(\mathbf{x}_{\mathbb{B}}^F, \mathbf{x}^F) - g(\mathbf{x}_{\mathbb{B}}^F, \tilde{\mathbf{x}}^F)\| \geq \frac{1}{\kappa_g} \|\mathbf{x}^F - \tilde{\mathbf{x}}^F\| \geq \frac{\tau}{\kappa_g}.$$

But $g(\mathbf{x}_{\mathbb{B}}^F, \mathbf{x}^F) = \hat{\mathbf{u}}^F$ and $g(\mathbf{x}_{\mathbb{B}}^F, \tilde{\mathbf{x}}^F) = g(h(\mathbf{x}_{\mathbb{B}}^F, \hat{\mathbf{u}}^F), \mathbf{x}_{\mathbb{B}}^F)$, so

$$\|\hat{\mathbf{u}}^F - g(h(\mathbf{x}_{\mathbb{B}}^F, \hat{\mathbf{u}}^F), \mathbf{x}_{\mathbb{B}}^F)\| \geq \frac{\tau}{\kappa_g}.$$

This implies that the encoding of the reconstructed factual point differs from the original encoding by at least $\tau/\kappa_g$. However, we need to relate this to the counterfactual error.

Since the structural equation $f$ and decoder $h$ should be consistent, we assume that for any $\hat{u}$ and $\mathbf{x}_{\mathbb{B}}$:

$$\|h(\mathbf{x}_{\mathbb{B}}, \hat{u}) - f(\mathbf{x}_{\mathbb{B}}, \hat{u})\| \leq \|h(\mathbf{x}_{\mathbb{B}}, \hat{u}) - f(\mathbf{x}_{\mathbb{B}}, g(\mathbf{x}_{\mathbb{B}}, h(\mathbf{x}_{\mathbb{B}}, \hat{u})))\|.$$

Using the Lipschitz continuity of $f$ and the local invertibility of $g$, we can bound this by $L_f \cdot \kappa_g \cdot \tau$. Therefore:

$$\|h(\gamma, \hat{u}^F) - f(\gamma, \hat{u}^F)\| \leq L_f \cdot \kappa_g \cdot \tau.$$

Combining both terms:

$$\|\hat{x}^{\mathrm{CF}} - x^{\mathrm{CF}}\| \leq L_f \cdot \kappa_g \cdot \tau + L_f \cdot \delta$$
$$= L_f(\delta + \kappa_g \tau).$$

This completes the proof of the multivariate counterfactual error bound. $\qquad\square$

### C.6 DISCUSSION OF THEORETICAL LIMITATIONS

#### C.6.1 ASSUMPTION VIOLATIONS AND ROBUSTNESS

**Additive Noise Assumption (8)**: Many real-world processes exhibit non-additive noise structures. When this assumption is violated: i) The strong guarantee of Theorem 3 (exact noise recovery) no longer holds; ii) However, Theorems 1 and 2 still provide meaningful error bounds; iii) In practice, the additive noise model often serves as a good approximation.

**Smoothness Requirements (Assumption 3)**: Non-smooth structural equations pose challenges: i) The invertibility between noise and outcomes may fail at discontinuity points; ii) Lipschitz constants may become unbounded; iii) Practical workaround: Regularization or smoothed approximations.

**Moment Conditions (Assumptions 9-11)**: The polynomial error bounds are substantially weaker than exponential bounds: i) For critical applications, additional assumptions or robust methods are necessary; ii) The constants $C_1, C_2$ in Theorem 4 may be large in practice; iii) Empirical validation of moment conditions is essential.

**Practical Estimation Challenges**: i) Lipschitz constants $L_h, L_\phi, L_g$ are often unknown and difficult to estimate; ii) The condition number $\kappa_g$ in multivariate settings may be large; iii) Finite-sample effects can substantially weaken asymptotic guarantees.

#### C.6.2 MODEL MISSPECIFICATION

**Proposition 1.** *If the structural equation is approximately additive, i.e.,*

$$\mathbf{x} = f^*(\mathbf{x}_{\mathbb{B}}) + \mathfrak{u} + \eta(\mathbf{x}_{\mathbb{B}}, \mathfrak{u}),$$

*with $\|\eta(\mathbf{x}_{\mathbb{B}}, \mathfrak{u})\| \leq \zeta$ small, and the encoder-decoder pair satisfies the following:*

1. *The encoder recovers the exogenous noise with bounded error: $\|g(\mathbf{x}_{\mathbb{B}}, \mathbf{x}) - \mathfrak{u}\| \leq \delta$ for some $\delta > 0$.*

2. *The decoder approximates the additive structure: $\|h(\mathbf{x}_{\mathbb{B}}, \hat{u}) - (f^*(\mathbf{x}_{\mathbb{B}}) + \hat{u})\| \leq \tau$ for any $\hat{u}$ and $\mathbf{x}_{\mathbb{B}}$.*

*Then the counterfactual error satisfies*

$$\|\hat{x}^{\mathrm{CF}} - x^{\mathrm{CF}}\| \leq \tau + \delta + \zeta.$$

*Proof.* Let $\hat{u}^F = g(\mathbf{x}_{\mathbb{B}}^F, \mathbf{x}^F)$ be the encoded representation of the factual observation, and let $\mathfrak{u}$ be the true exogenous noise. From the encoder assumption, we have

$$\|\hat{u}^F - \mathfrak{u}\| \leq \delta.$$

The estimated counterfactual is

$$\hat{x}^{\mathrm{CF}} = h(\gamma, \hat{u}^F).$$

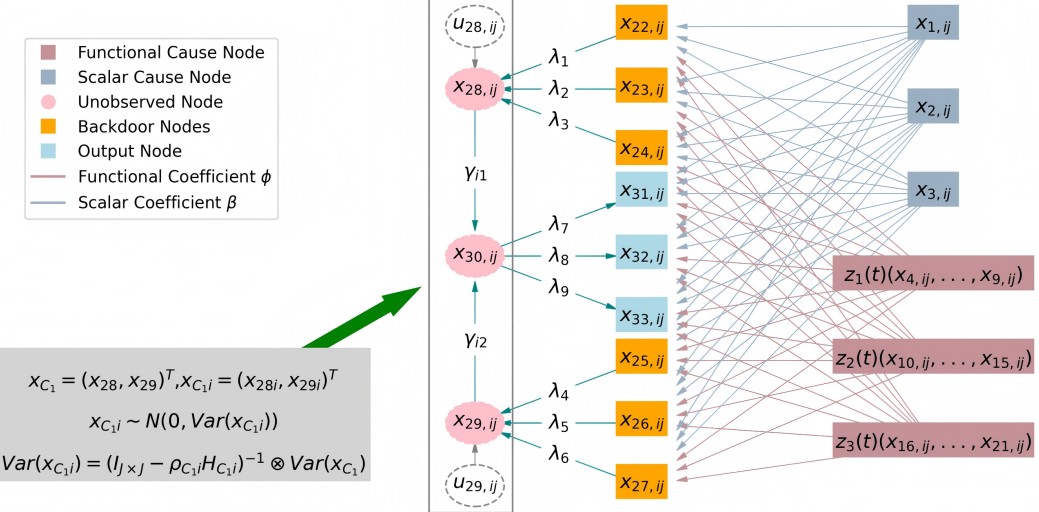

Figure 3: PFST-DSCM with 33 exogenous and endogenous nodes (where nodes $x_{28}$, $x_{29}$ and $x_{30}$ are unmeasured confounders with spatial heterogeneity and temporal dependencies, $z_1(t), z_2(t), z_3(t)$ are functional nodes, and $x_4, \cdots, x_{21}$ are the corresponding base expansion nodes)

The true counterfactual, given the intervention $\mathrm{do}(\mathbf{x}_{\mathbb{B}} := \gamma)$, is

$$\mathbf{x}^{\mathrm{CF}} = f^*(\gamma) + \mathbf{u} + \eta(\gamma, \mathbf{u}).$$

We decompose the counterfactual error as follows

$$\|\hat{\mathbf{x}}^{\mathrm{CF}} - \mathbf{x}^{\mathrm{CF}}\| = \|h(\gamma, \hat{\mathbf{u}}^F) - (f^*(\gamma) + \mathbf{u} + \eta(\gamma, \mathbf{u}))\|$$
$$\leq \|h(\gamma, \hat{\mathbf{u}}^F) - (f^*(\gamma) + \hat{\mathbf{u}}^F)\| + \|(f^*(\gamma) + \hat{\mathbf{u}}^F) - (f^*(\gamma) + \mathbf{u})\| + \|\eta(\gamma, \mathbf{u})\|.$$

Now, we bound each term separatel. First term: By the decoder assumption, $\|h(\gamma, \hat{\mathbf{u}}^F) - (f^*(\gamma) + \hat{\mathbf{u}}^F)\| \leq \tau$; Second term: $\|(f^*(\gamma) + \hat{\mathbf{u}}^F) - (f^*(\gamma) + \mathbf{u})\| = \|\hat{\mathbf{u}}^F - \mathbf{u}\| \leq \delta$; Third term: By the misspecification bound, $\|\eta(\gamma, \mathbf{u})\| \leq \zeta$. Thus, the final bound is:

$$\|\hat{\mathbf{x}}^{\mathrm{CF}} - \mathbf{x}^{\mathrm{CF}}\| \leq \tau + \delta + \zeta.$$

$\square$

## D    EXPERIMENT DETAILS

All experiments were implemented in Python (version 3.9.0) on a Windows server equipped with an Intel i9-13900K processor, NVIDIA RTX 4090 24G GPU, and 128 GB RAM.

### D.1    SIMULATION STUDY: SUPPLEMENTARY MATERIALS

Firstly, functional data $z_{ij}(t)$ was generated from a Gaussian process with $z_{ij}(t) \overset{D}{=} N(\{\sin(2\pi t) + 1.25\}/2, \sigma_x^2)$ for $t \in [0, 1]$, for the coefficient function, we set $c(t) \overset{D}{=} N(\{\sin(2\pi t) + 1.25\}/2, \sigma_c^2)$, where $\sigma_x^2 = 1$ and $\sigma_c^2 = 0.25$, and according to

$$x_{m,ij} = \int \mathbf{b}_m(t) z_{ij}(t) dt, \quad \phi_m = \int \mathbf{b}_m(t) c(t) dt, \tag{16}$$

it was transformed to obtain the corresponding basis expansion node $x_{m,ij}$ and coefficient $\phi_m, m = 1, \cdots, K_n$.

In our implementation of the $\varepsilon_\theta$ model for PFD-BDCM, PFD-DCM, BDCM and DCM, we employed a fully connected neural network architecture with three hidden layers of dimensions [128, 256, 256] using SiLU activation functions (Elfwing et al., 2018). The noise schedule parameters $\beta_t$ and $\alpha_t$ were configured as $\beta_t = (0.1 - 10^{-4})\frac{t-1}{T-1} + 10^{-4}$ and $\alpha_t = \prod_{i=1}^{t}(1 - \beta_t)$ respectively,

with $T$ fixed at 100. We established distinct training regimens for different intervention types: For random interventions sampled uniformly from $\mathrm{Unif}(-3, 3)$, we trained for 500 epochs; For interventions at the $(10\%, 90\%)$ quantiles, we extended training to $15,000$ epochs.

Across all configurations, we maintained a batch size of 64 and a learning rate of $10^{-4}$. The neural network architecture consistently featured three hidden layers with node counts of 128, 256, and 256 in the first, second, and third layers respectively.

In this simulation, we consider $\boldsymbol{G}(\mathbf{x}_{\mathbb{C}_1, ij}) = \mathbf{x}_{\mathbb{C}_1, ij}$ in Eq. 2, the PFD-BDCM model and defined the set of partially functional dynamic structure equations as follows

$$
x_{k,ij} = \begin{cases}
\mathrm{u}_{k,ij}, \text{ for } k = 1, 2, 3, \\
x_{m,ij} + \mathrm{u}_{k,ij}, \text{ for } m = 1, \cdots, 18, k = 4, \cdots, 21, \\
\sum\limits_{q=1}^{3} \beta_{pq} x_{q,ij} + \sum\limits_{q,n=4,1}^{9,6} \phi_{1pn} x_{q,ij} + \sum\limits_{q,n=10,1}^{15,6} \phi_{2pn} x_{q,ij} + \sum\limits_{q,n=16,1}^{21,6} \phi_{3pn} x_{q,ij} \\
+ \mathrm{u}_{k,ij}, \qquad \text{for } k = 22, \cdots, 27, p = 4, \cdots, 9 \\
\sum\limits_{p,q} \lambda_{p2} x_{q,ij} + \mathrm{u}_{k,ij}, \text{ for } k = 28, q = 22, 23, 24, p = 4, 5, 6, \\
\sum\limits_{p,q} \lambda_{p3} x_{q,ij} + \mathrm{u}_{k,ij}, \text{ for } k = 29, q = 25, 26, 27, p = 7, 8, 9, \\
\gamma_{i1} x_{28,ij} + \gamma_{i2} x_{29,ij} + \mathrm{u}_{k,ij}, \text{ for } k = 30, \\
\sum\limits_{q=1}^{3} \beta_{pq} x_{q,ij} + \sum\limits_{q,n=4,1}^{9,6} \phi_{1pn} x_{q,ij} + \sum\limits_{q,n=10,1}^{15,6} \phi_{2pn} x_{q,ij} + \sum\limits_{q,n=16,1}^{21,6} \phi_{3pn} x_{q,ij} \\
+ \lambda_{p1} x_{30,ij} + \mathrm{u}_{k,ij}, \qquad \text{for } k = 31, \cdots, 33, p = 1, \cdots, 3,
\end{cases}
\tag{17}
$$

where the weight coefficients in the graph structure were given by $\boldsymbol{\beta_p} = (\beta_{p1}, \beta_{p2}, \beta_{p3}) = (0.8, 0.8, 0.8)$, $p = 1, \cdots, 9$, and $\lambda_{21} = \lambda_{31} = \lambda_{52} = \lambda_{62} = \lambda_{83} = \lambda_{93} = 0.8$.

In this simulation, since $\mathbf{x}_{\mathbb{C}_1, ij} = (x_{28,ij}, x_{29,ij})^\top$ was two-dimensional, and $\mathbf{x}_{\mathbb{C}_2, ij} = x_{30,ij}$ was one-dimensional, we specified the following multivariate structures for $\mathbf{x}_{\mathbb{C}_1, ij}$ via the linear coregionalization model (LMC) (Song et al., 2012):

$$
\mathbf{x}_{\mathbb{C}_1, ij} = \boldsymbol{A}_1 \mathbf{w}_{ij},
\tag{18}
$$

where $\boldsymbol{A}_1$ is $\dim(\mathbf{x}_{\mathbb{C}_1, ij}) \times \dim(\mathbf{x}_{\mathbb{C}_1, ij})$ upper triangular matrices, $\mathbf{w}_{ij}$ be independent zero-mean and unit-variance random vectors of dimensions $\dim(\mathbf{x}_{\mathbb{C}_1, ij})$, $\mathrm{Cov}(\mathbf{x}_{\mathbb{C}_1}) = \boldsymbol{A}_1 \boldsymbol{A}_1^\top$. In this simulation, we only consider a one-adjacent-time $(l = j \pm 1)$ structure of $\boldsymbol{H}_{\mathbb{C}_1, i}$ and $\boldsymbol{H}_{\mathbb{C}_2, i}$, and assume that $\boldsymbol{H}_{\mathbb{C}_1, i} = \boldsymbol{H}_{\mathbb{C}_2, i} = (h_{jl})_{J \times J}$ in Eq. 4.

We considered the following structures for $\boldsymbol{A}_1$:

$$
\boldsymbol{A}_1 = \begin{pmatrix} a_{11} & a_{12} \\ 0.0 & a_{22} \end{pmatrix},
$$

where $a_{11} = a_{22} = 1.0$, and $a_{12} = 0.5$. For simplicity, we assumed that $\rho_{\mathbf{x}_{\mathbb{C}_1} i} = \rho_{\mathbf{x}_{\mathbb{C}_1}} = 0.15$ was invariant across individuals. We sampled exogenous nodes $\mathrm{u}_{\mathbb{C}_1, i} \sim \mathcal{N}(\mathbf{0}, \mathrm{Cov}(\mathbf{x}_{\mathbb{C}_1})), \mathbb{C}_1 = \{28, 29\}$, $\mathrm{u}_{k,ij} \sim \mathcal{N}(0, 0.8), k = 22, \cdots, 27$ and others from standard normal distribution $\mathcal{N}(0, 1)$.

We investigated three distinct experimental configurations: i): $n = 30, J = 6$ ; ii): $n = 80, J = 6$; iii): $n = 200, J = 6$. For each configuration, we extracted $(90, 240, 600)$ samples respectively using both PFD-DCM, PFD-BDCM, DCM and BDCM frameworks. These models were trained on corresponding sample sizes of $(180, 480, 1200)$. Under these experimental conditions, we employed the proposed PFD-BDCM algorithm to compute causal query results.

**Observational Evaluation.** For outcome variables $\{\mathrm{x}_{31}, \mathrm{x}_{32}, \mathrm{x}_{33}\}$, we generated above three cases samples from both the fitted and true observational distributions, and reported the MMD metrics between these distributions, computed as means over 100 replications per configuration. Comprehensive observational observational query results were presented in Appendix D (Table 3).

**Interventional Evaluation.** We implemented interventions on individual nodes. For an intervention node $k$, we established five randomly sampled intervention values from $\mathrm{Unif}(-3, 3)$ and five intervention values linearly interpolated between the $10\%$ and $90\%$ quantiles of node $k$'s marginal distribution. Subsequently, we generated samples for all three configurations from both the fitted and

true causal models following each intervention. Performance metrics (MMD) were averaged across 5 interventions and 20 replications per case. Intervention targets $\{x_1, x_2, x_3\}$ were selected as maximally upstream nodes to evaluate worst-case intervention complexity. Detailed interventional query results appeared in Appendix D (Tables 4 and 5).

**Counterfactual Evaluation.** Analogous to the interventional protocol, we established identical intervention regimes on individual nodes. For each intervention, we generated $(36, 96, 240)$ non-intervened factual samples $x^{\mathrm{F}}$ across the four configurations. Estimated $(\hat{x}^{\mathrm{CF}})$ and true $(x^{\mathrm{CF}})$ counterfactual values were then computed. We reported the mean MSE averaged over 5 interventions and 20 replications for all target nodes, maintaining the same intervention targets $\{x_1, x_2, x_3\}$. Comprehensive counterfactual query results were provided in Appendix D (Tables 6 and 7).

In summary, we evaluate PFD-BDCM across three dataset scales and two intervention mechanisms. Our results demonstrate consistent superiority over baseline methods. Specifically, for observational queries (Table 3), PFD-BDCM achieves superior MMD metrics, indicating enhanced data generation capability. Analysis of density distributions in Fig. 4 reveals that existing DCM and BDCM exhibit significantly degraded performance in the presence of spatiotemporal dynamic unmeasured confounders, highlighting the necessity of our approach. For counterfactual queries (Tables 6 and 7), PFD-BDCM maintains stable MSE across data scales, confirming robustness compared to PFD-DCM's scale-dependent performance. Comparative analysis of interventional (Tables 4 and 5) versus observational MMD further validates PFD-BDCM's effectiveness in interventional queries. Fig. 6 and 5 demonstrate that our model achieves superior stability compared to all baselines. Additionally, we report per-run computational time (Fig. 6 and 5), showing comparable time consumption between PFD-BDCM and other models, confirming the practical feasibility of our approach.

Table 3: Mean $\pm$ standard deviation of $\mathrm{MMD}^2(\times 10^{-3})$ and Time(seconds) of PFD-DCM and PFD-BDCM compared to the true target distribution(simulation results for observation )

| Type ($J = 6$) | | PFD-BDCM | PFD-DCM | BDCM | DCM |
|---|---|---|---|---|---|
| | | $\mathrm{MMD}^2(\times 10^{-3})$ | | | |
| $n = 30$ | $x_{31}$ | $3.618 \pm 3.820$ | $3.901 \pm 3.498$ | $3.650 \pm 3.615$ | $5.369 \pm 5.295$ |
| | $x_{32}$ | $3.203 \pm 4.829$ | $3.973 \pm 3.502$ | $4.026 \pm 4.363$ | $4.605 \pm 3.732$ |
| | $x_{33}$ | $4.028 \pm 4.454$ | $4.287 \pm 3.755$ | $3.540 \pm 3.149$ | $4.828 \pm 3.720$ |
| | Mean | $3.616 \pm 4.368$ | $4.054 \pm 3.585$ | $3.739 \pm 3.709$ | $4.934 \pm 4.249$ |
| $n = 80$ | $x_{31}$ | $1.496 \pm 1.344$ | $1.659 \pm 1.334$ | $3.354 \pm 4.741$ | $3.869 \pm 4.473$ |
| | $x_{32}$ | $1.523 \pm 1.453$ | $1.468 \pm 1.199$ | $3.479 \pm 5.628$ | $4.618 \pm 7.908$ |
| | $x_{33}$ | $1.464 \pm 1.440$ | $1.554 \pm 1.366$ | $3.294 \pm 4.439$ | $3.623 \pm 3.19$ |
| | Mean | $1.494 \pm 1.412$ | $1.560 \pm 1.300$ | $3.376 \pm 4.936$ | $4.037 \pm 5.191$ |
| $n = 200$ | $x_{31}$ | $0.547 \pm 0.467$ | $0.778 \pm 0.782$ | $3.179 \pm 4.596$ | $3.641 \pm 3.998$ |
| | $x_{32}$ | $0.545 \pm 0.514$ | $0.697 \pm 0.566$ | $3.146 \pm 4.892$ | $3.965 \pm 7.551$ |
| | $x_{33}$ | $0.509 \pm 0.477$ | $0.735 \pm 0.668$ | $2.772 \pm 3.925$ | $2.816 \pm 2.425$ |
| | Mean | $0.533 \pm 0.486$ | $0.737 \pm 0.672$ | $3.032 \pm 4.471$ | $3.474 \pm 4.658$ |
| | | Time(seconds) | | | |
| $n = 30$ | $x_{31}$ | $2.713$ | $2.718$ | $2.711$ | $2.712$ |
| | $x_{32}$ | $2.505$ | $2.520$ | $2.499$ | $2.518$ |
| | $x_{33}$ | $2.777$ | $2.876$ | $2.774$ | $2.868$ |
| | Mean | $2.665$ | $2.705$ | $2.660$ | $2.697$ |
| $n = 80$ | $x_{31}$ | $9.266$ | $9.296$ | $9.262$ | $9.295$ |
| | $x_{32}$ | $8.111$ | $8.212$ | $8.101$ | $8.210$ |
| | $x_{33}$ | $9.133$ | $8.202$ | $9.127$ | $8.195$ |
| | Mean | $8.837$ | $8.570$ | $8.830$ | $8.566$ |
| $n = 200$ | $x_{31}$ | $16.111$ | $16.430$ | $16.104$ | $16.424$ |
| | $x_{32}$ | $16.818$ | $16.616$ | $16.81$ | $16.610$ |
| | $x_{33}$ | $16.083$ | $15.880$ | $16.077$ | $15.872$ |
| Continued on next page | | | | | |

Table 3 continued.

| Type ($J = 6$) | | PFD-BDCM | PFD-DCM | BDCM | DCM |
|---|---|---|---|---|---|
| | Mean | 15.671 | 15.309 | 15.665 | 15.306 |

Table 4: Mean $\pm$ standard deviation of $\text{MMD}^2(\times 10^{-3})$ and Time(seconds) of PFD-BDCM, PFD-DCM, BDCM and PFD-DCM compared to the true target distribution(simulation results for intervention of $\text{Unif}(-3, 3)$ intervention values)

| type ($J = 6$) | | PFD-BDCM | PFD-DCM | BDCM | DCM |
|---|---|---|---|---|---|
| | | $\text{MMD}^2(\times 10^{-3})$(mean $\pm$ sd) | | | |
| | $x_1 \sim x_{31}$ | $3.256 \pm 2.928$ | $4.056 \pm 3.929$ | $3.634 \pm 3.415$ | $4.200 \pm 3.111$ |
| | $x_1 \sim x_{32}$ | $3.467 \pm 2.836$ | $3.887 \pm 3.504$ | $3.551 \pm 3.181$ | $3.492 \pm 3.039$ |
| | $x_1 \sim x_{33}$ | $3.489 \pm 2.903$ | $3.784 \pm 3.276$ | $3.331 \pm 3.409$ | $3.971 \pm 3.43$ |
| | $x_2 \sim x_{31}$ | $3.593 \pm 3.494$ | $4.434 \pm 3.731$ | $4.757 \pm 5.468$ | $4.270 \pm 3.951$ |
| $n = 30$ | $x_2 \sim x_{32}$ | $3.565 \pm 3.338$ | $4.907 \pm 4.638$ | $4.239 \pm 3.857$ | $3.964 \pm 3.631$ |
| | $x_2 \sim x_{33}$ | $4.065 \pm 3.147$ | $4.519 \pm 4.239$ | $4.620 \pm 5.656$ | $4.706 \pm 4.265$ |
| | $x_3 \sim x_{31}$ | $3.680 \pm 2.86$ | $4.422 \pm 4.093$ | $3.839 \pm 3.862$ | $3.289 \pm 2.384$ |
| | $x_3 \sim x_{32}$ | $3.477 \pm 2.869$ | $4.316 \pm 4.004$ | $3.974 \pm 3.197$ | $3.473 \pm 2.761$ |
| | $x_3 \sim x_{33}$ | $3.626 \pm 2.738$ | $4.215 \pm 4.157$ | $3.962 \pm 3.608$ | $3.928 \pm 2.652$ |
| | Mean | $3.58 \pm 3.013$ | $4.282 \pm 3.952$ | $3.990 \pm 3.962$ | $3.922 \pm 3.247$ |
| | $x_1 \sim x_{31}$ | $1.433 \pm 1.34$ | $1.649 \pm 1.313$ | $2.002 \pm 2.136$ | $1.971 \pm 1.685$ |
| | $x_1 \sim x_{32}$ | $1.535 \pm 1.34$ | $1.605 \pm 1.404$ | $2.201 \pm 1.997$ | $2.005 \pm 2.166$ |
| | $x_1 \sim x_{33}$ | $1.425 \pm 1.181$ | $1.439 \pm 1.324$ | $2.035 \pm 1.702$ | $1.853 \pm 1.952$ |
| | $x_2 \sim x_{31}$ | $1.410 \pm 1.132$ | $1.513 \pm 1.28$ | $2.057 \pm 2.241$ | $2.361 \pm 2.161$ |
| $n = 80$ | $x_2 \sim x_{32}$ | $1.499 \pm 1.223$ | $1.589 \pm 1.324$ | $1.904 \pm 2.235$ | $2.171 \pm 1.94$ |
| | $x_2 \sim x_{33}$ | $1.425 \pm 1.047$ | $1.541 \pm 1.366$ | $2.152 \pm 2.396$ | $2.327 \pm 2.424$ |
| | $x_3 \sim x_{31}$ | $1.355 \pm 1.268$ | $1.336 \pm 1.016$ | $1.745 \pm 1.618$ | $2.068 \pm 1.993$ |
| | $x_3 \sim x_{32}$ | $1.283 \pm 1.09$ | $1.546 \pm 1.227$ | $1.981 \pm 2.131$ | $1.936 \pm 1.830$ |
| | $x_3 \sim x_{33}$ | $1.309 \pm 1.215$ | $1.385 \pm 1.159$ | $2.004 \pm 2.010$ | $2.017 \pm 1.88$ |
| | Mean | $1.408 \pm 1.204$ | $1.511 \pm 1.268$ | $2.009 \pm 2.052$ | $2.079 \pm 2.003$ |
| | $x_1 \sim x_{31}$ | $0.572 \pm 0.425$ | $0.467 \pm 0.515$ | $2.925 \pm 3.850$ | $2.812 \pm 2.824$ |
| | $x_1 \sim x_{32}$ | $0.542 \pm 0.456$ | $0.466 \pm 0.42$ | $2.614 \pm 3.202$ | $3.266 \pm 3.599$ |
| | $x_1 \sim x_{33}$ | $0.543 \pm 0.395$ | $0.469 \pm 0.434$ | $2.575 \pm 3.113$ | $2.802 \pm 2.805$ |
| | $x_2 \sim x_{31}$ | $0.677 \pm 0.610$ | $0.668 \pm 0.786$ | $3.145 \pm 4.450$ | $3.232 \pm 2.826$ |
| $n = 200$ | $x_2 \sim x_{32}$ | $0.644 \pm 0.585$ | $0.641 \pm 0.658$ | $2.895 \pm 4.187$ | $3.583 \pm 3.704$ |
| | $x_2 \sim x_{33}$ | $0.608 \pm 0.517$ | $0.646 \pm 0.774$ | $2.654 \pm 3.532$ | $3.142 \pm 2.911$ |
| | $x_3 \sim x_{31}$ | $0.595 \pm 0.523$ | $0.673 \pm 0.554$ | $2.869 \pm 3.629$ | $3.124 \pm 2.565$ |
| | $x_3 \sim x_{32}$ | $0.581 \pm 0.541$ | $0.652 \pm 0.599$ | $2.723 \pm 3.521$ | $3.705 \pm 3.919$ |
| | $x_3 \sim x_{33}$ | $0.593 \pm 0.486$ | $0.648 \pm 0.500$ | $2.828 \pm 3.098$ | $3.017 \pm 3.313$ |
| | Mean | $0.595 \pm 0.504$ | $0.592 \pm 0.582$ | $2.803 \pm 3.620$ | $3.187 \pm 3.163$ |
| | | Time(seconds) | | | |
| | $x_1 \sim x_{31}$ | 2.239 | 2.154 | 2.237 | 2.149 |
| | $x_1 \sim x_{32}$ | 2.117 | 2.302 | 2.109 | 2.295 |
| | $x_1 \sim x_{33}$ | 2.282 | 2.133 | 2.275 | 2.131 |
| | $x_2 \sim x_{31}$ | 2.312 | 2.177 | 2.304 | 2.170 |
| $n = 30$ | $x_2 \sim x_{32}$ | 2.495 | 2.225 | 2.494 | 2.218 |
| | $x_2 \sim x_{33}$ | 2.250 | 2.113 | 2.248 | 2.106 |
| | $x_3 \sim x_{31}$ | 2.359 | 2.189 | 2.356 | 2.187 |
| | $x_3 \sim x_{32}$ | 2.244 | 2.111 | 2.24 | 2.108 |
| | $x_3 \sim x_{33}$ | 2.318 | 2.132 | 2.308 | 2.122 |
| | Mean | 2.291 | 2.171 | 2.286 | 2.165 |
| | $x_1 \sim x_{31}$ | 6.311 | 6.276 | 6.308 | 6.272 |
| | $x_1 \sim x_{32}$ | 6.630 | 6.497 | 6.620 | 6.496 |
| | $x_1 \sim x_{33}$ | 6.458 | 6.416 | 6.449 | 6.407 |

Continued on next page

Table 4 continued.

| type ($J = 6$) | | PFD-BDCM | PFD-DCM | BDCM | DCM |
|---|---|---|---|---|---|
| | $x_2 \sim x_{31}$ | 6.145 | 6.117 | 6.144 | 6.116 |
| $n = 80$ | $x_2 \sim x_{32}$ | 5.688 | 5.702 | 5.686 | 5.698 |
| | $x_2 \sim x_{33}$ | 5.661 | 5.646 | 5.659 | 5.644 |
| | $x_3 \sim x_{31}$ | 5.689 | 5.733 | 5.686 | 5.730 |
| | $x_3 \sim x_{32}$ | 5.636 | 5.634 | 5.631 | 5.627 |
| | $x_3 \sim x_{33}$ | 5.755 | 5.727 | 5.748 | 5.727 |
| | Mean | 5.997 | 5.972 | 5.992 | 5.969 |
| | $x_1 \sim x_{31}$ | 15.434 | 15.390 | 15.432 | 15.383 |
| | $x_1 \sim x_{32}$ | 15.451 | 15.460 | 15.442 | 15.454 |
| | $x_1 \sim x_{33}$ | 15.522 | 15.484 | 15.514 | 15.477 |
| $n = 200$ | $x_2 \sim x_{31}$ | 15.604 | 15.576 | 15.595 | 15.575 |
| | $x_2 \sim x_{32}$ | 15.541 | 15.524 | 15.538 | 15.516 |
| | $x_2 \sim x_{33}$ | 15.382 | 15.376 | 15.374 | 15.375 |
| | $x_3 \sim x_{31}$ | 16.007 | 15.996 | 16.006 | 15.991 |
| | $x_3 \sim x_{32}$ | 15.790 | 15.776 | 15.783 | 15.775 |
| | $x_3 \sim x_{33}$ | 15.527 | 15.502 | 15.523 | 15.499 |
| | Mean | 15.584 | 15.565 | 15.579 | 15.560 |

Table 5: Mean $\pm$ standard deviation of MMD$^2 (\times 10^{-3})$ and Time(seconds) of PFD-DCM and PFD-BDCM compared to the true target distribution(simulation results for intervention of $(10\%, 90\%)$ quantiles intervention values

| Type ($J = 6$) | Causal $\sim$ Outcome | PFD-BDCM MMD | PFD-BDCM Time | PFD-DCM MMD | PFD-DCM Time |
|---|---|---|---|---|---|
| | $x_1 \sim x_{31}$ | $3.656 \pm 3.306$ | 20.732 | $4.042 \pm 3.433$ | 20.686 |
| | $x_1 \sim x_{32}$ | $3.322 \pm 3.005$ | 25.514 | $4.082 \pm 3.419$ | 25.476 |
| | $x_1 \sim x_{33}$ | $3.657 \pm 3.584$ | 25.324 | $3.911 \pm 3.443$ | 25.305 |
| | $x_2 \sim x_{31}$ | $3.598 \pm 3.485$ | 25.501 | $3.864 \pm 3.572$ | 25.476 |
| $n = 30$ | $x_2 \sim x_{32}$ | $3.684 \pm 3.913$ | 25.633 | $4.264 \pm 4.102$ | 25.568 |
| | $x_2 \sim x_{33}$ | $3.660 \pm 3.158$ | 25.485 | $3.820 \pm 3.610$ | 25.503 |
| | $x_3 \sim x_{31}$ | $4.144 \pm 3.359$ | 25.267 | $4.148 \pm 3.145$ | 25.28 |
| | $x_3 \sim x_{32}$ | $4.485 \pm 3.868$ | 23.762 | $4.056 \pm 3.530$ | 23.714 |
| | $x_3 \sim x_{33}$ | $4.321 \pm 4.069$ | 23.122 | $4.071 \pm 3.003$ | 23.080 |
| | Mean | $3.836 \pm 3.527$ | 24.482 | $4.029 \pm 3.473$ | 24.454 |
| | $x_1 \sim x_{31}$ | $1.402 \pm 1.185$ | 63.060 | $1.240 \pm 1.147$ | 63.062 |
| | $x_1 \sim x_{32}$ | $1.358 \pm 1.191$ | 68.006 | $1.362 \pm 1.202$ | 67.936 |
| | $x_1 \sim x_{33}$ | $1.392 \pm 1.266$ | 66.797 | $1.258 \pm 1.158$ | 66.567 |
| $n = 80$ | $x_2 \sim x_{31}$ | $1.676 \pm 1.498$ | 57.344 | $1.616 \pm 1.368$ | 57.375 |
| | $x_2 \sim x_{32}$ | $1.572 \pm 1.310$ | 54.770 | $1.544 \pm 1.531$ | 54.955 |
| | $x_2 \sim x_{33}$ | $1.666 \pm 1.531$ | 54.763 | $1.524 \pm 1.450$ | 54.582 |
| | $x_3 \sim x_{31}$ | $1.347 \pm 1.109$ | 51.399 | $1.460 \pm 1.285$ | 51.435 |
| | $x_3 \sim x_{32}$ | $1.243 \pm 1.121$ | 45.124 | $1.439 \pm 1.392$ | 45.020 |
| | $x_3 \sim x_{33}$ | $1.340 \pm 1.120$ | 42.084 | $1.429 \pm 1.381$ | 41.984 |
| | Mean | $1.444 \pm 1.259$ | 55.927 | $1.430 \pm 1.324$ | 55.879 |
| | $x_1 \sim x_{31}$ | $0.610 \pm 0.538$ | 164.236 | $0.607 \pm 0.565$ | 164.12 |
| | $x_1 \sim x_{32}$ | $0.631 \pm 0.595$ | 145.306 | $0.597 \pm 0.553$ | 144.961 |
| | $x_1 \sim x_{33}$ | $0.651 \pm 0.614$ | 129.078 | $0.592 \pm 0.554$ | 128.734 |
| | $x_2 \sim x_{31}$ | $0.657 \pm 0.604$ | 99.195 | $0.606 \pm 0.563$ | 99.099 |
| $n = 200$ | $x_2 \sim x_{32}$ | $0.680 \pm 0.638$ | 89.394 | $0.626 \pm 0.564$ | 90.120 |
| | $x_2 \sim x_{33}$ | $0.681 \pm 0.666$ | 164.753 | $0.590 \pm 0.539$ | 164.494 |
| | $x_3 \sim x_{31}$ | $0.657 \pm 0.634$ | 143.317 | $0.696 \pm 0.687$ | 143.007 |
| | $x_3 \sim x_{32}$ | $0.635 \pm 0.641$ | 45.382 | $0.688 \pm 0.568$ | 45.364 |
| | $x_3 \sim x_{33}$ | $0.611 \pm 0.597$ | 35.955 | $0.667 \pm 0.604$ | 36.381 |
| | Mean | $0.646 \pm 0.614$ | 112.957 | $0.630 \pm 0.578$ | 112.920 |

Table 6: Mean $\pm$ standard deviation of MSE and Time(seconds) of PFD-BDCM, PFD-DCM,BDCM and DCM compared to the true target distribution(simulation results for Counterfactual of $\mathrm{Unif}(-3,3)$ intervention values) )

| type ($J=6$) | | PFD-BDCM | PFD-DCM | BDCM | DCM |
|---|---|---|---|---|---|
| | | \multicolumn{4}{c}{MSE(mean $\pm$ sd)} | | | |
| | $x_1 \sim x_{31}$ | $0.824 \pm 0.252$ | $0.649 \pm 0.254$ | $1.941 \pm 0.070$ | $1.946 \pm 0.078$ |
| | $x_1 \sim x_{32}$ | $0.837 \pm 0.250$ | $0.630 \pm 0.194$ | $1.95 \pm 0.064$ | $1.935 \pm 0.085$ |
| | $x_1 \sim x_{33}$ | $0.819 \pm 0.260$ | $0.597 \pm 0.190$ | $1.936 \pm 0.093$ | $1.944 \pm 0.093$ |
| | $x_2 \sim x_{31}$ | $0.852 \pm 0.251$ | $0.673 \pm 0.287$ | $1.934 \pm 0.085$ | $1.951 \pm 0.068$ |
| $n=30$ | $x_2 \sim x_{32}$ | $0.809 \pm 0.220$ | $0.639 \pm 0.249$ | $1.948 \pm 0.078$ | $1.952 \pm 0.066$ |
| | $x_2 \sim x_{33}$ | $0.836 \pm 0.276$ | $0.639 \pm 0.254$ | $1.931 \pm 0.088$ | $1.959 \pm 0.067$ |
| | $x_3 \sim x_{31}$ | $0.825 \pm 0.272$ | $0.639 \pm 0.249$ | $1.934 \pm 0.083$ | $1.939 \pm 0.072$ |
| | $x_3 \sim x_{32}$ | $0.847 \pm 0.269$ | $0.644 \pm 0.234$ | $1.914 \pm 0.112$ | $1.947 \pm 0.071$ |
| | $x_3 \sim x_{33}$ | $0.864 \pm 0.278$ | $0.617 \pm 0.215$ | $1.933 \pm 0.092$ | $1.947 \pm 0.079$ |
| | Mean | $0.835 \pm 0.259$ | $0.636 \pm 0.236$ | $1.936 \pm 0.085$ | $1.947 \pm 0.075$ |
| | $x_1 \sim x_{31}$ | $0.659 \pm 0.123$ | $0.216 \pm 0.053$ | $1.984 \pm 0.020$ | $1.982 \pm 0.024$ |
| | $x_1 \sim x_{32}$ | $0.661 \pm 0.122$ | $0.217 \pm 0.051$ | $1.979 \pm 0.029$ | $1.978 \pm 0.029$ |
| | $x_1 \sim x_{33}$ | $0.660 \pm 0.129$ | $0.213 \pm 0.074$ | $1.981 \pm 0.026$ | $1.983 \pm 0.021$ |
| | $x_2 \sim x_{31}$ | $0.618 \pm 0.128$ | $0.2 \pm 0.048$ | $1.981 \pm 0.029$ | $1.974 \pm 0.031$ |
| $n=80$ | $x_2 \sim x_{32}$ | $0.627 \pm 0.150$ | $0.214 \pm 0.047$ | $1.977 \pm 0.029$ | $1.976 \pm 0.029$ |
| | $x_2 \sim x_{33}$ | $0.641 \pm 0.129$ | $0.217 \pm 0.071$ | $1.982 \pm 0.027$ | $1.978 \pm 0.031$ |
| | $x_3 \sim x_{31}$ | $0.637 \pm 0.135$ | $0.202 \pm 0.041$ | $1.985 \pm 0.023$ | $1.978 \pm 0.032$ |
| | $x_3 \sim x_{32}$ | $0.642 \pm 0.125$ | $0.218 \pm 0.046$ | $1.989 \pm 0.015$ | $1.983 \pm 0.021$ |
| | $x_3 \sim x_{33}$ | $0.663 \pm 0.131$ | $0.214 \pm 0.062$ | $1.984 \pm 0.019$ | $1.983 \pm 0.023$ |
| | Mean | $0.645 \pm 0.130$ | $0.212 \pm 0.055$ | $1.982 \pm 0.024$ | $1.980 \pm 0.027$ |
| | $x_1 \sim x_{31}$ | $0.610 \pm 0.094$ | $0.075 \pm 0.018$ | $1.991 \pm 0.010$ | $1.994 \pm 0.015$ |
| | $x_1 \sim x_{32}$ | $0.615 \pm 0.092$ | $0.085 \pm 0.025$ | $1.991 \pm 0.01$ | $1.994 \pm 0.011$ |
| | $x_1 \sim x_{33}$ | $0.608 \pm 0.094$ | $0.084 \pm 0.020$ | $1.991 \pm 0.009$ | $1.994 \pm 0.013$ |
| $n=200$ | $x_2 \sim x_{31}$ | $0.602 \pm 0.095$ | $0.081 \pm 0.018$ | $1.991 \pm 0.010$ | $1.992 \pm 0.012$ |
| | $x_2 \sim x_{32}$ | $0.604 \pm 0.094$ | $0.089 \pm 0.028$ | $1.989 \pm 0.014$ | $1.989 \pm 0.013$ |
| | $x_2 \sim x_{33}$ | $0.599 \pm 0.093$ | $0.078 \pm 0.018$ | $1.990 \pm 0.009$ | $1.992 \pm 0.012$ |
| | $x_3 \sim x_{31}$ | $0.592 \pm 0.074$ | $0.074 \pm 0.015$ | $1.991 \pm 0.011$ | $1.992 \pm 0.013$ |
| | $x_3 \sim x_{32}$ | $0.593 \pm 0.084$ | $0.084 \pm 0.022$ | $1.992 \pm 0.009$ | $1.994 \pm 0.012$ |
| | $x_3 \sim x_{33}$ | $0.589 \pm 0.082$ | $0.081 \pm 0.019$ | $1.989 \pm 0.010$ | $1.992 \pm 0.015$ |
| | Mean | $0.601 \pm 0.089$ | $0.081 \pm 0.020$ | $1.990 \pm 0.010$ | $1.992 \pm 0.013$ |
| | | \multicolumn{4}{c}{Time(seconds)} | | | |
| | $x_1 \sim x_{31}$ | 1.172 | 1.061 | 1.033 | 1.051 |
| | $x_1 \sim x_{32}$ | 1.179 | 1.065 | 1.146 | 1.053 |
| | $x_1 \sim x_{33}$ | 1.164 | 1.046 | 1.114 | 1.017 |
| | $x_2 \sim x_{31}$ | 1.143 | 1.050 | 1.100 | 0.981 |
| $n=30$ | $x_2 \sim x_{32}$ | 1.131 | 1.066 | 1.181 | 1.061 |
| | $x_2 \sim x_{33}$ | 1.126 | 1.021 | 1.146 | 1.014 |
| | $x_3 \sim x_{31}$ | 1.150 | 1.058 | 1.198 | 1.086 |
| | $x_3 \sim x_{32}$ | 1.142 | 1.056 | 1.125 | 1.045 |
| | $x_3 \sim x_{33}$ | 1.012 | 0.913 | 1.185 | 1.071 |
| | Mean | 1.135 | 1.037 | 1.136 | 1.042 |
| | $x_1 \sim x_{31}$ | 4.119 | 3.989 | 4.136 | 4.097 |
| | $x_1 \sim x_{32}$ | 4.167 | 3.952 | 4.374 | 4.306 |
| | $x_1 \sim x_{33}$ | 4.230 | 4.021 | 4.575 | 4.472 |
| | $x_2 \sim x_{31}$ | 3.395 | 3.196 | 3.492 | 3.592 |
| $n=80$ | $x_2 \sim x_{32}$ | 3.203 | 3.005 | 3.409 | 3.392 |
| | $x_2 \sim x_{33}$ | 3.221 | 3.018 | 3.128 | 3.115 |

Continued on next page

Table 6 continued.

| type ($J = 6$) | | PFD-BDCM | PFD-DCM | BDCM | DCM |
|---|---|---|---|---|---|
| | $x_3 \sim x_{31}$ | 3.419 | 3.230 | 3.395 | 3.447 |
| | $x_3 \sim x_{32}$ | 3.573 | 3.371 | 3.564 | 3.413 |
| | $x_3 \sim x_{33}$ | 3.299 | 3.103 | 3.990 | 3.927 |
| | Mean | 3.625 | 3.432 | 3.812 | 3.751 |
| | $x_1 \sim x_{31}$ | 10.839 | 10.877 | 10.54 | 10.312 |
| | $x_1 \sim x_{32}$ | 10.570 | 10.504 | 9.65 | 9.599 |
| | $x_1 \sim x_{33}$ | 8.029 | 8.030 | 7.592 | 7.310 |
| $n = 200$ | $x_2 \sim x_{31}$ | 8.016 | 8.022 | 7.913 | 7.153 |
| | $x_2 \sim x_{32}$ | 7.946 | 7.951 | 7.074 | 7.603 |
| | $x_2 \sim x_{33}$ | 7.999 | 8.024 | 7.471 | 7.306 |
| | $x_3 \sim x_{31}$ | 7.958 | 7.985 | 7.795 | 7.670 |
| | $x_3 \sim x_{32}$ | 8.138 | 8.141 | 7.925 | 7.558 |
| | $x_3 \sim x_{33}$ | 8.047 | 8.054 | 7.601 | 7.694 |
| | Mean | 8.616 | 8.621 | 8.173 | 8.023 |

Table 7: Mean $\pm$ standard deviation of MSE and time(s) of PPFD-BDCM and FD-DCM compared to the true target distribution(simulation results for Counterfactual of $(10\%, 90\%)$ quantiles intervention values)

| type ($J = 6$) | Causal $\sim$ Outcome | PFD-BDCM MSE | PFD-BDCM Time | PFD-DCM MSE | PFD-DCM Time |
|---|---|---|---|---|---|
| | $x_1 \sim x_{31}$ | $0.736 \pm 0.221$ | 10.219 | $0.448 \pm 0.146$ | 10.210 |
| | $x_1 \sim x_{32}$ | $0.755 \pm 0.223$ | 10.174 | $0.510 \pm 0.204$ | 10.185 |
| | $x_1 \sim x_{33}$ | $0.728 \pm 0.233$ | 10.117 | $0.512 \pm 0.192$ | 10.097 |
| | $x_2 \sim x_{31}$ | $0.707 \pm 0.284$ | 10.194 | $0.443 \pm 0.172$ | 10.196 |
| $n = 30$ | $x_2 \sim x_{32}$ | $0.745 \pm 0.265$ | 10.141 | $0.503 \pm 0.246$ | 10.129 |
| | $x_2 \sim x_{33}$ | $0.749 \pm 0.274$ | 10.141 | $0.521 \pm 0.195$ | 10.134 |
| | $x_3 \sim x_{31}$ | $0.741 \pm 0.238$ | 10.115 | $0.434 \pm 0.121$ | 10.111 |
| | $x_3 \sim x_{32}$ | $0.723 \pm 0.246$ | 10.007 | $0.498 \pm 0.242$ | 10.006 |
| | $x_3 \sim x_{33}$ | $0.739 \pm 0.207$ | 10.164 | $0.469 \pm 0.145$ | 10.161 |
| | Mean | $0.736 \pm 0.244$ | 10.141 | $0.482 \pm 0.185$ | 10.137 |
| | $x_1 \sim x_{31}$ | $0.594 \pm 0.134$ | 27.008 | $0.164 \pm 0.045$ | 27.085 |
| | $x_1 \sim x_{32}$ | $0.616 \pm 0.133$ | 27.107 | $0.179 \pm 0.051$ | 27.089 |
| | $x_1 \sim x_{33}$ | $0.596 \pm 0.123$ | 26.883 | $0.173 \pm 0.046$ | 26.843 |
| | $x_2 \sim x_{31}$ | $0.562 \pm 0.119$ | 26.178 | $0.161 \pm 0.035$ | 26.118 |
| $n = 80$ | $x_2 \sim x_{32}$ | $0.564 \pm 0.129$ | 22.461 | $0.178 \pm 0.056$ | 22.389 |
| | $x_2 \sim x_{33}$ | $0.568 \pm 0.127$ | 21.731 | $0.177 \pm 0.051$ | 21.67 |
| | $x_3 \sim x_{31}$ | $0.589 \pm 0.141$ | 21.778 | $0.164 \pm 0.041$ | 21.778 |
| | $x_3 \sim x_{32}$ | $0.591 \pm 0.145$ | 21.649 | $0.179 \pm 0.048$ | 21.649 |
| | $x_3 \sim x_{33}$ | $0.593 \pm 0.134$ | 21.352 | $0.179 \pm 0.048$ | 21.312 |
| | Mean | $0.586 \pm 0.132$ | 24.016 | $0.173 \pm 0.047$ | 23.993 |
| | $x_1 \sim x_{31}$ | $0.563 \pm 0.088$ | 67.328 | $0.066 \pm 0.01$ | 67.338 |
| | $x_1 \sim x_{32}$ | $0.557 \pm 0.094$ | 62.729 | $0.064 \pm 0.012$ | 62.456 |
| | $x_1 \sim x_{33}$ | $0.563 \pm 0.086$ | 53.878 | $0.066 \pm 0.017$ | 53.876 |
| $n = 200$ | $x_2 \sim x_{31}$ | $0.555 \pm 0.081$ | 50.887 | $0.067 \pm 0.012$ | 50.931 |
| | $x_2 \sim x_{32}$ | $0.556 \pm 0.087$ | 42.043 | $0.067 \pm 0.014$ | 41.990 |
| | $x_2 \sim x_{33}$ | $0.556 \pm 0.081$ | 34.418 | $0.065 \pm 0.015$ | 34.428 |
| | $x_3 \sim x_{31}$ | $0.547 \pm 0.077$ | 66.902 | $0.065 \pm 0.011$ | 67.118 |
| | $x_3 \sim x_{32}$ | $0.546 \pm 0.081$ | 62.171 | $0.064 \pm 0.012$ | 62.027 |
| | $x_3 \sim x_{33}$ | $0.540 \pm 0.077$ | 48.415 | $0.063 \pm 0.014$ | 48.431 |
| | Mean | $0.554 \pm 0.084$ | 54.308 | $0.065 \pm 0.013$ | 54.288 |

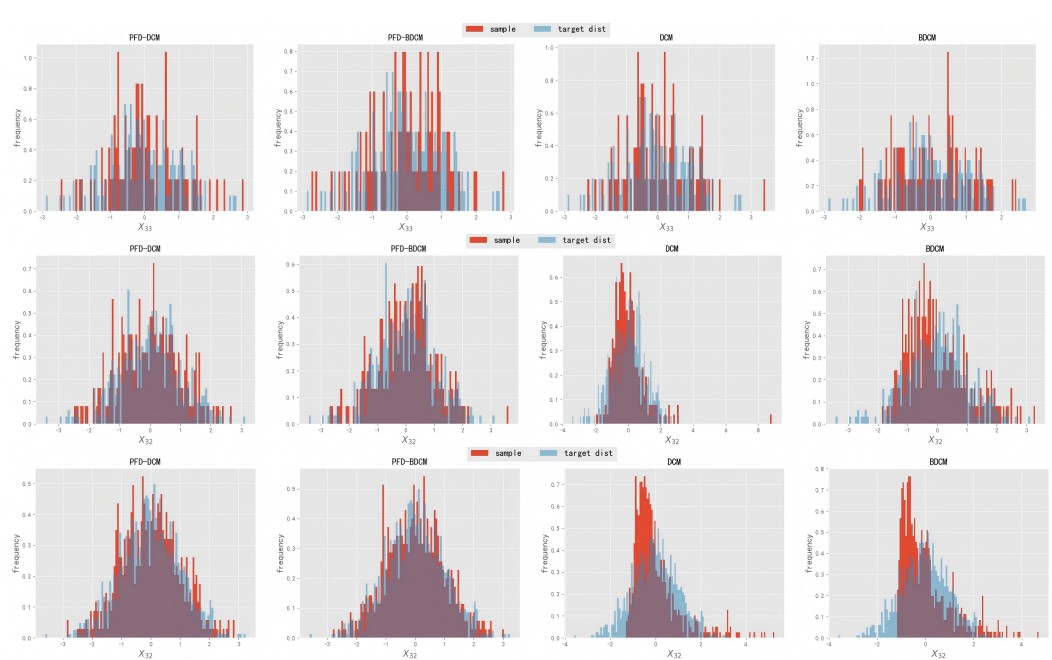

Figure 4: Empirical distributions of x$_{31}$ (From left to right, and from top to bottom: $type_1(n = 30, J = 6), type_2(n = 80, J = 6), type_3(n = 200, J = 6)$) sampled from PFD-DCM, PFD-BDCM, DCM and BDCM compared to the ground-truth target distribution (observation query) for simulation study

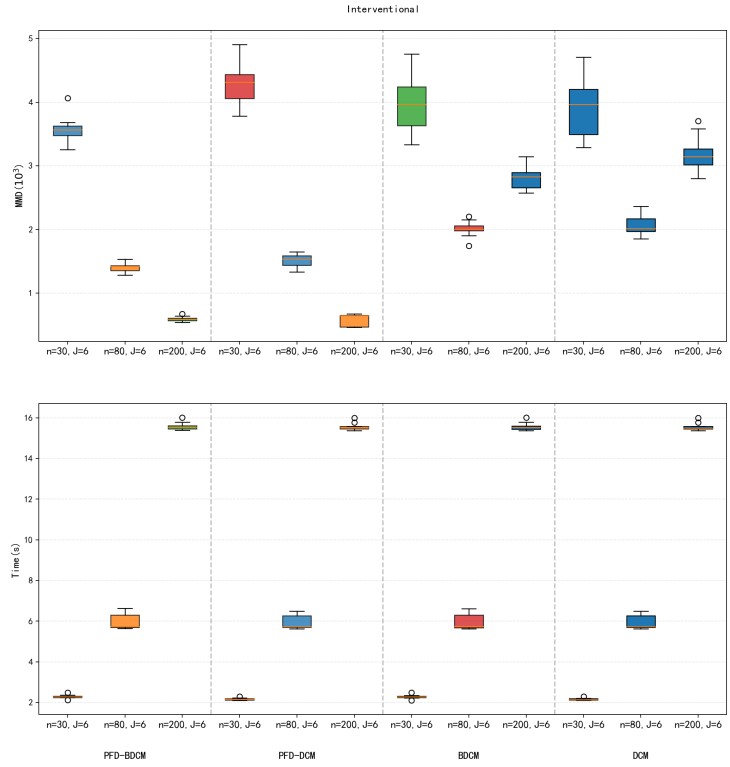

Figure 5: Box plots of interventional queries of the PFD-BDCM, PFD-DCM, BDCM and DCM over $9 * 20$ random initializations of the model and training data.

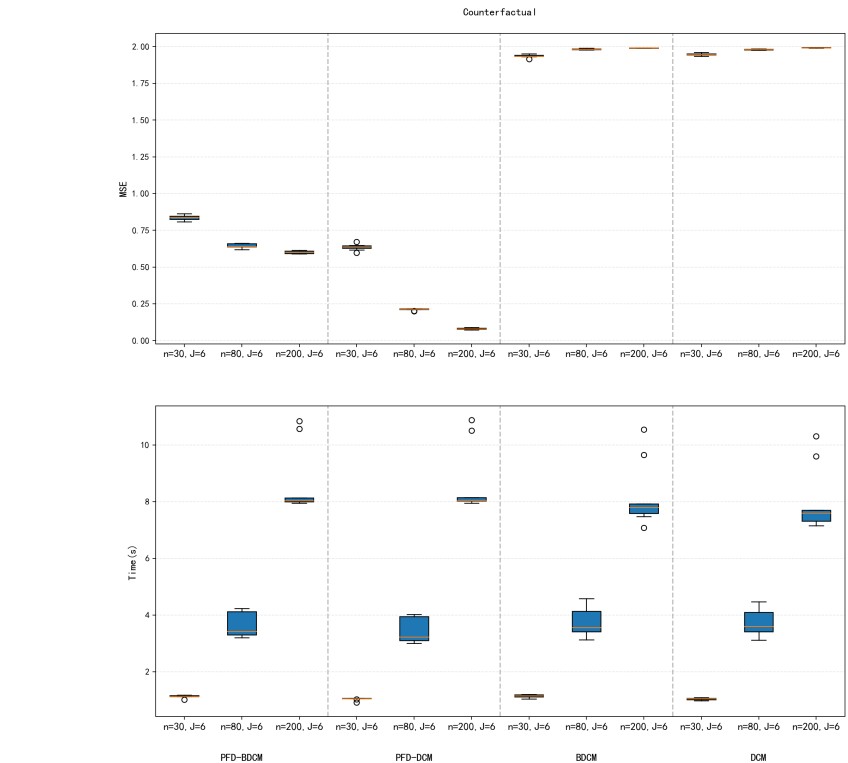

Figure 6: Box plots of counterfactual queries of the PFD-BDCM, PFD-DCM, BDCM and DCMand over $9 * 20$ random initializations of the model and training data.

## D.2 A REAL EXAMPLE: SUPPLEMENTARY MATERIALS

Our study integrated China's provincial $CO_2$ emission inventories from the China Emission Accounts and Datasets (CEADs)and emissions data for nine atmospheric pollutants from the Multi-scale Emission Inventory of China (MEIC) as response variables for air pollutant emissions.

The comprehensive dataset comprised 118 indicator variables, with meteorological data sourced from ERA5, environmental concern metrics derived from Baidu indices, and remaining variables extracted from the China Statistical Yearbook. Through collinearity diagnostics and random forest-based feature selection, we retained 49 statistically robust indicators for subsequent modeling. And conditioning factors were categorized into ten thematic groups comprising 80 nodes. Weight coefficients for the PFD-BDCM framework were derived from Dynamic Structural Equation Model (DSEM) estimated parameters (Song et al., 2012; Tang et al., 2017).

Table 8: Pollutant Emission Indicators and Influencing Factors Indicators(59)

| Indicator category | Number | Indicator Name |
|---|---|---|
| Emission of Air Pollutants | 10 | $SO_2$ |
| | | $NO_x$ |
| | | CO |
| | | VOC |
| | | $NH_3$ |
| | | $PM_{10}$ |
| | | $PM_{2.5}$ |
| | | BC |
| | | OC |
| | Continued on next page | |

Table 8 continued.

| Indicator | Number | Indicator Name |
|---|---|---|
| | | $CO_2$ |
| Meteorology | 5 | Runoff |
| | | Surface runoff |
| | | Total precipitation) |
| | | vertical-velocity-high |
| | | wind-high |
| Economic | 6 | GDP(per capita) |
| | | Industrial structure - first |
| | | Industrial added value |
| | | Disposable Personal Income(Urban) |
| | | Consumer Expenditure (Urban) |
| | | Private Automobile Ownership |
| Population structure | 5 | Fertility Rate |
| | | Demographic Aging |
| | | Mortality Rate |
| | | Natural Population Growth Rate |
| | | Unemployment Rate |
| Forest fires and and natural disasters | 5 | Frequency of Forest Fires |
| | | Casualties of Forest Fires |
| | | Affected Forest Area by Fires |
| | | Direct Economic Loss from Natural Disasters |
| | | Number of People Affected by Natural Disasters |
| Urban Vegetation and Sci-Tech innovation | 6 | Urban Green Space Coverage Rate |
| | | Urban Green Space Area |
| | | High Vegetation Cover |
| | | Low Vegetation Cover |
| | | Number of Valid Invention Patents Held |
| | | Technology Market Transaction Volume |
| Road&T raffic | 4 | Public Bus and Trolleybus Passenger Volume |
| | | Public Bus and Trolleybus Operation Count |
| | | Urban Roadway Lighting Fixtures |
| | | Urban Road Length |
| Urbanization | 3 | Urbanized Area |
| | | Urban Population Density |
| | | Urban Population Density |
| Family scale | 3 | Family scale-Small $(1-3)$ |
| | | Family scale-Medium$(4-6)$ |
| | | Family scale-Large$(7-)$ |
| Education | 3 | College or above |
| | | Senior middle school |
| | | Junior middle school and below |
| Environmental Concern | 4 | Environmental Pollution-Annual Search Index |
| | | Environmental Pollution-Annual Media Index |
| | | Haze-Annual Search Index |
| | | Haze-Annual Medis Index |
| $x_1$ | 1 | Environmental protection |
| $x_2$ | 1 | Government intervention |
| $x_3$ | 1 | Foreign direct investment (FDI) |
| $z_1(t)$ | 1 | Sas level pressure |

| Table 8 continued. | | |
|---|---|---|
| Indicator | Number | Indicator Name |
| $z_2(t)$ | 1 | 2 m temperature |

For simplicity, we assumed that they were continuous and

$$y_{pij} \stackrel{D}{=} N\left(\boldsymbol{\Lambda_p}.\boldsymbol{\omega}_{ij}, \psi_p\right), p = 1, \cdots, 54, i = 1, \cdots, 30, j = 1, \cdots, 6,$$

$$\eta_{ij} = \boldsymbol{\beta}\boldsymbol{x}_{ij} + \sum_{k=1}^{K_n} \boldsymbol{\phi}_k \boldsymbol{z}_{kij} + \sum_{h=1}^{10} \gamma_{hi}\xi_{hij} + \delta_{ij}, \tag{19}$$

where the first 10 variables $(y_1, \cdots, y_{10})$ were interpreted as "Emission inventory $(\eta)$", while the variables $(y_{11}, \cdots, y_{15})$, $(y_{16}, \cdots, y_{21})$ ,$(y_{22}, \cdots, y_{26})$, $(y_{27}, \cdots, y_{31})$, $(y_{32}, \cdots, y_{37})$, $(y_{38}, \cdots, y_{41})$, $(y_{42}, y_{43}, y_{44})$, $(y_{45}, y_{46}, y_{47})$, $(y_{48}, y_{49}, y_{50})$, $(y_{51}, \cdots, y_{54})$ were interpreted as "Meteorology $(\xi_1)$", "Economic $(\xi_2)$", "Population structure $(\xi_3)$", "Forest fires & natural disasters $(\xi_4)$", "Vegetation Cover& Sci-Tech innovation $(\xi_5)$", "Road & Traffic $(\xi_6)$", "Urbanization level $(\xi_7)$", "Family scale $(\xi_8)$", "Education $(\xi_9)$", and "Environmental Concern $(\xi_{10})$" respectively. $\boldsymbol{\omega}_{ij} = (\eta_{ij}, \boldsymbol{\xi}_{ij}^{\top})^{\top}$, $\boldsymbol{z}_{kij} = \int_{\mathcal{T}} \mathbf{b}_k(t)\boldsymbol{z}_{ij}(t)dt$. Scalar covariates $(x_1, x_2, x_3)$ corresponded to "Environmental governance", "Government intervention", and "Foreign direct investment (FDI)", respectively. Functional covariates $(z_1(t), z_2(t))$ represented "Sea level pressure" and "2 m temperature". To ensure scale uniformity, we standardized all raw data using complete observations.

We adopted the DSEM framework specified in Eq. (19), defining the partially functional dynamic structural equations as:

$$x_{k,ij} = \begin{cases} \mathcal{N}\left(0, \Sigma_{\xi ki}\right), \text{ for } k = 1, \cdots, 10, \\ \lambda_k x_{m,ij} + u_{k,ij}, \text{ for } k = 11, \cdots, 54, m = 1, \cdots, 10, \\ x_{\text{real}m,ij} + u_{k,ij}, \text{ for } m = 1, 2, 3, k = 55, \cdots, 57, \\ z_{n,ij} + u_{k,ij}, \text{ for } k = 58, \cdots, 69, n = 1, \cdots, 12, \\ \sum_{m=1}^{10} \gamma_{m,i} x_{m,ij} + \sum_{m=54}^{57} \beta_m x_{m,ij} + \sum_{m=58}^{69} \phi_m x_{m,ij} + u_{k,ij}, \text{ for } k = 70, \\ \sum_{m=11}^{54} \beta_m x_{m,ij} + \lambda_k x_{70,ij} + u_{k,ij}, \text{ for } k = 71, \cdots, 80. \end{cases} \tag{20}$$

For model (20), we sampled exogenous nodes $u_{k,ij}$ from a standard normal distribution $\mathcal{N}(0, 1)$. Our objective centered on accurate sampling from the target distribution of $x_k$ for outcome indices $k = 71, \cdots, 80$ and cause indices $l = 11, \cdots, 54$. For the neural networks, we set the epochs to 300, batch size to 64, and learning rate to $10^{-4}$ where each neural network consisted of three hidden layers whose numbers of nodes were 128, 256, and 256 for the first, second and third layers, respectively. Other Settings were similar to the simulation study.

Table 2 presents the observational query evaluation metrics for PFD-BDCM and PFD-DCM across ten atmospheric pollutants including PM$_{2.5}$. The results demonstrate that PFD-BDCM achieves superior data generation capability on this dataset, making it suitable for interventional and counterfactual queries in atmospheric pollution research. For instance, the model can simulate the impact of policy interventions or anthropogenic factors on pollutant emissions, thereby providing robust data support for developing effective emission reduction strategies.

Furthermore, PFD-BDCM's flexible architecture and transparent spatiotemporal dynamic structure ensure strong interpretability and extensibility. The model can be readily adapted to datasets with similar structures in other domains such as healthcare and economics, demonstrating broad applicability beyond environmental science.

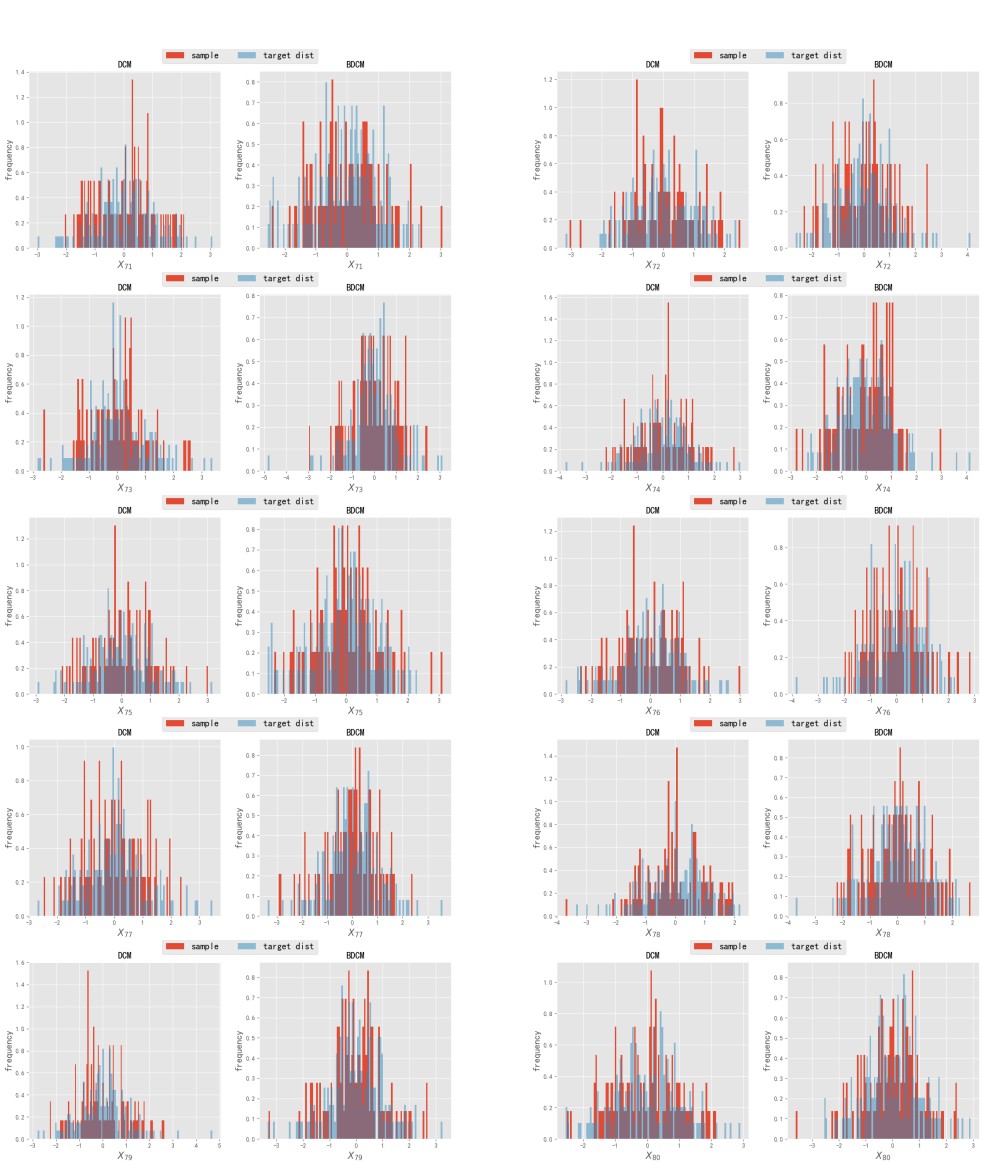

Figure 7: Empirical distributions of Pollutant emission indicators (From left to right, and from top to bottom: $SO_2$, $NO_x$, CO, $NH_3$, $PM_{10}$, $PM_{2.5}$, BC, OC, $CO_2$) sampled from PFD-DCM (left) and PFD-BDCM (right) compared to the ground-truth target distribution (observation query).

