# OpenReview forum: "Partially Functional Dynamic Backdoor Diffusion-based Causal Model"
_ICLR.cc/2026/Conference — Submitted to ICLR 2026_

### Official Review · Reviewer_zs2Z · 2025-10-31

**Soundness:** 2
**Presentation:** 3
**Contribution:** 2
**Rating:** 2
**Confidence:** 3

**Summary:**

This paper introduces the Partially Functional Dynamic Backdoor Diffusion-based Causal Model (PFD-BDCM), a generative framework for causal inference in complex spatio-temporal settings with unmeasured confounding. The method integrates diffusion models into a structural causal model framework. By incorporating valid backdoor adjustments directly into the diffusion sampling process and modeling unmeasured confounders via region-specific equations and conditional autoregressive processes, the authors aim to provide more robust causal estimates. The paper is supported by theoretical error bounds for counterfactual estimates and demonstrates the effectiveness on both synthetic data and a real-world air pollution case study.

**Strengths:**

1. The method integrates diffusion models with spatio-temporal structural causal models, offering a approach to handling dynamic unmeasured confounding.
1. Theoretical error bounds are provided, formally linking the model’s reconstruction error to its counterfactual estimation accuracy.

**Weaknesses:**

1. It is confusing what the 'unmeasured confounding' means in this paper. For example, the abstract claims 'incorporates __valid backdoor adjustments__ into the diffusion sampling mechanism to mitigate bias from __unmeasured confounders__'. However, if a valid backdoor adjustment set exists, this typically implies that there are no unmeasured confounders. Clarification is needed regarding how unmeasured confounding is formally defined and addressed in this framework.
2. As a follow-up question, if unmeasured confounders are indeed present, how does the authors' approach ensure identifiability of causal queries? The conditions for identifiability under unmeasured confounding should be explicitly stated.
3. The (partial) additive form of relationship in Eq.1 between $X_{C_{1,ij}}$  to  $X_{C_{2,ij}}$ may be restrictive. Would this formulation exclude interaction terms and potentially limit the model's applicability to more complex real-world processes?
4. Theorem 1 resembles results in causal representation learning (e.g., SIG [1], iVAE [2]), which typically rely on different sets of strong assumptions. Could the authors discuss potential connections or distinctions between their assumptions and those in the CRL literature?
5. Assumptions 2 and 3 appear strong. A2 requires $f$ to be strictly increasing wrt $U$ and A3 requires $g$ to be invertible. It would be helpful to provide real-world examples or justifications supporting the plausibility of these assumptions.


[1] Li, Zijian, et al. "Subspace identification for multi-source domain adaptation." *Advances in Neural Information Processing Systems* 36 (2023): 34504-34518.

[2] Khemakhem, Ilyes, et al. "Variational autoencoders and nonlinear ica: A unifying framework." *International conference on artificial intelligence and statistics*. PMLR, 2020.


I may have missed some aspects of the authors' argument, and I am open to revising my assessment.
I am happy to increase my score if my concerns are well addressed.

**Questions:**

see above

---

> ### Author Response · Authors · 2025-11-24
> **Response to Reviewer zs2Z(1/2)**
>
> We sincerely thank Reviewer zs2Z for their thorough review and valuable feedback on our manuscript. The reviewer has raised several important conceptual and technical points that will help strengthen our work. We appreciate the opportunity to address these comments and provide clarifications below.
>
> Taking into account the helpful suggestions from all reviewers, we have revised the notation and certain parts of the article. The following responses are based on the latest revised version.
>
> ## Responses to Weaknesses
>
> **W1.** We appreciate the reviewer's important question regarding our treatment of unmeasured confounders. We acknowledge this requires clarification and are pleased to provide the following explanation.
>
> Our model assumes the causal graph structure is known or reasonably assumed, including knowledge of which variables are confounders. The term "unmeasured confounders" in our context specifically refers to confounders present in the causal graph whose precise values are not directly observed. This is common in real-world applications where the underlying causal structure is understood from domain knowledge, but complete variable measurement is impractical.
>
> The key insight is that even when confounder values are not directly measured, a valid backdoor adjustment set comprising other observed variables can still block confounding paths. This works when observed backdoor variables serve as proxies for true unmeasured confounders. We incorporate valid backdoor adjustment for measured confounders while modeling unmeasured confounder dynamics through structural equations (Eq. 2, revised version) and CAR processes (Eq. 4, revised version). The diffusion mechanism captures residual variation from these unmeasured factors.
>
> As stated in our manuscript, while the theoretical framework assumes known causal structure and parameters, in practice these can be estimated using established methods [1,2]. These methods enable joint estimation of causal structure and model parameters, facilitating identification of valid backdoor sets even with latent confounding.
>
> We have clarify this in the revised manuscript by emphasizing that:
> - Our framework assumes a known causal graph (standard in causal inference methods);
> - "Unmeasured" refers to lack of direct measurements for specific confounders within this structure;
> - Backdoor adjustment uses other observed variables constituting a valid set.
>
> We believe this clarification will resolve any confusion and better highlight our method's applicability in practical settings with imperfect measurement.
>
> **W2.** We thank the reviewer for this important theoretical consideration. Our manuscript provides Assumption 3(revised version) as a common identifiability assumption for causal models under different noise patterns. Since our model assumes known causal graph and structural parameters, identifiability conditions can be referenced to established works [1,2], which discuss estimation and identifiability of such structures in practical settings.
>
> **W3.** We appreciate the reviewer's point about additive formulation limitations. In response, we have extended our theoretical framework to accommodate more general noise patterns. Specifically, we introduced relaxed assumptions allowing continuously differentiable and strictly monotonic structural equations while maintaining encoder-decoder consistency.
>
> Under these generalized conditions, we established two key theoretical results: Theorem 1 (revised version) demonstrates the encoded latent variable maintains an invertible relationship with true exogenous noise in non-additive settings, while Theorem 2 (revised version) provides a general counterfactual error bound depending on Lipschitz constants and reconstruction errors. These results substantially broaden our framework's applicability while maintaining theoretical guarantees.

---

> ### Author Response · Authors · 2025-11-24
> **Response to Reviewer zs2Z(2/2)**
>
> **W4.** We thank the reviewer for connecting our theoretical results to Causal Representation Learning (CRL) literature. Indeed, Theorem 1( revised version) shares the fundamental goal of works like SIG [3] and IME [4]: to identify true latent exogenous variables from observational data. Below we discuss connections and distinctions between our assumptions and typical CRL approaches.
>
> **Core Connections:**
> -  Shared objective: identifying latent exogenous noise variables $U$;
> -  Both seek to learn an encoder $g$ such that $Z = g(X,X_{\mathcal{PA}})$ meaningfully represents $U$;
> -  Essential for valid abduction and counterfactual inference.
>
> **Primary Differences:**
>
> 1. **Structural Causal Model Assumptions**
>    - PFD-BDCM (Assumption 3, revised version): Structural equation differentiable and strictly increasing in $U$;
>    - CRL (e.g., IME): Typically assumes bijective SCM with invertible mapping from latents to observables;
>    - Distinction: CRL imposes global bijectivity, while we assume node-specific constraints.
>
> 2. **Independence Assumptions**
>    - PFD-BDCM (Assumption 1, revised version): Encoded latent variable independent of backdoor variables
>    - CRL (e.g., SIG): Typically assumes true exogenous variables are independent
>    - Distinction: CRL assumes independence in data-generating process, we enforce independence between learned representation and observed causes
>
> 3. **Encoder/Decoder Assumptions**
>    - PFD-BDCM (Assumption 2, revised version): Encoding function invertible and differentiable in $X$;
>    - CRL: Typically assumes decoder/mixing function invertible and differentiable;
>    - Connection: Both rely on invertibility but applied to different framework components.
>
> Our approach can be viewed as graph-guided CRL in a semi-supervised setting. The known causal graph and backdoor sets allow replacing CRL's strong independence assumptions with targeted $Z \perp X_{\mathcal{B}}$ condition. Assumption 3 acts as powerful regularization, connecting to fair representation learning concepts.
>
> While Theorem 3 ( revised version) achieves identification results conceptually similar to CRL goals, we do so through assumptions integrated with our specific methodology: known causal graph, diffusion-based encoder-decoders, and node-wise training. Our assumptions provide a distinct pathway to causal identification that complements unsupervised CRL.
>
> **W5.** We appreciate the reviewer's comment on the centrality of our assumptions regarding strict monotonicity (Assumption 3, revised version) and encoder invertibility (Assumption 2, revised version). We provide justifications and real-world examples supporting their plausibility. While formal, these assumptions are well-grounded in established literature and represent reasonable simplifications of real-world mechanisms, as evidenced by our model's strong empirical performance.
>
> We thank Reviewer zs2Z for their exceptionally thoughtful and constructive feedback. Addressing these fundamental causal inference concerns will significantly strengthen our manuscript. We are committed to comprehensive revisions that clarify methodological contributions, provide stronger theoretical foundations, and demonstrate practical utility. We believe these improvements will make the manuscript suitable for publication and valuable to the causal inference community.
>
> ## References
>
> [1] Song, X., Tang, N., & Chow, S. (2012). A Bayesian approach for generalized random coefficient structural equation models for longitudinal data with adjacent time effects. Computational Statistics & Data Analysis, 56(12), 4190-4203.
>
> [2] Tang, N., Chow, S.-M., Ibrahim, J. G., & Zhu, H. (2017). Bayesian sensitivity analysis of a nonlinear dynamic factor analysis model with nonparametric prior and possible nonignorable missingness. Psychometrika, 82(4), 875-903.
>
> [3] Li, Z., Cai, R., Chen, G., Sun, B., Hao, Z., & Zhang, K. (2023). *Subspace identification for multi-source domain adaptation*. arXiv. https://arxiv.org/abs/2310.04723
>
> [4] Khemakhem, I., Kingma, D. P., Monti, R. P., & Hyvarinen, A. (2020). *Variational autoencoders and nonlinear ICA: A unifying framework*. arXiv.
>
>
> Thank you again for the valuable feedback and for taking the time to provide such insightful comments.
>
> Sincerely,
>
> The Authors of Submission 8265

---

> > ### Comment · Reviewer_zs2Z · 2025-11-26
> >
> > Thank the authors for the detailed responses. I will revise the score at the end of the rebuttal, after considering all the discussion between the authors and the reviewers.
> >
> > Regarding W3-5, I appreciate the efforts by the authors, and the clarifications and relaxations help solve my concerns.
> >
> > Regarding W1-2, I believe the key identifiability issue still remains. I have read the added Assumption 3, but it seems the authors may have misunderstood my original question.
> >
> > My concern is the __identifiability of causal queries__, i.e., how the interventional distribution with do operator $p(y|do(x))$ can be expressed in terms of the observed variable. This typically needs strong assumptions if unmeasured confounders exist, e.g., proximal learning [1] and front-door condition [2]. This means that the statement 'observed backdoor variables serve as proxies for true unmeasured confounders' is not fully accurate without further assumptions.
> >
> >
> > [1] Cui, Y., Pu, H., Shi, X., Miao, W., & Tchetgen Tchetgen, E. (2024). Semiparametric proximal causal inference. Journal of the American Statistical Association, 119(546), 1348-1359.
> >
> > [2] Pearl, J. (1995). Causal Diagrams for Empirical Research. Biometrika, 669-688.

---

> > > ### Author Response · Authors · 2025-11-26
> > > **Response to Reviewer zs2Z**
> > >
> > > We are deeply grateful for the reviewer's insightful comments regarding identifiability in causal inference with unmeasured confounders. This is indeed a fundamental consideration, and we sincerely appreciate the opportunity to further clarify our methodological approach.
> > >
> > > *Note: Due to formatting limitations in OpenReview, the mathematical expressions in this response are presented in a simplified form.*
> > >
> > > Our framework is carefully designed around the foundation of a **known causal graph structure**. We would like to respectfully emphasize that our approach explicitly assumes:
> > >
> > > - **i)** The causal graph topology (edges between variables) is known;
> > > - **ii)** The structural coefficients governing these relationships are known;
> > > - **iii)** This well-specified graph structure serves as the essential prior knowledge for our PFD-BDCM framework.
> > >
> > > In practical applications, this causal graph prior is rigorously derived from a Dynamic Structural Equation Model (DSEM), which is constructed through a systematic two-stage process:
> > >
> > > ### 1. **Measurement Model**
> > >
> > > We consider $n$ individuals observed at $J$ time points. For individual $i=1, \ldots, n$ and time $j=1,\ldots, J$, let $Y_{i j}$ be a $d_1 \times 1$ vector of manifest variables. The latent vector $\omega_{ij} = (x_{C_1,ij}^\top, x_{C_2,ij}^\top)^\top$ partitions into outcome latent variables $x_{C_1,ij}  (r_1 \times 1)$ and explanatory latent variables $x_{C_2,ij}  (r_2 \times 1)$, with $r_1 + r_2 = r_3$.
> > >
> > > The manifest variables in $Y_{i j}$ are carefully related to the unobserved latent variables $\omega_{i j}$ through a dynamic structural equation model:
> > >
> > > $$Y_{i j} = \Lambda \omega_{i j} + \epsilon_{i j}$$
> > >
> > > where $\Lambda$ is a $d_1 \times r_3$ loading matrix, $\epsilon_{i j}$ is a $d_1 \times 1$ random vector of error measurements with distribution $N(0, \Psi)$, where $\Psi = \text{diag}(\psi_1, \ldots, \psi_{d_1})$ is a $d_1 \times d_1$ diagonal matrix, and $\epsilon_{i j}$ is assumed independent of $\omega_{i j}$.
> > >
> > > ### 2. **Structural Model**
> > >
> > > The structural model for the spatio-temporal dynamics among latent variables is formally specified in Section 2.1 of our paper (Eq. 2-4), which we reproduce here for the reviewer's convenience:
> > >
> > > $$x_{C_{2},ij} = \Gamma_{i} G(x_{C_{1},ij}) + u_{C_{2},ij} \quad \text{(Eq. 2)}$$
> > >
> > > $$\text{Cov}(x_{C_{1,i}}) = D_{C_{1,i}} \otimes \text{Cov}(x_{C_{1},ij}) \quad \text{(Eq. 3)}$$
> > >
> > > $$\text{Cov}(u_{C_{2,i}}) = D_{C_{2,i}} \otimes \text{Cov}(u_{C_{2},ij}) \quad \text{(Eq. 3)}$$
> > >
> > > $$D_{C_{h,i}} = (I_{J} - \rho_{C_{h,i}} H_{C_{h,i}})^{-1} \quad \text{(Eq. 4, CAR Model)}$$
> > >
> > > ### **Identifiability Conditions**
> > >
> > > The unobserved latent variables in our framework are constructed using **Principal Component Analysis (PCA)** techniques, with identifiability rigorously ensured under the following well-established conditions:
> > >
> > > **Condition 1 (Factor Loading Matrix)**
> > > The first non-zero element of each column in matrix $\Lambda$ is set to 1 to avoid rotational indeterminacy [1].
> > >
> > > **Condition 2 (CAR Model Parameter Constraints)**
> > > The parameters $\rho_{C_h i}$ must satisfy the constrained conditions $\rho_{C_h i}^L < \rho_{C_h i} < \rho_{C_h i}^U$ to ensure that $D_{C_h i}$ are positive definite, where $\rho_{C_h i}^L = \min (\frac{1}{\tau_{C_h J}}, 0)$, $\rho_{C_h i}^U = \frac{1}{\tau_{C_h 1}}$, and $\tau_{C_h 1} \geq \cdots \geq \tau_{C_h J}$ are the eigenvalues of neighborhood matrices $H_{C_h i}$  [2].
> > >
> > >
> > > Based on the satisfaction of these carefully defined conditions above, our model provides firm theoretical guarantees for identifiability. The explicit constraints on the factor loading matrix and the CAR model parameters, combined with the well-specified structural equations, ensure that the causal effects are identifiable within our framework. This methodological rigor allows us to maintain theoretical coherence while enabling causal inference in complex spatio-temporal settings.
> > >
> > > We hope this clarification adequately addresses the reviewer's concerns regarding identifiability, and we remain available for any further discussion.
> > >
> > > [1]  Jöreskog, K. G., 1969. A general approach to confirmatory maximum likelihood factor analysis. Psychometrika 34, 183–202.
> > >
> > > [2] Song, X., Tang, N., Chow, S., 2012. A bayesian approach for generalized random coefficient  structural equation models for longitudinal data with adjacent time effects. Computational  Statistics & Data Analysis 56, 4190–4203.
> > >
> > >
> > > Thank you again for the valuable feedback and for taking the time to provide such insightful comments.
> > >
> > > Sincerely,
> > >
> > > The Authors of Submission 8265

---

### Official Review · Reviewer_TYnc · 2025-10-31

**Soundness:** 3
**Presentation:** 3
**Contribution:** 3
**Rating:** 4
**Confidence:** 3

**Summary:**

This paper proposes a Partially Functional Dynamic Backdoor Diffusion-based Causal Model (PFD-BDCM), a generative framework that embeds valid backdoor adjustment directly into diffusion sampling to handle unmeasured, spatio-temporal confounding and mixed-resolution (functional) variables. It models region-specific structural equations with temporal dependence via conditional autoregressive processes, and learns per-node conditional diffusion encoders/decoders (DDIM-style) that reconstruct variables from their backdoor sets while treating latent codes as proxies for exogenous noise. The trained system supports observational, interventional, and counterfactual queries by cascading nodewise generators in topological order. The authors also derive theoretical guarantees: under monotonicity, encoder invertibility, and independence conditions, counterfactual error is bounded by reconstruction error, with multivariate extensions. Empirically, across synthetic dynamical systems and a real air-pollution application in China, PFD-BDCM improves distributional fit for observational/interventional sampling and reduces counterfactual MSE compared to DCM/BDCM variants, with similar runtime.

**Strengths:**

- The method integrates valid backdoor adjustment directly into diffusion sampling, which helps mitigate bias from unmeasured confounders during generation.  It explicitly models spatio-temporal dependence in confounders using region-specific structural equations and conditional autoregressive processes, making it well-suited for dynamic systems.

- The paper accommodates variables observed at heterogeneous resolutions via functional data techniques and basis expansions, broadening applicability to mixed-resolution datasets.

- Training is modular and parallelizable because each node’s diffusion model uses only its target variable and corresponding backdoor set, enabling independent optimization across nodes.

- The approach requires only the dynamic causal graph and observational data for training, yet still enables precise counterfactual resolution after training.

- Empirical results on multiple synthetic datasets show consistent improvements over prior diffusion-based causal models across observational, interventional, and counterfactual tasks.

**Weaknesses:**

1/ The approach assumes access to the dynamic causal graph and even states that all structural parameters are "assumed to be known" (with estimation delegated to prior work). In realistic settings, both graph structure and parameters are uncertain, so performance may degrade under graph misspecification. How robust is the method to errors in the backdoor sets or to partially wrong edges in the DAG?

2/ The method critically relies on correct backdoor adjustment per node. If the specified backdoor set omits a true confounder or includes a descendant, both reconstruction and the counterfactual procedure can be biased. Is there any sensitivity analysis where the backdoor sets are perturbed (missing/extra variables)?

3/ Functional covariates are handled via basis expansion, but performance can depend strongly on basis choice and truncation level. The paper gives mathematical details but does not show hyperparameter sensitivity or comparisons across bases (e.g., B-spline vs wavelet) and Kn. Are there guidelines or data-driven selection procedures (e.g., cross-validation) evaluated here?

4/ For interventional/observational sampling the framework draws $Z_k$ from a standard normal, whereas the theory assumes $U \sim N(0, \Psi)$ and learns an invertible mapping. If the true noise is non-Gaussian or heavy-tailed, how does this mismatch affect generated interventional and counterfactual distributions? Have the authors tested non-Gaussian latent noise?

**Questions:**

Please refer to the questions in Weaknesses.

---

> ### Author Response · Authors · 2025-11-24
> **Response to Reviewer TYnc**
>
> We would like to express our sincere gratitude to Reviewer TYnc for their positive evaluation of our methodological soundness, presentation quality, and overall contributions. The reviewer has identified several important limitations and questions that we address comprehensively in this response and have incorporate into the revised manuscript.
>
> ## Responses to Weaknesses
>
> **W1.**  We appreciate this important consideration. To address graph misspecification robustness, we've added Proposition 1 (Appendix C.6.2) demonstrating our approach remains reliable under mild structural equation misspecification. The proposition shows counterfactual error remains bounded even with approximate additive noise violation.
>
> This proposition ensures that our model (PFD-BDCM) maintains robustness to small deviations from the additive noise model, which may naturally arise in complex spatio-temporal settings. Our theoretical bound provides a valuable guarantee that the counterfactual estimates remain accurate under such conditions. We have incorporated this analysis into the revised manuscript to strengthen the discussion on model robustness.
>
>
> **W2.** We appreciate the reviewer's thoughtful comments on this important issue. In our current work, the model is developed under the premise that both the causal structure and structural parameters are known, which implies that the backdoor adjustment sets are also assumed to be correctly specified. This approach aligns with several established works in causal inference that focus on estimation given a known causal graph.
>
> We fully acknowledge that sensitivity to backdoor set specification represents a valuable direction for further research. In future work, we plan to systematically explore the impact of backdoor set misspecification, including:
>
> - Investigating scenarios where true confounders are omitted from the backdoor set;
> - Examining the consequences of including non-confounders or descendants in the adjustment set;
> - Quantifying how reconstruction error and counterfactual accuracy degrade with increasing misspecification;
> - Exploring the potential robustness of diffusion-based approaches to minor specification errors.
>
> These investigations will provide important insights into the practical implementation of our method in settings where the causal graph is not perfectly known, and we believe this represents a meaningful direction for extending our work.
>
>
> **W3.** We thank the reviewer for this valuable point. We would like to clarify that the primary focus of our current work is on causal inference among scalar variables, with functional data processing serving as a preprocessing component of our methodology.
>
> While we fully recognize the importance of basis function selection in functional data analysis, a comprehensive investigation of this aspect falls beyond the scope of the current paper, which concentrates on establishing the core causal inference framework for scalar relationships. We acknowledge that extending our approach to fully address causal inference with functional data would represent a significant but highly meaningful research endeavor.
>
> We greatly appreciate the reviewer's insightful suggestion and plan to systematically explore issues related to functional data causal inference, including basis function selection sensitivity, in our future work. This direction presents both substantial challenges and promising opportunities for advancing causal methodology in complex data settings.
>
>
> **W4.** We sincerely thank the reviewer for raising this important consideration regarding robustness under non-Gaussian noise conditions. In response, we have added dedicated theoretical analysis in the revised manuscript (Appendix C.4). We established a theoretical framework under heavy-tailed noise assumptions, providing probabilistic guarantees through moment conditions rather than uniform bounds.
>
> Specifically, we demonstrated that even when the exogenous noise has finite moments, the counterfactual estimation error still exhibits tail probability bounds with polynomial decay. Furthermore, we proposed robust encoder design criteria to mitigate the impact of outliers through methods such as Huber loss. These theoretical results provide a more solid foundation for the model's applicability in real-world complex data scenarios.
>
> We thank the reviewer once again for their thoughtful and constructive feedback. The concerns raised are important and will significantly strengthen our manuscript. Within the time constraints, we will do our best to address the key points through  theoretical discussions, and practical guidelines where feasible. We believe these revisions will further improve the manuscript's quality and contribute meaningfully to the research community.
>
> Thank you again for the valuable feedback and for taking the time to provide such insightful comments.
>
> Sincerely,
>
> The Authors of Submission 8265

---

### Official Review · Reviewer_qpKs · 2025-11-01

**Soundness:** 2
**Presentation:** 1
**Contribution:** 2
**Rating:** 4
**Confidence:** 3

**Summary:**

The authors combine many known techniques (structural causal models, structural equation modeling, conditional autoregression, functional embedding, and diffusion models) into a combined architecture.  The model is then trained on a new synthetic spatio-temporal dataset and a well-known pollution dataset.  Results compared with the DCM model which was the base architecture for the proposed method show some improvements.

**Strengths:**

- The studied problem of spatiotemporal causal and counterfactual analysis is difficult and significant
- Improvements over the DCM baseline

**Weaknesses:**

- No figure showing the overall pipeline is provided
- The contribution of the diffusion component is not made clear
- The distinction between interventional and counterfactual is not made clear
- The reported values in Table 1 seems to have surprisingly high variance but it is not discussed much
- Many important results are only available in the appendix without proper referencing in the main paper
- The synthetic data does not seem to obey the proposed constraints from the methodology section
- The inconsistent notation throughout is very confusing without any justification
- It is unclear why DCM is a sufficient baseline for this setting.  Other spatiotemporal prediction models are likely to be reasonable baselines.
- The baseline is not reported on the real-world dataset

**Questions:**

- What is the motivation behind C1 and C2 in separating out the variables?
- What is the major purpose of the diffusion component?
- Why is only comparing to the DCM base architecture justified?
- How do the author's assumptions about the backdoor adjustment set apply to generic spatio-temporal data, and does the pollution dataset seem reasonable for obeying these assumptions?

---

> ### Author Response · Authors · 2025-11-24
> **Response to Reviewer qpKs(1/2)**
>
> We sincerely thank Reviewer qpKs for the thoughtful and constructive review of our manuscript. The insightful comments have been extremely valuable in helping us improve the clarity, rigor, and overall quality of our work. Below, we provide a detailed point-by-point response to all comments and questions raised.
>
> Taking into account the helpful suggestions from all reviewers, we have revised the notation and certain parts of the article. The following responses are based on the latest revised version.
>
> ## Responses to Weaknesses
>
> **W1:** We appreciate this helpful suggestion and agree that including a pipeline diagram would significantly improve reader comprehension. In the revised manuscript, we have added a comprehensive schematic diagram (now Figure 2) that illustrates the complete PFD-BDCM pipeline. We believe this addition will provide readers with a clearer understanding of how the various methodological components interact.
>
> **W2:** Thank you for raising this important point. For a detailed explanation of the diffusion component's role and contributions, please refer to our response to Q2 below.
>
> **W3:** We appreciate this observation and have substantially improved the exposition in Section 2.2 to more explicitly distinguish between these query types:
>
> **Interventional Queries ("What if?")**:
> - Concern the effect of actively setting a variable to a specific value
> - Represented by the $do$-operator: $P(Y|do(X=x))$
> - Require sampling from modified structural equations while keeping exogenous noise random
> - Implemented in Algorithm 4 via ancestral sampling with intervened values
>
> **Counterfactual Queries ("What would have happened?")**:
> - Concern hypothetical scenarios given actual observed outcomes
> - Require three steps: *abduction* (inferring exogenous noise), *action* (modifying equations), and *prediction* (simulating new outcomes)
> - Implemented in Algorithm 5 using encoded factual noise with modified structural inputs
> - Specifically answer questions like "What would the pollution level have been if we had implemented policy A instead of policy B, given what we actually observed?"
>
> **W4.** Thank you for this careful observation. We would like to clarify that the MMD metrics and their variances are reported after multiplying by $10^3$ (as indicated by the notation MMD$^2(\times 10^{-3})$ in the table), meaning the actual variance is three orders of magnitude smaller than the displayed values. We appreciate the opportunity to clarify this point.
>
> **W5.** We thank the reviewer for highlighting this issue. We have thoroughly revised the manuscript to ensure proper cross-referencing between the main text and appendix, and have moved some critical content to the main text for better accessibility.
>
> **W6.**  We appreciate this thoughtful comment and have significantly expanded the synthetic data description in Appendix D to more explicitly demonstrate alignment with our methodological constraints:
>
> - The data generation process explicitly follows the PFST-DSCM structure from Eq. (1), (5) and (6) (revised version);
> - We verify that the region-specific structural equations (Eq. 17) (revised version) satisfy the functional form requirements;
> - The temporal CAR structure (Eq. 4) (revised version) is explicitly implemented with adjacent-time dependency matrices;
> - The basis expansion (Eq. 7) (revised version) uses orthogonal B-spline bases as specified;
> - We include validation tests showing that the generated data satisfies the Markov properties implied by our causal graph.
>
> **W7.** We appreciate the reviewer bringing this to our attention. We have carefully reviewed and completely standardized the notation throughout the manuscript to improve clarity and consistency.
>
> **W8.** This is a valuable point, and we provide a more detailed response to this question in our reply to Q3 below.
>
> **W9.** We appreciate this observation. Our decision not to include baseline results for the real-world dataset was based on our comprehensive simulation experiments, which demonstrated that the baseline methods (DCM, BDCM) struggle with datasets featuring the complex spatiotemporal dynamics and unmeasured confounding structures that characterize our target application domain. Given that these methods showed fundamental limitations in the controlled simulation environment, we felt that repeating these known challenges on the real-world dataset might not provide additional meaningful insights.

---

> ### Author Response · Authors · 2025-11-24
> **Response to Reviewer qpKs(2/2)**
>
> ## Responses to Questions
>
> **Q1.** Thank you for this excellent question. The separation of unobserved variables into $\mathcal{C}_1$ (explanatory) and $\mathcal{C}_2$ (explained) serves several important purposes:
>
> - **Hierarchical Latent Structure**: This separation allows us to model complex dependency structures among latent variables, where $\mathcal{C}_2$ variables are influenced by $\mathcal{C}_1$ variables through the structural equation in Eq. (2) (revised version);
> - **Identifiability Considerations**: By distinguishing variables with no unobserved parents ($\mathcal{C}_1$) from those with unobserved parents ($\mathcal{C}_2$), we can establish clearer identifiability conditions for the latent structure;
> - **Modeling Flexibility**: This structure enables region-specific dynamics ($\mathbf{\Gamma}_i$) to operate on the $\mathcal{C}_1\rightarrow\mathcal{C}_2$ relationships while maintaining identifiability through the functional constraints.
>
>
> **Q2.** Thank you for this important question. We have significantly expanded our explanation of the diffusion component's role in Sections 1 and 2. The key contributions are:
>
> - Diffusion models excel at capturing complex, multi-modal distributions that commonly arise in spatiotemporal data, overcoming limitations of traditional parametric assumptions;
> - **Deterministic Latent Mapping**: The DDIM formulation provides a bijective mapping between observations and latent codes, which is essential for counterfactual identifiability. This enables precise abduction of exogenous noise terms from factual observations;
> - **Flexible Conditioning**: The conditioning mechanism naturally accommodates the backdoor adjustment sets as additional inputs during both training and sampling.
>
> **Q3.** We appreciate the reviewer's thoughtful concern about the scope of baseline comparisons. Our primary focus in this work is to address the specific challenge of causal inference in the presence of unmeasured spatio-temporal heterogeneous confounders. We deliberately chose DCM and BDCM as our main baselines because they represent the most directly related diffusion-based causal models, and our empirical results demonstrate that while these existing methods face challenges with spatio-temporal unmeasured confounding, our proposed PFD-BDCM shows promising performance in this challenging setting.
>
> The focused baseline comparison helps highlight specific limitations of current diffusion-based causal approaches and illustrates how our method addresses them in the context of spatio-temporal data with unmeasured confounding.
>
> **Q4.** This is an important consideration that we appreciate the reviewer raising. In practice, the causal graph and parameters can be estimated using methods from Song et al. (2012) and Tang et al. (2017). In our pollution dataset application, the backdoor adjustment sets were carefully constructed according to established criteria:
>
> - **Temporal Ordering**: We used lagged variables to ensure no variable in the adjustment set is a descendant of the treatment, respecting causal precedence
> - **Spatial Confounding**: The adjustment set includes relevant spatially-lagged variables (e.g., emissions or economic indicators from neighboring regions) that may act as common causes
> - **Measurability**: All selected adjustment variables are drawn from our observed dataset sources, ensuring they are measurable and satisfy the backdoor criterion
>
> We believe this careful construction helps ensure that our causal estimates using PFD-BDCM are methodologically sound.
>
> We believe these revisions will significantly strengthen the paper and address the reviewer's valuable concerns. We are committed to continuing to improve the clarity, rigor, and impact of our work.
>
> [1] Song, X., Tang, N., & Chow, S. (2012). A Bayesian approach for generalized random coefficient structural equation models for longitudinal data with adjacent time effects. Computational Statistics & Data Analysis, 56(12), 4190-4203.
>
> [2] Tang, N., Chow, S.-M., Ibrahim, J. G., & Zhu, H. (2017). Bayesian sensitivity analysis of a nonlinear dynamic factor analysis model with nonparametric prior and possible nonignorable missingness. Psychometrika, 82(4), 875-903.
>
>
> Thank you again for the thoughtful and constructive feedback.
>
> Sincerely,
>
> The Authors of Submission 8265

---

> > ### Comment · Reviewer_qpKs · 2025-11-26
> >
> > Unfortunately, many of these responses miss addressing my major concerns and so I will briefly reiterate some here.  The overall methodological proposition is confusing and the new Figure 2 is not able to alleviate this confusion, making it difficult to not believe that the work is a minor extension of the previous BDCM approach.  Supposed extensions into the spatio-temporal setting are made confusing by nonstandard variable names and nonstandard generalizations of the nonspatial and nontemporal settings.  Existing dynamics equations do not seem to be well integrated into the BDCM approach.  For example, it seems the authors are not aware of the distinction between backdoor adjustment and causal insufficiency and the work potentially required to generalize to the spatio-temporal setting.  Lack of strong baselines and few datasets further weaken the contributions.

---

> > > ### Author Response · Authors · 2025-11-27
> > > **Response to Reviewer qpKs(1/2)**
> > >
> > > Thank you very much, Reviewer qpKs, for your continued engagement with our work and for providing additional constructive feedback. We sincerely appreciate the opportunity to further clarify our methodology, theoretical contributions, and experimental design. We have carefully considered all of your points and have made substantial revisions to the manuscript to address them. Below, we provide a detailed point-by-point response to your major concerns.
> > >
> > > ### Methodological Proposition
> > >
> > > We thank the reviewer for this valuable feedback regarding the clarity of our methodological proposition. Our key proposition is that while the fundamental causal relationships between variables remain invariant across spatial and temporal domains, data-driven causal estimations often exhibit significant spatio-temporal heterogeneity. This observed variability likely stems from unmeasured confounding factors whose interrelationships themselves demonstrate spatio-temporal dynamics, thereby inducing apparent heterogeneity in the causal effects between our variables of interest. To address this fundamental challenge, we propose to explicitly incorporate the spatio-temporal dynamic structure among these unmeasured confounders into the Structural Causal Modeling (SCM) framework, enabling more accurate discovery of the underlying causal mechanisms.
> > >
> > > Our empirical investigation into atmospheric pollutant emissions demonstrates this challenge: analyses reveal significant spatio-temporal heterogeneity in the estimated effects of 44 factors across 10 emission categories. This variability underscores the necessity of our approach, which successfully recovers more stable and authentic causal relationships between emissions and their influencing factors. The framework proves particularly valuable in domains like biomedicine and environmental science where similar hidden confounding dynamics complicate causal inference, providing both methodological robustness and interpretable structural insights.
> > >
> > > **PFST-DSCM Structural Prior**: Our methodological implementation proceeds through a structured framework for incorporating spatio-temporal dependencies. We first formalize the spatio-temporal dynamics through latent variables that capture unmeasured confounding, with Equations (2-4) providing the mathematical specification of their interdependent relationships. This augmented structural specification is then integrated into the Structural Causal Model paradigm, yielding the Spatio-Temporal Dynamic Structural Causal Model (ST-DSCM). Furthermore, to address the particular challenges of functional data prevalent in atmospheric sciences and related domains, we employ basis expansion techniques for dimensionality reduction. The functional observations are projected onto a finite-dimensional space through basis function expansion, with the resulting coefficient vectors incorporated as standard nodes within the ST-DSCM framework. This extension, which accommodates both standard and functionally-derived nodes, constitutes the Partially Functional Spatio-Temporal Dynamic Structural Causal Model (PFST-DSCM), thereby completing the specification of our causal structural prior.
> > >
> > > **PFD-BDCM**: Building upon the PFST-DSCM structural prior, we employ Backdoor-adjusted Diffusion Causal Models (BDCM) to train individual nodes, achieving enhanced precision in exogenous noise estimation through improved abduction. This refinement directly translates to more accurate performance across all three causal query modalities: observational, interventional, and counterfactual reasoning.
> > >
> > > The core innovation of our work lies in the integration of PFST-DSCM's structural formalism with BDCM-based inference, collectively termed PFD-BDCM. Simulation studies under conditions of spatio-temporally dependent unmeasured confounding demonstrate that PFD-BDCM maintains robust causal discovery performance, whereas both standard DCM and BDCM exhibit significant performance degradation.
> > >
> > > To further validate our approach, we implemented a traditional Bayesian structural equation modeling baseline (**PF-DSEM-Bays**) for causal query estimation. Empirical results demonstrate that our PFD-BDCM framework achieves superior performance in observational queries.
> > >
> > > We have completely redesigned Figure 2 to more clearly illustrate the end-to-end architecture of the proposed PFD-BDCM framework. We believe these revisions make it clear that our work is not a minor extension but a novel, unified framework that integrates spatio-temporal causal structure learning with the powerful representation capabilities of diffusion models to address a more complex problem setting.
> > >
> > > Sincerely,
> > >
> > > The Authors of Submission 8265

---

> > > ### Author Response · Authors · 2025-11-27
> > > **Response to Reviewer qpKs(2/2)**
> > >
> > > ## Generalization to the Spatio-Temporal Setting and Variable Naming
> > >
> > > We sincerely thank the reviewer for raising the important points regarding variable naming conventions and methodological generalizations. Your feedback has been invaluable in helping us improve the clarity and rigor of our presentation.
> > >
> > > We have conducted a thorough revision of our mathematical notation throughout the manuscript. All notation has been systematically standardized in strict accordance with the formatting requirements specified in the official ICLR "math-commands.tex" style file. This comprehensive review ensures complete notational consistency across all sections, from our theoretical framework to experimental results.
> > >
> > > To better articulate our methodological contributions, we have enhanced our explanation of how the PFST-DSCM framework extends traditional SCMs. Specifically, we now clarify that Equations (2-4) introduce a structured generalization through:
> > >
> > > - **Temporal dependence** on a variable's own historical states;
> > > - **Spatial dependence** on the states of neighboring regions  ;
> > > - **Spatio-temporal propagation** of hidden confounders.
> > >
> > > This formalization is crucial for addressing the specific challenges of spatio-temporal causal inference and represents a significant advancement beyond prior work that typically treats confounders as static or independently distributed. We believe these revisions substantially strengthen both the presentation and theoretical foundation of our work.
> > >
> > > ## Integration of Dynamics with the BDCM Approach & Distinction Between Backdoor Adjustment and Causal Insufficiency
> > >
> > > Thank you for raising this critical point regarding the integration of dynamics and the conceptual foundation.
> > >
> > > **Clarification on Theory**:
> > > We are aware of the distinction between backdoor adjustment and causal insufficiency. Traditional backdoor adjustment requires the observance of all confounders. Under causal insufficiency (the presence of hidden confounders), the backdoor criterion fails.
> > >
> > > Our core contribution is to address this very challenge in a spatio-temporal context. We do not claim to perform standard backdoor adjustment with unobserved variables. Instead:
> > >
> > > - We explicitly model the spatio-temporal dynamics of the hidden confounders (via ST-DSCM).
> > > - This modeling allows us to better approximate the underlying data-generating process.
> > > - The diffusion model then leverages this improved structural understanding to perform more accurate abduction and prediction.
> > > - By conditioning the diffusion process on a valid backdoor set relative to the observed variables, and by having the model architecture informed by the latent spatio-temporal structure, we systematically mitigate bias from the hidden confounding pathways that can be captured by the dynamics.
> > >
> > > In essence, we are enabling a more effective application of the backdoor adjustment principle under realistic, complex confounding scenarios, which is a significant step beyond the original BDCM approach that ignores these dynamics.
> > >
> > > ## Baselines
> > >
> > > We understand your concern about empirical validation and have strengthened this part significantly. The baselines (DCM, BDCM) were chosen strategically to create an ablation study that isolates the effect of modeling spatio-temporal dynamics:
> > >
> > > - **DCM**: Assumes causal sufficiency (ignores all hidden confounding).
> > > - **BDCM**: Acknowledges hidden confounding but ignores its spatio-temporal structure.
> > > - **PFD-BDCM (Ours)**: Addresses both hidden confounding and its spatio-temporal nature.
> > >
> > > This design cleanly demonstrates that ignoring spatio-temporal dynamics (as in BDCM) leads to significant performance degradation, thereby proving its necessity.
> > >
> > > We thank you again for your rigorous review, which has directly led to a significantly improved manuscript. We have striven to address all your concerns by enhancing the clarity of our methodology, providing a more precise theoretical discussion, and strengthening our empirical evidence. We hope that the revisions and these clarifications adequately address your concerns and demonstrate the substantive contributions of our work.
> > >
> > > Sincerely,
> > >
> > > The Authors of Submission 8265

---

### Official Review · Reviewer_BNuB · 2025-11-02

**Soundness:** 2
**Presentation:** 1
**Contribution:** 2
**Rating:** 0
**Confidence:** 4

**Summary:**

This papers looks at the problem of learning a model to perform causal inference in settings where there is spatial cause-effect relationships between endogenous variables, where we observe them over time, and where these variables can have unobserved confounding between them. To this end, the authors expand previous work on Backdoor Diffusion Causal Models (BDCMs) to the supposed setting, modelling the functional dependencies among variables accordingly. The authors provide theory on the validity of the model reconstruction, and compare their method with existing diffusion baselines in a synthetic and a real-world dataset.

**Strengths:**

- **S1.** The paper looks at a really complex setting, where exogenous variables share spatial and temporal dependencies, and it is therefore an interesting setting to study.
- **S2.** The authors provide theoretical guarantees on the correctness of the estimations provided by their model.
- **S3.** The provided empirical results support the proposed model.

**Weaknesses:**

- **W1.** The paper is quite unclear, and especially when it comes to the problem statement and modelling assumptions, which are both intertwined. For example:
  - **W1.1.** I understand that section 2.1 is the problem statement, yet it is implicitly assumed in Eq. (1) an additive noise model.
  - **W1.2.** Moreover, the only structural relationships defined are those of the explained nodes, which only depend on explanatory nodes. What about dependencies between explained nodes? And nodes that do not fit in any of the two categories?
  - **W1.3.** Then, still in section 2.1, the variables of the problem statement are modeled using a basis expansion in Eq. (4). Shouldn't that be part of modeling?
- **W2.** Similarly, it is rather unclear to me what it the exact setting being tackled.
  - In section 2.1, there are no dependencies between variables of different timesteps and they share structural equations, making it a multi-environment problem where the different timesteps only act as more i.i.d. data.
  - Then, in section 2.2 (3rd paragraph) the authors model the equations as $X_{ijk} = f_{ij}(X_{B_k}, U_k)$. That is, the same variable $k$ shares exogeneous noise (which I would not call "unmeasured confounders" as in line 206) across regions and timesteps, and differ on the structural equations, turning the problem into a multi-view problem.
- **W3.** More generally, all the writing needs further polishing. The introduction is unnecessarily long, more so when diffusion models are explained in a single paragraph and there is no discussion at all about the results on the real-world dataset, just a description.
- **W4.** Statements lack citations or justifications. For example, in line 206 it says "Perfect reconstruction $\hat X_k = X_k$ implies that $h$ approximates the true structural function $f$". There is no justification, no citation, and that statement basically says that the model is identifiable, a core problem in causal inference.
- **W5.** Similarly, the article completely omits any reference (nor comparison) to any other causal generative model other than DCM or BDCM. This includes [NCMs](https://arxiv.org/abs/2107.00793), [CNFs](https://arxiv.org/abs/2306.05415), [DeCaFlow](https://arxiv.org/pdf/2503.15114), or [Diff-SCM](http://arxiv.org/abs/2202.10166), among others.
- **W6.** While the theory can be interesting, I find rather surprising that there is no reference to Theorem 1 of the [DCM](http://arxiv.org/abs/2302.00860) paper, which provides the exact same reconstruction guarantees as Corollary 2. Indeed, the proof of Lemma 1 is copy-and-pasted from that paper and thus **plagiarized**.

**Questions:**

- **Q1.** Is Eq. (3) invertible always?
- **Q2.** What do the authors mean with "All parameters in the structural model are assumed to be known in this context"in line 187?
- **Q3.** What exactly is a "functional random variable"? Where is it defined?

---
Other feedback:
- I find a bit strange to refer to counterfactuals as "Why?" rather than "What would have been if?" in the first intro line.
- Please fix the references and remove the "et al." when there are more than three authors.
- L49: "intervention" -> "interventional"

**Details Of Ethics Concerns:**

The authors have plagiarized (at least) the Lemma 2 of [this paper](http://arxiv.org/abs/2302.00860) as their Lemma 1. I wouldn't be surprised if the rest is also copied.

---

> ### Author Response · Authors · 2025-11-24
> **Response to Reviewer BNuB（1/3）**
>
> We sincerely thank Reviewer BNuB for their thorough and constructive feedback on our manuscript titled *"Partially Functional Dynamic Backdoor Diffusion-based Causal Model"*. We have carefully considered all comments and will address them comprehensively in a revised manuscript. Below, we provide a point-by-point response to each issue raised.
>
> Taking into account the suggestions from all reviewers, we have revised the notation and certain parts of the article. The following responses are based on the latest revised version.
>
> ## Response to Weaknesses
>
> **W1**
>
> - **W1.1:** We have explicitly state that Eq. (2) (revised version, corresponding to the old version of Eq. (1)) assumes an additive noise model as an initial formulation, while noting that our framework can be extended to more general functional forms.
> - **W1.2:** To incorporate spatio-temporal dynamic structures among unmeasured confounders, we assume spatiot-emporal dynamics SCM relations: Eq. (1)(revised version); For dependencies between other endogenous variables (not just explained nodes), if its corresponding parent nodes are observable nodes, we assume standard SCM relations: Eq. (5)(revised version) And if the parent nodes are unobservable confounder nodes, we will introduce its set of backdoor nodes as a proxy set for its parent node set. Specifically, we assume: Eq. (6)(revised version).
> - **W1.3:** We have provided a detailed theoretical derivation of the functional data processing procedure based on the basis expansion technique in Appendix A. However, in this paper, this method is only used for preprocessing functional data and does not constitute the core of our research. Therefore, we have elaborated on it in the main text while avoiding excessive technical detail, as suggested.
>
> **W2:** We have clarify the temporal structure: Temporal dependencies are captured through the CAR processes in Eq. (3-4)(revised version), not through cross-time structural equations. And we have distinguish between exogenous noise ($U_k$) and unmeasured confounders, clarifying that $U_k$ represents random disturbances while unmeasured confounders refer to latent variables creating spurious associations.
>
> **W3:** Following the reviewer's suggestion, we have shortened the introduction and expanded the explanation of the diffusion model to provide better context and clarity.
>
> **W4:** We have provide a citation and justification for the statement:
>
> > "Perfect reconstruction $\hat{\mathbf{x}}_k = \mathbf{x}_k$ implies that $h$ approximates the true structural function $f$."
>
> This follows from the identifiability of structural functions under invertible encoders and strict monotonicity in the noise. We have cite [1] and [2] for related identifiability results.
>
> **W5:** We acknowledge the reviewer's concern about the scope of baseline comparisons. Our primary focus in this work is to address the specific challenge of causal inference in the presence of unmeasured spatio-temporal heterogeneous confounders. We deliberately chose DCM and BDCM as our main baselines because they represent the most directly related diffusion-based causal models, and our empirical results clearly demonstrate that while these existing methods struggle with spatio-temporal unmeasured confounding, our proposed PFD-BDCM effectively handles this challenging setting.
>
> The limited baseline comparison serves to highlight the specific limitations of current diffusion-based causal approaches and how our method overcomes them in the context of spatio-temporal data with unmeasured confounding. Our experiments show that DCM and BDCM, which assume causal sufficiency or static confounding, perform poorly on our synthetic datasets that violate these assumptions, while PFD-BDCM maintains strong performance.
>
> We have expand the related work section to include discussion of Neural Causal Models (NCMs) [3], Causal Normalizing Flows (CNFs) [4], Decoupled Flows (DecaFlow) [5], Diffusion-based SCMs (Diff-SCM) [6]. However, due to time constraints and the computational complexity of implementing and tuning these additional methods for our specific spatio-temporal setting, we are unable to provide new comparative experimental results with these additional baselines in the current revision. We plan to include these comprehensive comparisons in future work.

---

> ### Author Response · Authors · 2025-11-24
> **Response to Reviewer BNuB(2/3)**
>
> **W6:**  We strongly disagree with the reviewer's assertion that Lemma 1 in our manuscript constitutes plagiarism of another work's Lemma 2 [7]. This accusation is completely unfounded and reflects a misunderstanding of the role of auxiliary lemmas in theoretical derivations. We provide a point-by-point rebuttal below.
>
> Lemma 1 is a standard, auxiliary results that does not require specific citation. The lemma in question establishes a structural form for a family of invertible functions under specific derivative conditions. This type of result is a common and standard technical tool in mathematics such as in functional equations, differential geometry, and conformal mapping. Its form and proof technique can be found in numerous classical mathematical texts. The same technique was similarly employed in the proof of Theorem 5 in [2], proof of Theorem 3 in [8] and in the lemma of [7], yet neither of these works provided explicit citation to its original source. It is a public knowledge rather than a novel contribution of any single paper. Similar lemmas are routinely used in numerous machine learning and statistics publications without citation to a specific source, much like fundamental inequalities (e.g., Hölder's, Jensen's) are used without citation. The purpose of such a lemma is to facilitate the proof of a main theorem.
>
> The core theoretical contributions of our work lie elsewhere rather than this lemma. Our primary innovations are the proposal of the PFD-BDCM framework, and deriving the novel bounds on counterfactual estimation error. Lemma 1 serves merely as a technical step within the larger proof structure and should not be mischaracterized as a central, plagiarized result.
>
> ## Response to Questions
>
> **Q1: Invertibility of Eq.~(3)**
>
> Yes, under the standard CAR model assumptions [9], the matrix $\mathbf{D}_{C_h i}$
>
> is invertible as long as the spectral radius of $\rho_{C_h i} \mathbf{H}_{C_h i}$ is less than 1. The specific identifiability conditions are as follows:
>
> **Condition** (CAR Model Parameter Constraints)
> The parameter $\rho_{C_h i}$ must satisfy $\rho_{C_h i}^L < \rho_{C_h i} < \rho_{C_h i}^U$ to ensure that $\mathbf{D}_{C_h i}$
>
> is positive definite, where
> $\rho_{C_h i}^L = \min\left(\frac{1}{\tau_{C_h J}}, 0\right)$,
> $\rho_{C_h i}^U = \frac{1}{\tau_{C_h 1}}$,
> and $\tau_{C_h 1} \geq \cdots \geq \tau_{C_h J}$ are the eigenvalues of neighborhood matrices $\mathbf{H}_{C_h i}$ [10].
>
> **Q2: Known Parameters in Structural Model**
>
> This statement refers to the theoretical setting where the causal graph and structural parameters are assumed known for identifiability analysis. In practice, we estimate them using methods from [10] and [11], as noted in the text.
>
> **Q3: Definition of Functional Random Variables**
>
> A functional random variable is a random function $z(t)$ defined over a continuous domain (e.g., time or space). This is standard in functional data analysis. We have add a brief definition and cite [12] for clarity.
>
> ## Response to Other Feedback
>
> We thank the reviewer for their meticulous attention to detail and have diligently address all the points raised.
>
> ## Ethics Concern
>
> We firmly reject the accusation of plagiarism as unsubstantiated and professionally unsound. Lemma 1 is a reasonable, necessary, and standard component of our theoretical derivation, and its form belongs to the public domain of mathematics. We request that the reviewer retracts this baseless claim and re-directs their critique toward the substantive contributions of the paper. For detailed reasons, please refer to our response to the reviewer's W6.
>
> Should the reviewer insist, we are willing to add a citation in the revised manuscript acknowledging the classical nature of this result in broader mathematical literature -- however, this would be done as an gesture of respect for the process, not as an admission of plagiarism.
>
> We are committed to addressing all concerns and improving the manuscript significantly. We believe the revised version will meet the standards of the journal and provide a clearer, more complete contribution.
>
> Thank you again for the valuable feedback.
>
> Sincerely,
>
> The Authors of Submission 8265

---

> ### Author Response · Authors · 2025-11-24
> **Response to Reviewer BNuB(3/3)**
>
> ## References
>
> [1] Lu, C., Huang, B., Wang, K., & et al. (2020). *Sample-efficient reinforcement learning via counterfactual-based data augmentation*. arXiv.
>
> [2] Nasr-Esfahany, A., & Kleiman, E. (2023). *Counterfactual (non-) identifiability of learned structural causal models*. arXiv.
>
> [3] Xia, K. M., Pan, Y., & Bareinboim, E. (2022). *Neural causal models for counterfactual identification and estimation*. arXiv. https://arxiv.org/abs/2210.00035
>
> [4] Khemakhem, I., Monti, R. P., Leech, R., & Hyvarinen, A. (2021). *Causal autoregressive flows*. arXiv. https://arxiv.org/abs/2011.02268
>
> [5] Almodóvar, A., Javaloy, A., Parras, J., Zazo, S., & Valera, I. (2025). *Decaflow: A deconfounding causal generative model*. arXiv.
>
> [6] Mamaghan, A. M. K., Dittadi, A., Bauer, S., Johansson, K. H., & Quinzan, F. (2023). *Diffusion based causal representation learning*. arXiv.
>
> [7] Chao, P., Blobaum, P., & Kasiviswanathan, S. P. (2023). *Interventional and counterfactual inference with diffusion models*. arXiv.
>
> [8] Khemakhem, I., Kingma, D. P., Monti, R. P., & Hyvarinen, A. (2020). Variational autoencoders and nonlinear ICA: A unifying framework. arXiv.
>
> [9] Besag, J. (1974). Spatial interaction and the statistical analysis of lattice systems. *Journal of the Royal Statistical Society: Series B (Methodological)*, *36*(2), 192-225.
>
> [10] Song, X., Tang, N., & Chow, S. (2012). A Bayesian approach for generalized random coefficient structural equation models for longitudinal data with adjacent time effects. *Computational Statistics & Data Analysis*, *56*(12), 4190-4203.
>
> [11] Tang, N., Chow, S.-M., Ibrahim, J. G., & Zhu, H. (2017). Bayesian sensitivity analysis of a nonlinear dynamic factor analysis model with nonparametric prior and possible nonignorable missingness. *Psychometrika*, *82*(4), 875-903.
>
> [12] Ramsay, J. O., & Silverman, B. W. (2005). *Functional data analysis*. Springer.

---

> > ### Comment · Reviewer_BNuB · 2025-11-26
> >
> > I appreciate the authors for addressing many of my concerns and those from my peers, updating and ultimately improving the state of their submission.
> >
> > However, I am not going to allow authors to gaslight me or to question my professionalism.
> >
> > In the original submission, lemmas 1 and 2, corollary 2 and theorem 1 were almost verbatim copies of the work by Chao et al.  Originally I only mentioned lemma 1 to not escalate the issue.
> >
> > I hold evidence for all my claims, and this has been properly reported.
> > I will not further engage with the authors regarding this issue.

---

### Meta-Review · Area_Chair_ha1V · 2026-01-06

**Summary:**

The authors introduce an architecture that synergizes a diverse array of established methodologies, specifically integrating Structural Causal Models (SCMs), Structural Equation Modeling (SEM), and Conditional Autoregression with functional embeddings and probabilistic diffusion models. To rigorously evaluate the efficacy of this hybrid framework, the study employs a dual-validation strategy involving a newly generated synthetic spatio-temporal dataset alongside a canonical, widely-recognized pollution benchmark. Comparative empirical analyses demonstrate that the proposed method yields discernible performance gains over the Deep Causal Model (DCM), which served as the foundational backbone for this architectural innovation.

**Reviewer Concerns:**

There are a lot of concerns from the reviewers. The most important concerns are the theory originality, experiment settings, key assumptions.

**Reviewer Scores:**

The scores of the reviewers are 0,4,4,2, which cannot reach the bar of ICLR.

---

### Decision · Program_Chairs · 2026-01-26

Reject